# De novo emergence of adaptive membrane proteins from thymine-rich genomic sequences

Nikolaos Vakirlis [1], Omer Acar [2,3], Brian Hsu [4], Nelson Castilho Coelho [2,3], S. Branden Van Oss[2,3], Aaron Wacholder[2,3], Kate Medetgul-Ernar[4], Ray W. Bowman II [5], Cameron P. Hines[4], John Iannotta[2,3], Saurin Bipin Parikh[2,3], Aoife McLysaght [1], Carlos J. Camacho [2], Allyson F. O'Donnell[3,5*], Trey Ideker [4*] & Anne-Ruxandra Carvunis[2,3*]

Recent evidence demonstrates that novel protein-coding genes can arise de novo from non-genic loci. This evolutionary innovation is thought to be facilitated by the pervasive translation of non-genic transcripts, which exposes a reservoir of variable polypeptides to natural selection. Here, we systematically characterize how these de novo emerging coding sequences impact fitness in budding yeast. Disruption of emerging sequences is generally inconsequential for fitness in the laboratory and in natural populations. Overexpression of emerging sequences, however, is enriched in adaptive fitness effects compared to over-expression of established genes. We find that adaptive emerging sequences tend to encode putative transmembrane domains, and that thymine-rich intergenic regions harbor a widespread potential to produce transmembrane domains. These findings, together with in-depth examination of the de novo emerging *YBR196C-A* locus, suggest a novel evolutionary model whereby adaptive transmembrane polypeptides emerge de novo from thymine-rich non-genic regions and subsequently accumulate changes molded by natural selection.

[1] Smurfit Institute of Genetics, Trinity College Dublin, University of Dublin, Dublin 2, Ireland. [2] Department of Computational and Systems Biology, School of Medicine, University of Pittsburgh, Pittsburgh, PA 15213, United States. [3] Pittsburgh Center for Evolutionary Biology and Medicine, School of Medicine, University of Pittsburgh, Pittsburgh, PA 15213, United States. [4] Department of Medicine, Division of Medical Genetics, University of California San Diego, La Jolla, CA 92093, United States. [5] Department of Biological Sciences, University of Pittsburgh, Pittsburgh, PA 15260, United States. *email: allyod@pitt.edu; tideker@ad.ucsd.edu; anc201@pitt.edu

The molecular mechanisms and dynamics of de novo gene birth are poorly understood[1,2]. It is particularly unclear how non-genic sequences could spontaneously encode proteins with specific and useful capacities. To resolve this paradox, it has been proposed that pervasive translation of non-genic transcripts can expose genetic variation, in the form of novel polypeptides, to natural selection, thereby purging toxic sequences and providing adaptive potential to the organism[3,4]. The genomic sequences encoding these novel polypeptides have been called "proto-genes", to denote that they correspond to a distinct class of genetic elements that are intermediates between non-genic sequences and established genes[3]. Several non-mutually exclusive models of de novo gene birth exist[2]. The proto-gene model is supported by several studies which reported that de novo emerging coding sequences tend to display features intermediate between those observed in non-genic sequences and those observed in established genes; these features include length, transcript architecture, transcription level, strength of purifying selection, sequence composition, structural properties and integration in cellular networks[3,5–9]. Furthermore, pervasive translation of non-genic sequences has been observed repeatedly by ribosome profiling and proteo-genomics[3,10–13], and studies have shown that random sequence libraries can form defined secondary structures and harbor bioactive effects[14–19]. Nonetheless, it remains unknown if, how, how often, and how rapidly, may native proto-genes accumulate adaptive fitness-enhancing changes to become established genes.

Mutations that cause changes to the sequence or expression of established genes are typically constrained by preexisting selected effects—the specific physiological processes mediated by the gene products that are maintained by natural selection[20]. In contrast, emerging proto-genes are expected to mostly lack such constraints because they do not have selected effects. This would leave them more readily accessible to evolutionary changes that have the potential to increase fitness (adaptive changes)[3,4]. We reasoned that this initial potential for adaptive changes would give way as proto-genes mature and the adaptive changes engender novel selected effects, in turn increasing constraints and reducing the possibility of future change. This reasoning is akin to Sartre's "existence precedes essence" dictum[21], and predicts that mutations affecting the sequence or regulation of proto-genes should impact fitness differently than mutations affecting the sequence or regulation of established genes. Specifically, proto-genes are predicted to evolve under weaker constraints, and thereby to display a higher potential for adaptive change, than established genes (Fig. 1a).

In what follows, we confront these theoretical predictions with systematic measurements of how disruption and overexpression of open reading frames (ORFs) impact fitness in budding yeast as a function of the evolutionary emergence status of the ORFs. We find that most emerging ORFs can be disrupted without detectable fitness cost, consistent with a lack of selected effect. Approximately 10% of emerging ORFs show beneficial fitness effects when overexpressed, a 3-fold enrichment relative to established ORFs consistent with a higher potential for adaptive change. In emerging but not established ORFs, beneficial fitness effects are associated with a high propensity to encode transmembrane (TM) domains. Analyses of genome-wide TM propensities led us to hypothesize that novel adaptive TM peptides may spontaneously emerge when thymine-rich non-genic regions become translated: a "TM-first" model of gene birth. The plausibility of this model is supported by a detailed reconstruction of the evolutionary history of one locus where an ORF (YBR196C-A) emerged de novo in a thymine-rich ancestral non-genic region, accumulated substantial changes under positive selection and progressively increased its TM propensity to give rise to a protein that integrates into the membrane of the endoplasmic reticulum (ER) while retaining the potential for adaptive change. Overall, our results support an experiential model for de novo gene birth whereby a fraction of incipient proto-genes can subsequently mature and, as adaptive changes engender novel selected effects, progressively become established in genomes in a species-specific manner.

## Results

**Emergence status.** Two criteria were considered to determine the emergence status of ORFs: whether they appeared to be emerging de novo, and whether they appeared to encode a useful protein product under selective constraints. In keeping with rigorous best practices[1,22,23], young de novo ORFs were identified based on a combination of inter-specific sequence similarity searches (phylostratigraphy) and syntenic alignments. Similarly, ORFs encoding useful protein products were identified stringently based on multiple lines of evidence[1,8,24,25]: inter-specific conservation, translation signatures, length, and evidence of intra-specific purifying selection at the codon level. We classified annotated *S. cerevisiae* ORFs into two categories: emerging ORFs, which appear to have arisen de novo and to lack a useful protein product; and established ORFs, which encode a useful protein product irrespective of whether they emerged de novo or not (Fig. 1b; Supplementary Data 1; Methods). As expected, emerging ORFs tend to be short and weakly transcribed relative to established ORFs (Cliff's Delta $d < -0.7$, Mann-Whitney U-test, $P < 2.2 \times 10^{-16}$ in both cases). Most emerging ORFs (>95%) are annotated as Dubious or Uncharacterized (Methods). Thus, based on these data, there is no evidence that emerging ORFs correspond to canonical protein-coding genes.

**Selected effects.** We compared estimated fitness costs of disrupting emerging and established ORFs. To this end, we first examined fitness estimates generated from a large collection of systematic deletion (non-essential ORFs) and hypomorphic (essential ORFs) alleles[26]. After removing ORFs with genomic locations overlapping other annotated ORFs, we obtained fitness estimates for the disruption of 239 emerging and 4,410 established ORFs (Fig. 1b; Supplementary Data 1). Fitness cost estimates were markedly lower when comparing emerging ORFs to established ORFs, as expected for loci that lack evidence of encoding a useful protein product (Cliff's Delta $d = -0.32$, Mann-Whitney U-test, $P = 1.5 \times 10^{-17}$). For example, only 8% of emerging ORFs were associated with even a small fitness cost ($n = 19$; mutant fitness estimate < 0.9), relative to 29% of established ORFs ($n = 1290$; Odds ratio = 0.2; Fisher's exact test $P < 3.6 \times 10^{-15}$) (Fig. 2a). The true difference in fitness costs between emerging and established ORFs is more pronounced in reality, given that hypomorphic alleles were used for essential ORFs instead of deletion alleles, which would have been lethal. The low fitness cost of disrupting emerging ORFs in laboratory conditions was similar to that of established ORFs with similarly low native expression level (Odds Ratio = 0.6; Fisher's exact test $P = 0.052$), and more pronounced than expected compared to established ORFs of matched length distribution (Odds Ratio = 0.2; Fisher's exact test $P = 2.9 \times 10^{-16}$) (Supplementary Fig. 1a).

We next investigated how the disruption of emerging ORFs impacts fitness in natural conditions by analyzing intraspecific sequence variation across 1011 *S. cerevisiae* isolates[27]. Counting the number of isolates in which the ORF structures (defined as start, stop and reading frame without considering sequence similarity) were intact in each group, we found ORF structures to be markedly more variable across isolates for emerging than established ORFs (Fig. 2b; Supplementary Data 1), including

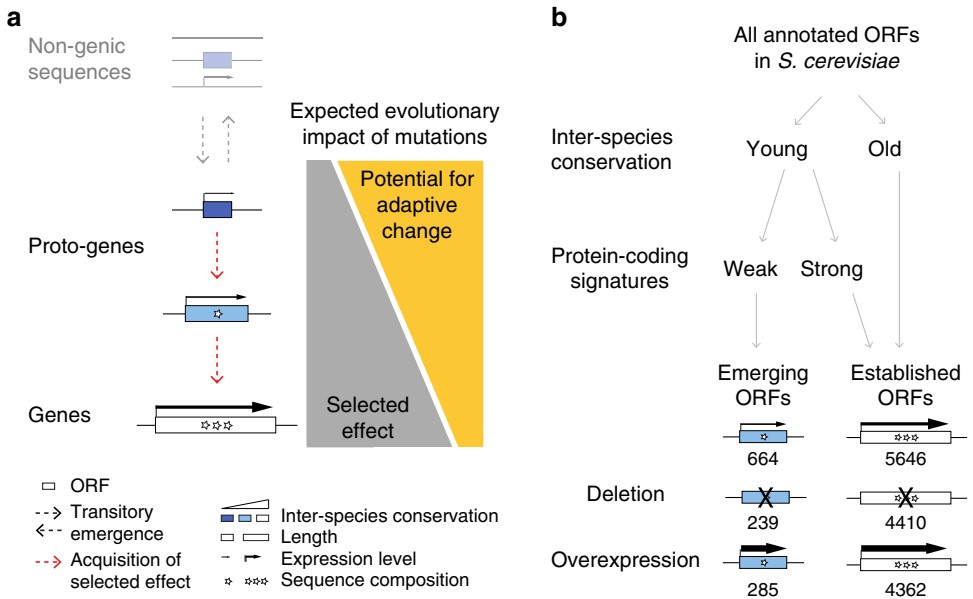

**Fig. 1 Adaptive proto-gene evolution: theory and empirical testing. a** Theoretical model. Left: The evolution from non-genic sequences to proto-genes to genes is represented as in ref. [3]. Sequence composition (stars) refers to the distribution of nucleotides and codons in the sequence (more stars signify a more "gene-like" sequence composition). Left: The transition from non-genic sequences to proto-genes is mediated by gains of ORFs, transcription and translation; the transition from proto-genes to genes occurs along a continuum; the processes governing transitory emergence (gains and losses) of proto-genes through neutral mutations or toxic purging are not investigated in this manuscript (faded out). Right: Our focus is to understand how evolutionary changes to proto-genes impact fitness. Proto-genes are predicted to display an increased potential for adaptive evolutionary change because they are depleted in selected effects relative to established genes. **b** Operational classification of emerging and established ORFs and summary of strain resources. We empirically test the theoretical model by systematically assessing how emerging ORFs impact fitness in budding yeast. Emerging ORFs are young (our inter-species conservation analyses found no detectable homologs outside of the *Saccharomyces sensu stricto* genus and no conserved syntenic homolog in *S. kudriavzevii* and *S. bayanus*) and do not display strong evidence that they encode a useful protein product under intraspecific purifying selection (Methods). Empirical testing of the theoretical prediction (a) involves experimentally measuring the fitness of disruption and overexpression alleles for both classes of ORFs. The numbers of emerging and established ORFs subjected to each analysis are indicated.

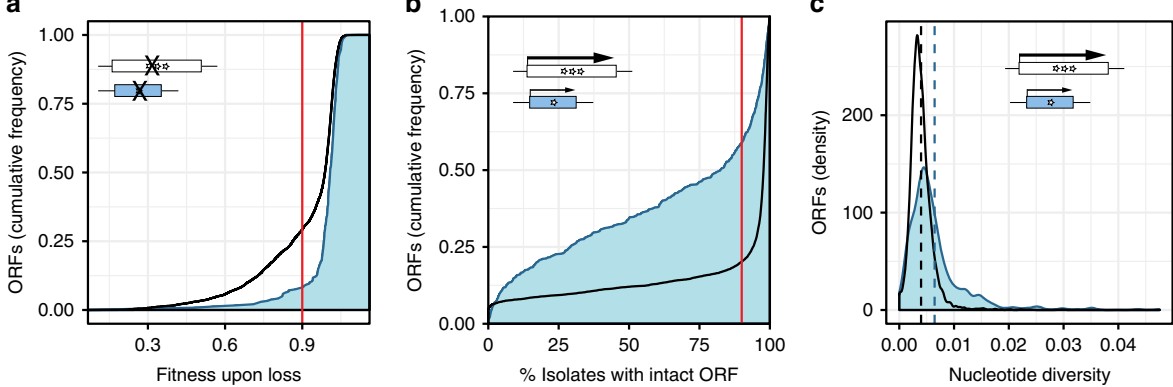

**Fig. 2 Disruption of emerging ORFs is generally inconsequential for fitness. a** Disruption of emerging ORFs imposes lesser fitness costs than disruption of established ORFs. Empirical cumulative distribution function for emerging (blue) and established (black) ORFs; fitness of mutant strains in rich media (YPD) at 30 °C as estimated by ref. [26], averaged over multiple alleles per ORF when applicable. Vertical red line illustrates the fraction of ORFs for each group with fitness effects less than 0.9. Note that the x axis extends beyond 1. **b** ORF structures are more variable for emerging than established ORFs across 1011 *S. cerevisiae* isolates. Empirical cumulative distribution function for emerging (blue) and established (black) ORFs; ORF structure defined as intact in a pairwise alignment if the positions of the start codon and stop codons are maintained, the frame is maintained, and intermediate stop codons are absent. Vertical red line illustrates the fraction of ORFs for each group found intact in less than 90% of isolates. **c** Emerging ORFs display higher nucleotide diversity than established ORFs across *S. cerevisiae* isolates. Density distributions for emerging (blue) and established (black) ORFs; nucleotide diversity estimated over multiple alignments lacking unknown base calls exclusively. Vertical dashed lines represent group means.

established ORFs with matched length and expression level distributions (Odds ratio > 1.8 in both cases; Fisher's exact test $P < 3.1 \times 10^{-13}$ in both cases) (Supplementary Fig. 1b). For example, while 80% of established ORFs were intact in more than 90% of isolates, indicating that they are fixed within the species, this was only the case for 41% of emerging ORFs. Furthermore, estimates of within-species nucleotide diversity were markedly higher for emerging ORFs than established ORFs

(Cliff's Delta $d = -0.34$, Mann-Whitney $U$-test $P = 6.4 \times 10^{-46}$) (Fig. 2c; Supplementary Data 1). Altogether, our results confirmed that disrupting emerging ORFs is generally inconsequential for survival of yeast in both laboratory and natural settings, as expected for loci that lack evidence of encoding a useful protein product, and consistent with a lack of selected effects. The findings in natural isolates, in particular, show that emerging ORFs evolve under weaker selective pressures than established ORFs. It is thus unlikely that emerging ORFs correspond to canonical protein-coding genes whose physiological implications outside of the laboratory remain to be discovered[24,25].

**Potential for adaptive change.** Across kingdoms, one type of evolutionary change that typically accompanies the maturation of young genes is an increase in expression level[28]. It follows that, according to our prediction (Fig. 1a), increasing the expression level of emerging ORFs should increase the organism's fitness more frequently than when the same perturbation is imposed on established ORFs (whose expression levels have presumably been molded by natural selection). Alternatively, if emerging ORFs mostly correspond to spurious non-genic loci with no role in de novo gene birth, increasing their expression level should generally be neutral or toxic, and not provide fitness benefits.

Systematic overexpression screens have been shown to identify adaptive mutations that also occur in laboratory evolution experiments[29]. We thus developed a dedicated overexpression screening strategy to identify ORFs that increased relative fitness upon increased expression. In brief, colony sizes of individual overexpression strains were compared with those of hundreds of replicates of a reference strain with the same genetic background on ultra-high-density arrays (Fig. 3a; Methods). This strategy allowed us to identify ORFs whose overexpression significantly increased colony size relative to the reference ("increased relative fitness"), ORFs whose overexpression significantly decreased colony size relative to the reference ("decreased relative fitness"), and ORFs whose colony sizes were statistically indistinguishable from those of the reference strain ("unchanged relative fitness") (Fig. 3b).

We deployed our screening strategy on a plasmid-based overexpression collection[30] containing 285 emerging ORFs and 4362 established ORFs (Figs. 1b, 3a; Supplementary Data 1), having verified that the presence of an overexpression plasmid did not lead to a detectable growth defect relative to a plasmid-free strain (Supplementary Fig. 2) and that our strategy could detect significant changes in relative fitness with high specificity (False positive rate for increased relative fitness: 0.08%; for decreased relative fitness: 2.65%; Methods). Strains overexpressing 14 emerging and 49 established ORFs displayed increased relative fitness, representing 4.9 and 1.1% of the total number of emerging and established ORFs tested, respectively (Fig. 3b, Supplementary Data 2–3). Overall, overexpressing most ORFs did not significantly change colony sizes relative to the reference strain. Nevertheless, overexpression of emerging ORFs was 4.5 times more likely to increase relative fitness, and 3.1 times less likely to decrease relative fitness, than overexpression of established ORFs (Fig. 3c). The tendency of emerging ORFs to increase fitness when overexpressed was also observed in the context of a pooled competition in the same media (Mann-Whitney U-test, $P = 5.5 \times 10^{-32}$) (Supplementary Fig. 3a). Emerging ORFs associated with increased relative fitness displayed effect sizes ranging from 7.9 to 19% (Supplementary Fig. 3b, Supplementary Data 2). One of the beneficial emerging ORFs identified by our screens was *MDF1* (*YCL058C*), one of the best-studied examples of adaptive de novo origination[31,32].

Expanding our screening strategy to five environments of varying nitrogen and carbon composition (Supplementary Table 1, Supplementary Data 3), we found that strains over-expressing emerging ORFs were consistently 3- to 6-fold more likely to increase relative fitness and 3- to 4-fold less likely to decrease relative fitness, compared to strains overexpressing established ORFs, across all environments tested (Fig. 3d). Notably, while overexpression of only 2.9% of established ORFs increased relative fitness in at least one environment ($n = 126$), this was the case for 9.8% of emerging ORFs ($n = 28$) (Fisher's exact test $P = 1.2 \times 10^{-7}$; Odds ratio: 3.7; Fig. 3e, Supplementary Fig. 4a, b). Sixty percent (17/28) of these adaptive emerging ORFs provided fitness benefits across two or more environmental conditions, rejecting a null model where adaptive emerging ORFs would correspond to stochastic samples of non-deleterious emerging ORFs (empirical $P$-value $< 10^{-5}$; Supplementary Fig. 4b, c). The higher likelihood of observing adaptive effects in emerging ORFs relative to established ORFs could not be explained by their short length or low native expression levels (Fig. 3e).

Disruption of the 28 adaptive emerging ORFs appeared similarly inconsequential for fitness as disruption of other emerging ORFs, both in laboratory and in natural settings ($P > 0.05$ when comparing fitness cost of deletion using Fisher's exact test, ORF intactness using Fisher's exact test and nucleotide diversity across isolates using Mann-Whitney U-test). Furthermore, these ORFs were never found to be toxic in any of the conditions we tested, in contrast with established ORFs which can be toxic in one environment even when increasing fitness in another (Supplementary Data 3). The pronounced differences in how disruption and overexpression of emerging ORFs impact fitness compared to established ORFs are in line with the adaptive proto-gene evolution prediction (Fig. 1a). Overall, our results (Figs. 2, 3) show that overexpression of unconstrained emerging ORFs can provide fitness benefits across multiple environments.

**Beneficial capacities.** The molecular mechanisms that may mediate the beneficial effects we observed remain mysterious. It has been suggested that high levels of intrinsic structural disorder may be associated with adaptive fitness effects[4], and it was recently shown that random sequences with high intrinsic structural disorder have low aggregation propensity and are generally well-tolerated by cells[19]. However, the relationship between disorder and de novo gene birth has been contested[3,8,33–37]. In *S. cerevisiae*, in particular, recently-evolved ORFs are predicted to be less disordered than conserved ones[3,8,34,36] and increasing the expression of disordered proteins causes deleterious promiscuous interactions[38]. We investigated whether the 28 adaptive emerging ORFs identified in our screens exhibited high intrinsic disorder, after verifying that adaptive, neutral and deleterious emerging ORFs presented indistinguishable ORF length distributions (Mann-Whitney U-tests $P > 0.3$ for all comparisons). Disorder predictions suggested that the translated products of the 28 adaptive emerging ORFs were slightly less disordered than neutral and deleterious emerging ORFs (Cliff's Delta $d = -0.25$ and $d = -0.31$, respectively; Mann-Whitney U-test $P = 0.03$ and $P = 0.02$, respectively). Our data (Fig. 4a, Supplementary Fig. 5a, Supplementary Data 1) thus indicate that high disorder is unlikely to be a beneficial capacity that promotes de novo gene birth in *S. cerevisiae*, although it may be in other lineages with differing regulatory systems[39].

In *S. cerevisiae*, young ORFs display high GC content[8] and a high propensity to encode transmembrane (TM) domains relative to ancient ORFs, the latter presumably mediated by a sequence

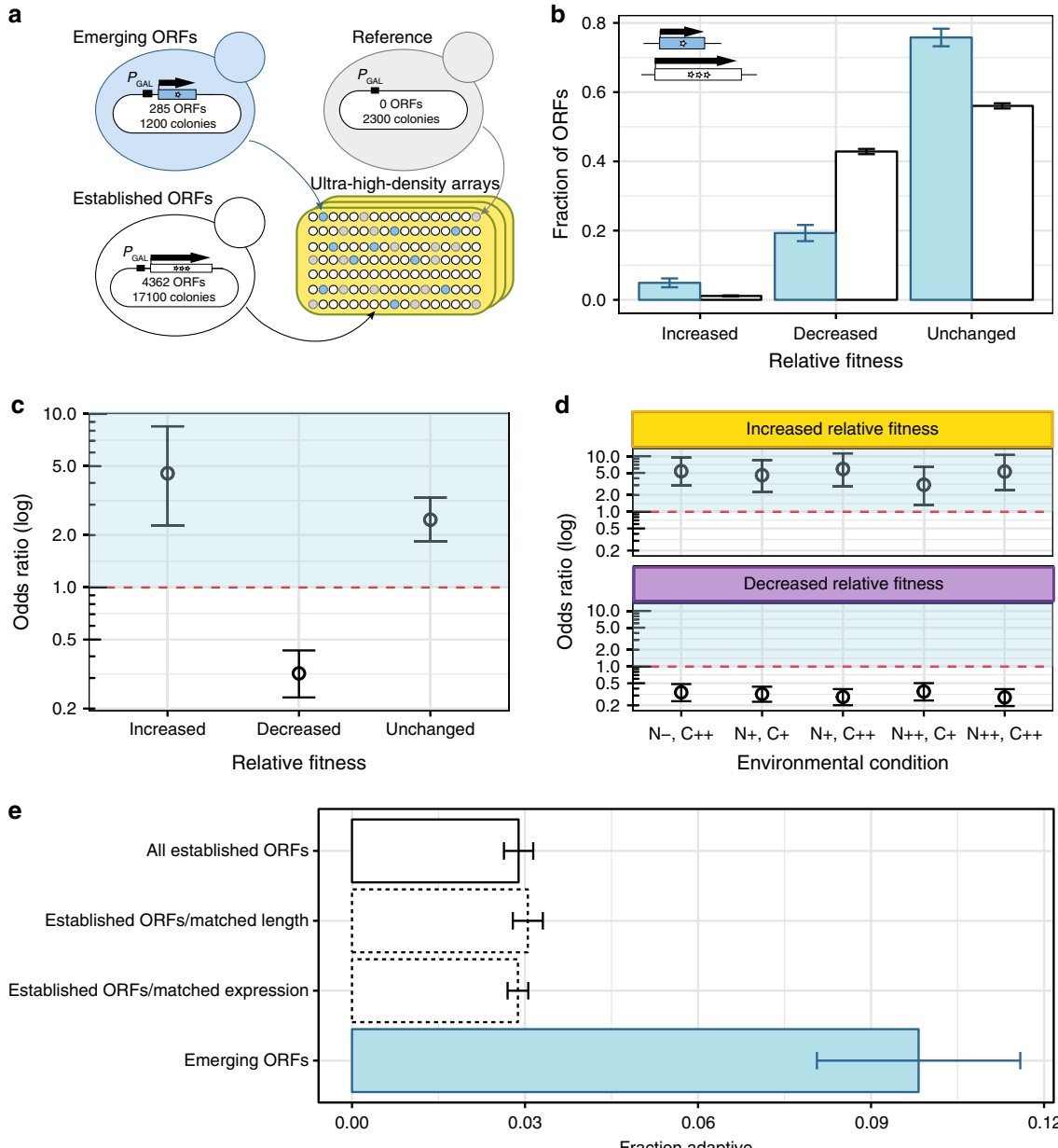

**Fig. 3 Overexpression of emerging ORFs can increase relative fitness. a** Strategy for estimating relative fitness of overexpression strains. Yeast strains overexpressing emerging and established ORFs, and reference strains, are arrayed at ultra-high-density on plates containing agar media. Fitness is estimated from the distributions of colony sizes of technical replicates. Number of colonies are rounded to the nearest hundred. See Methods. **b** Fraction of emerging (blue) and established (white) ORFs displaying increased, decreased and unchanged fitness effects relative to the reference. Environmental condition: SC-URA + GAL + G418 media (Supplementary Table 1). Error bars: standard error of the proportion. Data provided in Supplementary Data 2. **c** Emerging ORFs are 4.5 times more likely to increase relative fitness when overexpressed than established ORFs, and 3.1 times less likely to decrease relative fitness. Odds ratios represent the likelihood of emerging ORFs to increase or decrease relative fitness compared to established ORFs (calculated from **b**). Error bars: 95% confidence intervals. Horizontal dashed line: odds ratio of 1, where the likelihood to display increased, decreased or unchanged fitness relative to the reference strain would be indistinguishable for emerging and established ORFs. All odds ratios are significantly different from 1 (Fisher's exact $P < 0.00002$). **d** Emerging ORFs are consistently more likely to increase fitness and less likely to decrease fitness than established ORFs when overexpressed in five different environments. "N": poor (−), complete (+) or rich (++) supplementation of amino acids; C: complete (+) or rich (++) supplementation of carbon sources (Supplementary Table 1). Odds ratios, horizontal dashed lines, error bars: as in c. All odds ratios are significantly different from 1 (Fisher's exact $P < 0.00002$). Data provided in Supplementary Data 3. **e** Proportion of ORFs displaying increased relative fitness effects in at least one environment (adaptive ORFs). Blue: emerging ORFs (28/285); White with solid contour line: established ORFs (126/4305); White with dashed contour line: established ORFs sampled with replacement according to the distribution of lengths and native expression levels of emerging ORFs. While sampling shorter or less expressed established ORFs did marginally increase the proportion found adaptive, none of these factors was sufficient to explain the high proportion of emerging ORFs found adaptive. Error bars: standard error of the proportion.

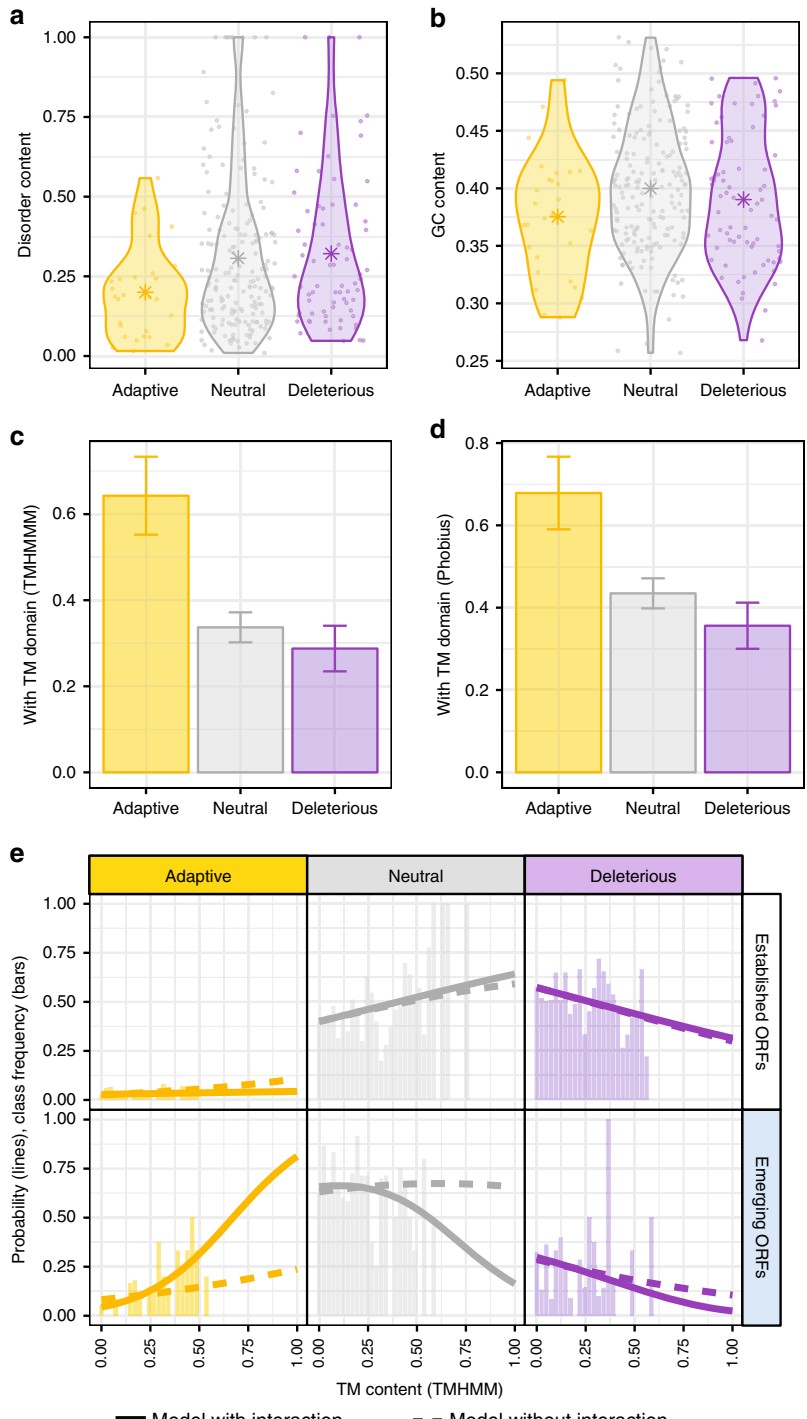

**Fig. 4 TM propensity is associated with beneficial fitness effects in emerging ORFs. a** High disorder is not associated with adaptive fitness effects. The distributions of the fraction of ORF length predicted to encode disordered residues (Disorder content) in adaptive, neutral and deleterious emerging ORFs are shown as violin plots. Adaptive emerging ORFs are less disordered than neutral and deleterious emerging ORFs (Mann-Whitney U-test, $P = 0.03$ and $P = 0.02$, respectively). Stars represent averages of the populations. **b** High GC content is not associated with adaptive fitness effects. The distributions of the fraction of ORF length that is G/C in adaptive, neutral and deleterious emerging ORFs are shown as violin plots. Adaptive emerging ORFs have a lower GC content than neutral emerging ORFs (Mann-Whitney U-test, $P = 0.004$). Stars represent averages of the populations. **c, d** Strong association between high TM propensity and adaptive fitness effects. The fraction of ORFs predicted to contain at least one full TM domain according to TMHMM (**c**) and Phobius (**d**) in adaptive, neutral and deleterious emerging ORFs are shown. TM propensity is significantly greater in adaptive than neutral emerging ORFs in both cases. Error bars: standard error of the proportion. See also Supplementary Fig. 6. **e** Multinomial logistic regression modeling supports a statistical interaction between emergence status and TM content. Frequency of adaptive, neutral and deleterious ORFs as a function of their predicted TM content (TMHMM) is indicated by vertical bars, for established and emerging ORFs separately. Probabilities predicted at 42 values in the range [0, 1] (matching the frequency bins) by multinomial logistic regression models are indicated by lines (full: model with interaction; dashed: model without interaction). The model with interaction between TM content and emergence status is a better fit to the data than the model without interaction.

composition biased towards hydrophobic and aromatic residues[3,9,40]. We investigated whether these properties may promote beneficial fitness effects. GC content was slightly lower in adaptive than neutral emerging ORFs (Cliff's Delta $d = -0.24$; Mann-Whitney U-test $P = 0.04$) but statistically indistinguishable between adaptive and deleterious emerging ORFs (Mann-Whitney U-test $P = 0.33$) (Fig. 4b, Supplementary Fig. 5b, Supplementary Data 1). It is thus unlikely that high GC content promotes de novo gene birth in *S. cerevisiae* by conferring beneficial capacities to the expression products of emerging ORFs.

Adaptive emerging ORFs however displayed a strikingly higher propensity to form TM domains than neutral and deleterious emerging ORFs, according to two prediction algorithms with high specificity and sensitivity, TMHMM and Phobius[41–44]. Comparing the proportion of ORFs with predicted TM domains between adaptive and neutral emerging ORFs yielded Odds Ratio > 2.7 and Fisher's exact $P < 0.025$ for both algorithms; comparisons between adaptive and deleterious emerging ORFs yielded Odds Ratio > 3.7 and Fisher's exact $P < 0.007$ for both algorithms (Fig. 4c, d, Supplementary Data 1). In contrast, there were no significant differences between adaptive, neutral and deleterious established ORFs (Odds Ratio < 1.5 and Fisher's exact $P > 0.06$ for both algorithms; Supplementary Fig. 5c, d, Supplementary Data 1).

Similarly, the fraction of residues predicted as TM over the length of the ORFs (TM content) was significantly associated with fitness benefits in emerging ORFs. Comparing TM content between adaptive and neutral emerging ORFs yielded Cliff's Delta $d > 0.3$ and Mann-Whitney U-test $P < 0.003$ for both algorithms, and comparing TM content between adaptive and deleterious emerging ORFs yielded Cliff's Delta $d > 0.4$ and Mann-Whitney U-test $P < 0.0005$ for both algorithms (Supplementary Fig. 6, Supplementary Data 1). This association was again negligible and insignificant in established ORFs (comparisons between adaptive and neutral or deleterious established ORFs: Cliff's Delta $d < 0.07$ and Mann-Whitney U-test, $P > 0.19$ for both algorithms; Supplementary Fig. 6, Supplementary Data 1).

To formalize these observations, we re-analyzed the relationship between ORF emergence status, TM content and fitness measurements using multinomial logistic regression modeling. Two nested models were fitted to predict experimental relative fitness (categorical: adaptive, neutral, deleterious). In the first one, the predictor variables were TM content (continuous) and ORF emergence status (categorical: emerging, established). In the second one, we added an interaction term between TM content and emergence status. We found that adding this interaction term significantly increased the fit to the data (ANOVA. $P = 0.005$; $D = 10.7$; $df = 2$; Fig. 4e). In a ten-fold cross validation of the two models, the model with the interaction term consistently showed better predictive power than the model without (higher Mathews Correlation Coefficient in all ten iterations, with an average of 0.29 versus 0.23 for the model without the interaction term; Paired Wilcoxon test $P = 0.006$). The interaction term had a coefficient statistically different from zero in the comparison between adaptive and neutral ORFs (Z-test $P = 0.002$) but not in the comparison between neutral and deleterious ORFs (Z-test $P = 0.97$). Thus, this statistical modeling was also consistent with the notion that overexpression of emerging ORFs containing TM domains promotes higher fitness.

**Evolutionary origins of transmembrane domains**. To interrogate the evolutionary origins of these adaptive emerging TM domains, we compared the TM propensities of established and emerging ORFs with those of artificial ORFs corresponding to the hypothetical translation products of intergenic sequences predicted after removing intervening stop codons (iORFs; Supplementary Data 4; Methods). TM propensities were lowest in established ORFs (23% predicted to have a TM domain), intermediate in emerging ORFs (46%) and highest in iORFs (57%) (overall $\chi^2$ test $P < 2.2 \times 10^{-16}$; Fig. 5a). These results, consistent with a previous study[40], held when established ORFs and iORFs were sampled to match the length distribution of emerging ORFs (established ORFs: 21%, iORFs: 56%; overall $\chi^2$ test $P < 2.2 \times 10^{-16}$; Supplementary Fig. 7a).

Interestingly, scrambled control sequences retaining the same length and nucleotide composition as the real genomic sequences displayed high TM propensities (Fig. 5a). Previous analyses encompassing multiple species have shown that GC content negatively correlates with expected TM propensity[34] and that the TM domains of established membrane proteins consist of stretches of hydrophobic and aromatic residues encoded by thymine-rich codons[45]. We thus investigated whether the high thymine content of yeast intergenic sequences[46] may facilitate the emergence of novel polypeptides containing TM domains.

A strong influence of thymine content on TM propensity was observed regardless of ORF emergence status, ORF length, or whether the sequences were real or scrambled (Fig. 5b, Supplementary Fig. 7b, Supplementary Data 4). Established ORFs appeared depleted in TM domains given their thymine content (Fisher's exact $P < 2.2 \times 10^{-16}$, Odds Ratio: 0.54); yet, for those with TM domains, the fraction of sequence length predicted to be in the domains was higher than expected from their thymine content (Cliff's Delta $d = 0.37$, Mann-Whitney U-test, $P < 2.2 \times 10^{-16}$). In contrast, emerging ORFs and iORFs appeared enriched in TM domains relative to their thymine content (iORFs: Fisher's exact $P < 2.1 \times 10^{-16}$, Odds Ratio: 2.1; emerging ORFs: Fisher's exact $P = 0.02$, Odds Ratio: 1.3). We also estimated the TM propensity of small unannotated ORFs that pervasively occur throughout the genome (sORFs). TM propensity in sORFs was also largely driven by their thymine content, and markedly increased when they occupied a larger portion of the intergenic region from which they were extracted (Spearman's rho = 0.92, $P = 4 \times 10^{-8}$; Fig. 5b, c, Supplementary Data 5). Altogether, these results showed that the yeast genome harbors a pervasive TM propensity, facilitated by a high thymine content, and further magnified by additional intergenic sequence properties.

This discovery converges with our finding that overexpressing emerging ORFs with TM domains tends to increase relative fitness (Fig. 4). Together, they suggest the plausibility of a TM-first model of gene birth, whereby thymine sequence biases in intergenic regions that pre-date the acquisition of translation signals may facilitate the emergence of adaptive proto-genes with TM domains (Fig. 5d).

**A TM-first emerging ORF caught in the process of fixation**. We sought to test the plausibility of the TM-first model by retracing the evolutionary history of one specific locus. We focused on *YBR196C-A*, one of the 28 adaptive emerging ORFs identified in our screens. *YBR196C-A* is a 150 nt uncharacterized ORF located on chromosome II with a putative TM domain that accounts for almost half of the protein length (23/49 aa). We could not find published experimental evidence that *YBR196C-A* is natively translated, yet its ORF structure appears stable within *S. cerevisiae* (intact ORF in 95% of isolates).

We visualized cells overexpressing Ybr196c-a-EGFP by confocal microscopy (Methods). The protein colocalized with two markers of the ER membrane: Scs2-TM and Sec13 (Fig. 6a, b,

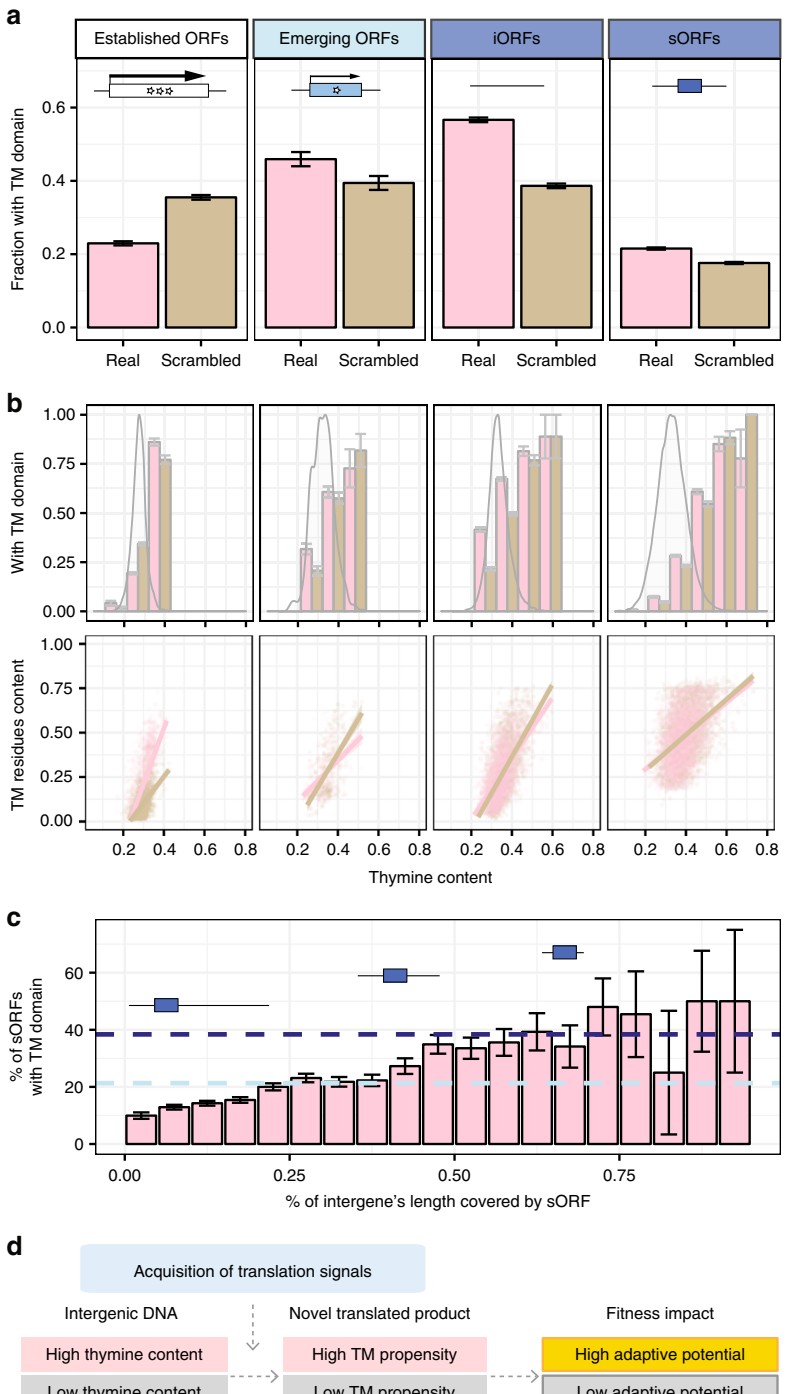

**Fig. 5 Pervasive transmembrane propensity throughout the genome. a** Propensity of ORFs to form TM domains. TM domains predicted by Phobius in this and following panels. Scrambled sequences maintain the same length and nucleotide composition as the real genomic sequences. Intervening stop codons are removed from iORFs and scrambled sequences. sORFs occur pervasively in the genome. Error bars: standard error of the proportion. Data can be found in Supplementary Data 4. **b** Thymine content influences TM propensity. From left to right: established ORFs, emerging ORFs, iORFs, sORFs, as in **a**. Top panel: Bar graph represents the fraction of ORFs that encode a putative TM domain, binned by thymine content (bin size = 0.1) and compared between real (pink) and scrambled (brown) sequences. Error bars represent standard error of the proportion. Overlaid density plots represent the distribution of all sequences per category. Bottom panel: scatterplot showing the fraction of sequence length predicted to be TM residues as a function of thymine content. Only ORFs predicted to encode a putative TM domain are included in the bottom panel. Individual real (pink) and scrambled (brown) sequences are shown in the scatterplot with transparency; points of higher intensity indicate that sequences were sampled multiple times. Linear fits with 95% confidence intervals are shown. Data can be found in Supplementary Data 4. **c** Additional intergenic sequence signals increase the probability that sORFs contain TM domains. Only sORFs between 25 and 75 codons that are fully contained within intergenic regions were included in this analysis. Blue rectangles on horizontal black lines illustrate how sORFs of a given length can occupy a small (left) or large (right) fraction of intergenic sequence length. Horizontal dashed lines represent the fraction of emerging ORFs (light blue) and iORFs (dark blue) between 25 and 75 codons predicted to contain putative TM domains. Error bars represent standard error of the proportion. Data can be found in Supplementary Data 5. **d** A new hypothesis for adaptive proto-gene evolution. Our data suggest that intergenic thymine content influences the TM propensity of novel translated products, which in turn may influence their potential for adaptive change.

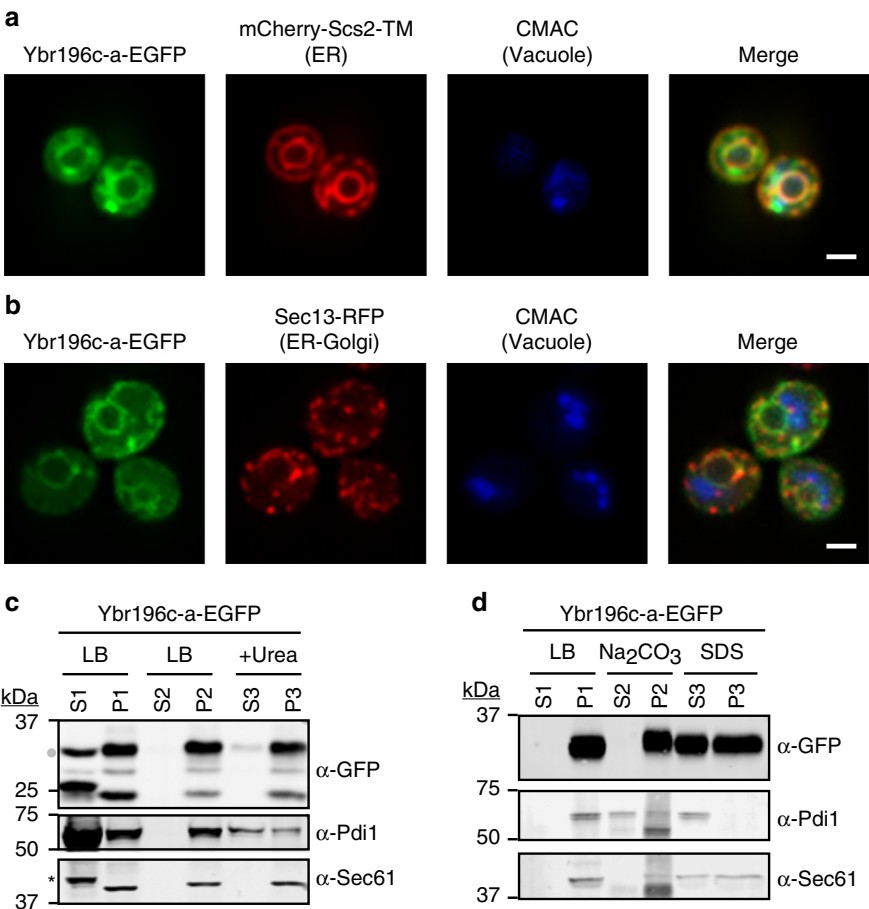

**Fig. 6 TM propensity of the Ybr196c-a protein. a** Ybr196c-a colocalizes at the ER with Scs2-TM. Chromosomally integrated mCherry-Scs2-TM and plasmid-borne Ybr196c-a-EGFP localization was assessed using confocal microscopy. White line is scale bar = 2μ. **b** Ybr196c-a colocalizes at the ER with Sec13. Chromosomally integrated Sec13-RFP and plasmid-borne Ybr196c-a-EGFP localization was assessed using confocal microscopy. White line is scale bar = 2μ. **c** Membrane association assay. Cell lysates were fractionated by centrifugation into a cytosolic fraction in the supernatant (S1) or a pelleted membrane fraction (P1). The pellet was suspended in either lysis buffer (LB) and centrifuged to generate S2 and P2 fractions, or buffer containing 6M urea, which solubilizes peripheral membrane proteins, to generate S3 and P3 fractions. Fractions were then subjected to SDS-PAGE and immunoblotted with anti-GFP (to detect Ybr196c-a), anti-Sec61 (an integral ER-membrane protein), and anti-Pdi1 (an ER luminal protein) antibodies. A dot indicates the full-length Ybr196c-a-EGFP fusion protein and the bands of lower MW are degradation products of this fusion. A star indicates a spurious soluble protein of higher MW than Sec61 that is recognized non-specifically by the anti-Sec61 antibody. Primary uncropped blots are provided in the Source Data file. **d** Carbonate extraction assay. Cellular membranes were treated with a buffer control (S1/P1), Na₂CO₃ (S2/P2) to extract peripherally associated membrane proteins or luminal proteins, such as the ER luminal protein Pdi1, or 1% SDS (S3/P3) to at least partially solubilize integral membrane proteins, such as the Sec61 integral membrane protein control. Integral membrane proteins such as Sec61 and Ybr196c-a remain in the pellet fraction post carbonate treatment (P2), unlike soluble proteins like Pdi1 which shift to the solubilized supernatant (S2). Fractions were assessed by immunoblotting as in Fig. 6c. Primary uncropped blots are provided in the Source Data file.

Supplementary Fig. 8). In a fraction of the cells, the protein also localized to puncta, which colocalized with Scs2-TM but not Sec13 (Supplementary Fig. 8). We did not observe localization at the cell periphery, nor colocalization with mitochondrial, peroxisomal or vacuolar markers (Supplementary Fig. 8). As a control, we visualized using the same methods the protein encoded by another emerging ORF (*YAR035C-A*) which has no predicted TM domains and was not found to be adaptive in our screens. We observed colocalization with the mitochondria (Supplementary Fig. 9). In sum, microscopy confirmed that Ybr196c-a associates with a select subset of cellular membranes.

Next, we used biochemical approaches to ascertain whether Ybr196c-a was merely peripherally associated with the membrane or was a *bona fide* integral membrane protein. We performed membrane association assays using a combination of buffer and centrifugation treatments to define the fraction of Ybr196c-a partitioning with membranes from cell extracts. The majority of

Ybr196c-a pelleted with the membrane fraction (Fig. 6c, compare lanes S1 to P1). Neither washing the pellet in lysis buffer nor treating it with 6M urea removed a significant amount of Ybr196c-a from the membrane (Fig. 6c, compare P1 to P2 and P3), as expected for an integral membrane protein and consistent with the Sec61 control. Ybr196c-a remained in the pelleted fraction after carbonate treatment, as did the Sec61 integral membrane protein control, whereas Pdi1, an ER luminal protein, was released (Fig. 6d, compare P1 to P2). Ybr196c-a and Sec61 only became considerably solubilized and released from the membrane fraction (~50% each) when 1% SDS was added to solubilize membranes (Fig. 6d, compare S3 to P3). Taken together, these biochemical data provide strong evidence that the *YBR196C-A* locus can encode an ER-resident, integral membrane protein.

Having experimentally verified the TM propensity of Ybr196c-a, we retraced its evolutionary origins. Extensive sequence

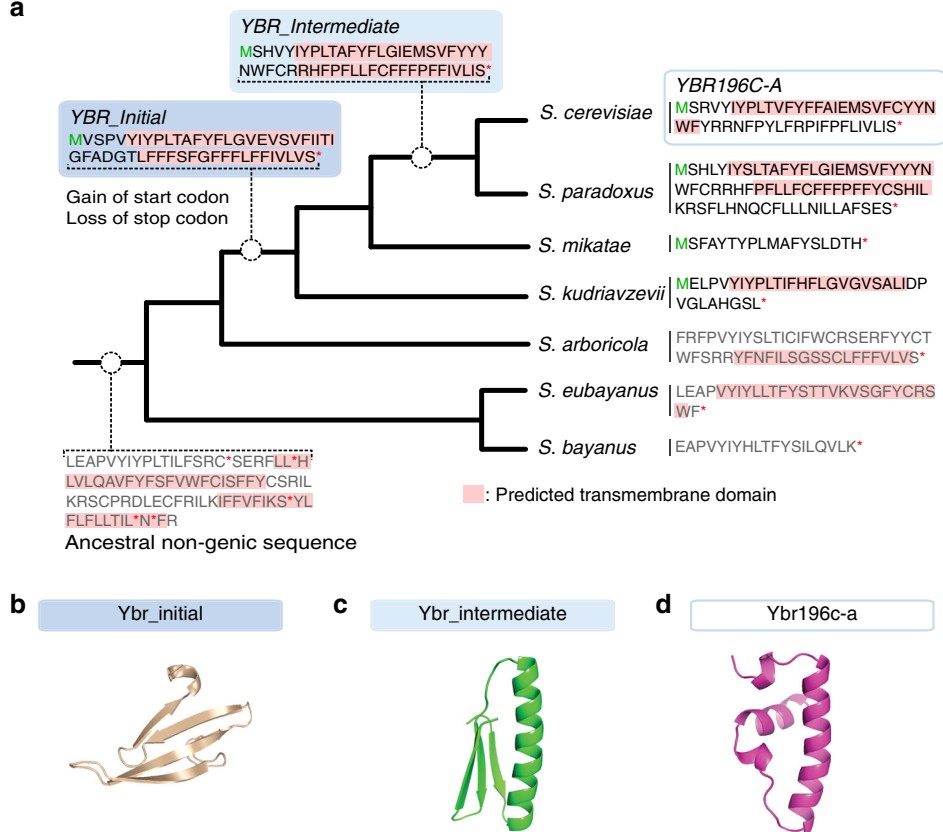

**Fig. 7 Evolutionary history of the *YBR196C-A* locus. a** *YBR196C-A* emerged in an ancestral non-genic sequence with high TM propensity. Results of ancestral reconstruction software are summarized along the *Saccharomyces* genus phylogenetic tree. Branch lengths are not meaningful. Theoretical translations of extant and reconstructed sequences are shown, with ORF boundaries indicated by a green M and a red star. For ORF-less sequences, frames displayed were chosen for illustration purposes. Except for the intergenic ancestor, the translation of the sequence that aligns to the start codon of *YBR196C-A* is shown, in the relevant frame, and until the first stop codon is reached. TM propensity as predicted by Phobius is shown in pink. **b–d** Predicted 3D structures for the translation products of *YBR_Initial* (**b**), *YBR_Intermediate* (**c**) and *YBR196C-A* (**d**). Model 1 predictions by Robetta are shown. *YBR_Initial* is predicted to encode an all-β strand protein. *YBR_Intermediate* is predicted to encode a protein with β-strands and an α-helix. *YBR196C-A* is predicted to encode a protein with a long α-helix and no β-strands.

similarity searches across a broad phylogenetic range (BLAST E-value threshold 0.001; Methods) failed to identify sequences similar to *YBR196C-A* in species beyond the *Saccharomyces* genus, consistent with a recent origin within the past 18 Myrs[47]. Aligning syntenic sequences across six *Saccharomyces* species revealed that ORFs of varying lengths in different reading frames were present in some, but not all, species of the clade, with highly variable primary sequences (Supplementary Fig. 10a). Ancestral reconstruction of the genomic region along the clade (Methods; Fig. 7a) showed that no potential ORF longer than 30 codons was present in the *Saccharomyces* ancestor, in any reading frame (Supplementary Fig. 10b), confirming the de novo origination of *YBR196C-A*.

The initial ORF that became *YBR196C-A* (*YBR_Initial*) likely originated at the common ancestor of *S. kudriavzevii*, *S. mikatae*, *S. paradoxus* and *S. cerevisiae* and already encoded putative TM domains. In fact, the ancestral non-genic sequence at the base of the clade already contained a suite of codons that would have had the capacity to encode TM domains, had it not been interrupted by stop codons (Fig. 7a). This TM propensity persisted in most extant sequences despite substantial primary sequence changes. Consistent with our previous analyses (Fig. 5), *YBR196C-A* is extremely T-rich (48%, 99th percentile of all annotated ORFs) and so are its extant relatives and reconstructed ancestors. The

inferred evolutionary history of the *YBR196C-A* locus was therefore consistent with a TM-first scenario.

*YBR_Initial* underwent major changes in primary sequence including frameshifts, truncations and elongation throughout *Saccharomyces* evolution (Fig. 7a). Furthermore, examination of syntenic loci in five *S. paradoxus* isolates showed that the *YBR196C-A* homolog in this species failed to display an equivalent level of intraspecific constraints as *YBR196C-A* in *S. cerevisiae*. An ORF was present in the syntenic loci of four out of five isolates but it displayed frameshift-induced variation in length and sequence, despite substantial conservation in the TM-containing N-terminal region of the alignment (Supplementary Fig. 10c). Thus, our screening possibly captured *YBR196C-A* in the process of becoming established in the *S. cerevisiae* genome, after going through substantial changes since *YBR_Initial* may have first presented its TM domains to the action of natural selection.

We further determined that adaptive mutations are actively shaping the molecular changes observed in the protein sequence, consistent with our model (Fig. 1). A positive selection test across the four *Saccharomyces* species containing ORFs yielded statistically significant results ($P = 0.01$, LRT: $D = 8.9$, $df = 2$; see Methods) and identified three sites under positive selection. These results were robust to the choice of statistical model (Methods)

and showed slightly increased significance when focusing on the TM region of the alignment (First 30 codons: $P = 0.007$, LRT: $D = 10$, $df = 2$). Furthermore, a pairwise $d_N/d_S$ (omega) calculated over the first 30 codons of the alignment between *S. cerevisiae* and *S. paradoxus* was significantly greater than 1 regardless of the model used (YN: 4.5, LWL85: 1.7, LPB93: 3.8, NG86: 1.9).

To investigate the impact of this rapid sequence evolution at the protein level, we inspected the predicted 3D structures of the putative proteins encoded along the *S. cerevisiae* branch of the phylogenetic tree: Ybr_initial, Ybr_intermediate and Ybr196c-a (Fig. 7). All three shared a conserved predicted TM domain in the N-terminal region, but this domain contained a Gly in Ybr_initial and Ybr_intermediate that mutated to Ala in Ybr196c-a (Supplementary Fig. 10d). A second, Phe-rich, low complexity TM domain was predicted for Ybr_initial and Ybr_intermediate in the C-terminal region of the alignment, presenting Gly and Pro residues (Supplementary Fig. 10d). Gly and Pro are known for their low helix propensity in solution[48] and in a membrane environment[49], and partially hindered the formation of α-helical structures in 3D models generated with the ab initio prediction server Robetta[50]. In fact, the protein models for Ybr_initial were almost all β strands rather than helices (Fig. 7b). Protein models of Ybr_intermediate as well were either mostly β strands (two out of five models) or presented a helix at the N-terminal region and β strands at the C-terminal region (three out of five models; Fig. 7c). In contrast, all models of Ybr196c-a were devoid of β strands and showed α helices flanked by Pro in the N-terminus and a pair of charged Arg in the C-terminus. Four out of five models yielded a robust TM helical structure spanning 20 residues (Fig. 7d). Explicit solvent molecular dynamics simulations of these 3D structures in a membrane bilayer strongly supported the potential for *YBR196C-A* to encode a single-pass membrane-spanning protein (Supplementary Fig. 11). These simulations suggest that evolutionary changes accumulating over millions of years since *YBR_Initial* first emerged have increased the TM propensity at this locus.

To explore this hypothesis, we used confocal microscopy to visualize EGFP-tagged extant and reconstructed homologous ORFs in the family of *YBR196C-A* when they were overexpressed in *S. cerevisiae* cells (Fig. 8; Methods). We observed colocalization with Scs2-TM at the ER membrane and in puncta again for Ybr196c-a, as in Fig. 6a. Similar fluorescence patterns were observed for the *S. paradoxus* homolog of *YBR196C-A*, but not for the *S. mikatae* or *S. kudriavzevii* homologs. The closest reconstructed ancestor of Ybr196c-a and its *S. paradoxus* homolog, Ybr_intermediate, also colocalized with Scs2-TM at the ER membrane and in puncta. In contrast, cells overexpressing Ybr_initial showed generally diffuse cytoplasmic fluorescence and intense puncta, with only exceedingly rare cases where faint ER localization could be discerned. These observations are consistent with the evolutionary and structural models presented in Fig. 7, supporting the hypothesis that *YBR_Initial* has accumulated adaptive changes that increased its TM propensity in the phylogenetic branch that lead to *S. cerevisiae* and *S. paradoxus*.

## Discussion

By combining evolutionary, structural and overexpression analyses of the *YBR196C-A* locus (Figs. 6–8), we provided an unprecedented view of how a thymine-rich intergenic sequence with high TM propensity may, upon acquisition of translation signals, be molded by positive selection into a genuine TM protein with the potential for adaptive change, and mature over millions of years. Future studies are needed to determine in which circumstances Ybr196c-a is natively translated and uncover what

specific activities of the protein are under positive selection. To date, this is the only locus whose evolutionary history has been investigated in enough detail to corroborate a TM-first model of de novo gene emergence (Fig. 5d). The TM-first model is an attractive hypothesis that may explain how sequences that were not translated previously could spontaneously exhibit secondary structures with the potential for adaptive change.

Our analyses suggest that a simple thymine bias suffices to generate a diverse reservoir of novel TM peptides (Fig. 5a–c), and that incipient proto-genes with TM domains are more likely to increase fitness than proto-genes without TM domains (Fig. 4). This could account for the observation that young ORFs have high TM propensities across multiple yeast species[3,40]. Beyond yeast, putative de novo genes with TM domains have also been characterized[51–54]. Furthermore, evidence suggests that the fitness-enhancing capacities of small TM proteins might extend to bacteria as well as to mouse[18,55–57]. Finally, unannotated TM sequences may also be pervasively translated in bacteria, insects and mammals[58–60]. The TM-first model could therefore represent a prevalent route of molecular innovation across phyla. The membrane environment might provide a natural niche for novel TM peptides, shielding them from degradation by the proteasome, and allowing subsequent evolution of specific local interactions while reducing the potential for deleterious promiscuous interactions throughout the cytoplasm. The TM domains may be lost in the subsequent stages of de novo gene evolution. We hope that future studies will quantify the prevalence of the TM-first mechanism and investigate the many exciting questions it raises.

How might expression of a TM proto-gene confer a growth advantage? No current model anticipated this and, to our knowledge, this is the first report of a structural feature being empirically associated with adaptiveness in the context of de novo emergence besides life-saving antifreeze glycoproteins in polar fishes[61]. One might consider several speculative models. Expression of TM proto-genes may cause a preconditioning stress, modestly inducing the unfolded protein response, or the expression of heat-shock proteins or other protein chaperones. This type of preconditioning stress has been shown across species to confer a benefit in responding to subsequent exposure to stressors[62–66], which might extend more generally to nutrient-related stresses such as the ones used in our experiments[67]. Novel TM domains might also associate with larger membrane proteins, such as transporters or signaling proteins, mediating beneficial rewiring of the cellular network[68,69]. Alternatively, insertion of proto-gene TM domains could alter the biophysical properties of the lipid bilayer itself, slowing diffusion of lipids within the bilayer, reducing diffusion of molecules across the membrane or altering membrane curvature through molecular crowding[70–72]. No single model likely explains their adaptive effects universally.

Why are most adaptive proto-genes not fixed within *S. cerevisiae*? It is estimated that adaptive mutations resulting in a 10% fitness increase would reach 5% of the population in ~200 generations and fix in ~500 generations[29]. Given that the adaptive effects we observed were in this range (Supplementary Fig. 3), why haven't increases in expression levels been selected for? Part of the answer likely relates to the artificial nature of our screening strategy. The highly increased expression levels triggered by our plasmid-based system may be unattainable from single regulatory mutations, and weaker changes in expression levels might be outcompeted by other genomic mutations in the wild. It is also quite likely that sequences found to be beneficial in our specific laboratory growth conditions may not have the same effects in natural environments. However, it is also possible that the circumstances that would enable these adaptive effects to manifest in nature simply have not occurred yet since these ORFs emerged.

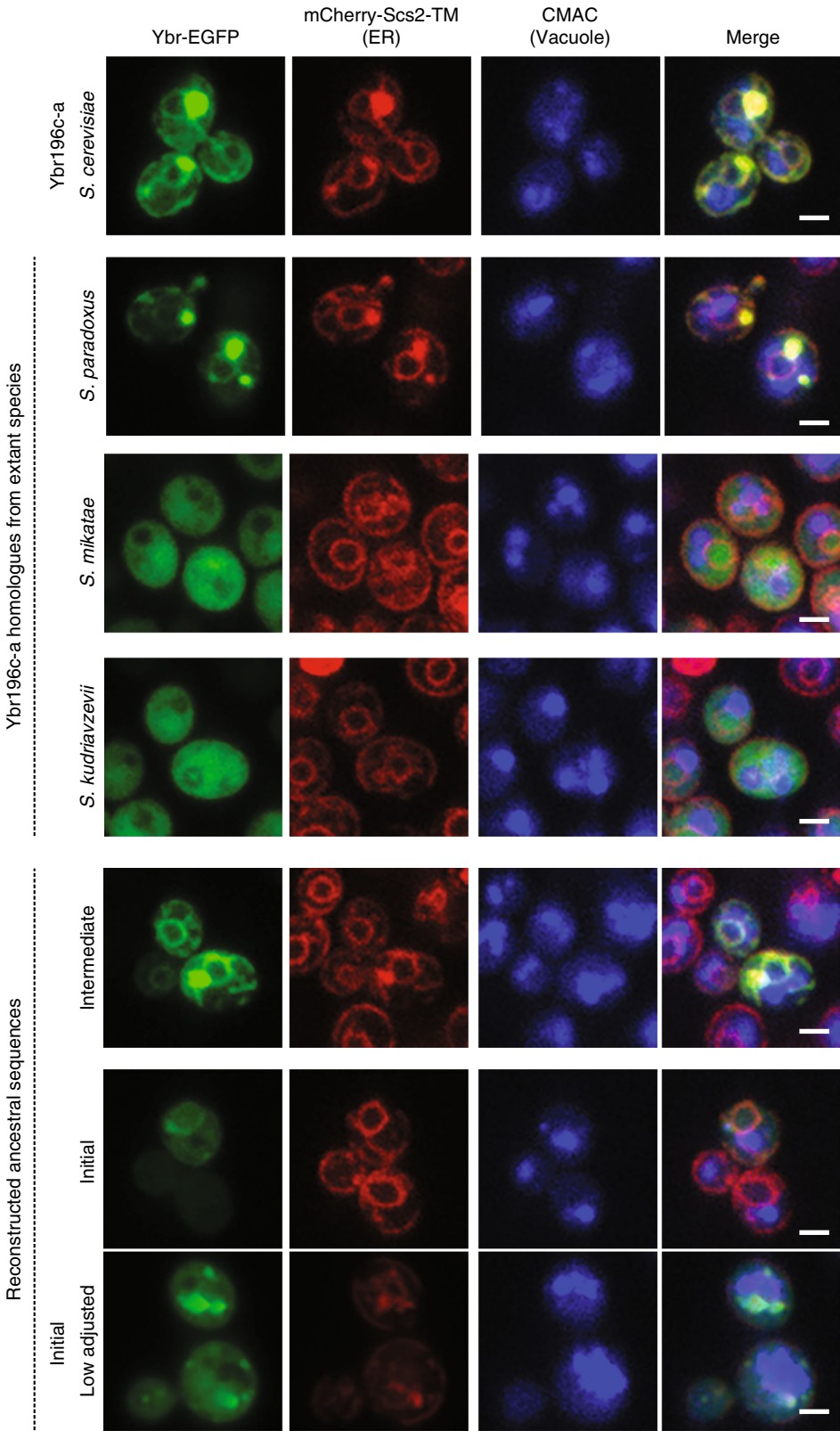

**Fig. 8 Evolution of TM domain propensity throughout the process of de novo emergence.** The sequences of extant and reconstructed homologs of *YBR196C-A* (Fig. 6a) were fused to EGFP and expressed in *S. cerevisiae* cells with chromosomally integrated mCherry-Scs2-TM (Methods). A representative micrograph taken with consistent imaging and adjustments is shown for each, except for the strain overexpressing *YBR_Initial*, for which two micrographs are shown. The top one uses consistent imaging and adjustments, but it is not representative. Instead, it shows a rare instance where colocalization with the ER is observed. The bottom one is representative for this strain, with largely cytosolic fluorescence and a few puncta. It is adjusted differently than the other micrographs because the cytosolic signals were significantly higher than those for other strains. The label "Ybr-EGFP" indicates the imaging column where homologs of *YBR196C-A* are shown. White line is scale bar = 2μ.

Another explanation might be that evolutionary tradeoffs constrain the evolution of adaptive proto-genes. For instance, expressing novel sequences could be beneficial in some environments or life stages of the organism, but deleterious in others—as has been reported for multiple genes in yeast[73,74]. Tradeoffs could also act on the molecular properties of proto-genes as they undergo substantial evolutionary changes over time (Figs. 1 and 7). We showed that certain TM proto-genes can confer fitness-enhancing effects if their expression increases (Figs. 3–4), but what would happen if their length also increased, as is thought to often occur during the maturation of young proteins[2]? This might trigger deleterious fitness effects, since, in yeast, elaborate cellular systems control the insertion and folding of TM proteins and prevent their aggregation[75]. Such tradeoffs could explain why TM propensity is associated with adaptiveness in emerging ORFs, but not in established ORFs (Fig. 4), and why TM domains are generally underrepresented in the established proteome (Fig. 5). We are keen to speculate that evolutionary tradeoffs prevent the majority of incipient proto-genes from reaching established status –translated unconstrained sequences vastly outnumber species-specific established genes across species[3,13,76]—and maintain transitory emerging sequences in the genome for millions of years[6,28,37,77–79]. Future studies are needed to understand why sequences with the potential for adaptive change may never realize this potential.

Overall, our results indicate that increased expression of a substantial proportion of emerging coding sequences, especially those with TM domains, is likely beneficial under some conditions. We expect that this proportion might rise above the 10% observed in this study (Fig. 3), a fraction that already far exceeds observations based on random sequences[17,18,55,80], if emerging sequences were screened across additional conditions and wider ranges of expression levels. Hence, while emerging sequences show no evidence of encoding a useful protein product in the present state of the organism, they have the potential to do so in the future. Might the potential for adaptive change also be carried by the hundreds of emerging ORFs, excluded from this study, that are translated but not annotated in the yeast genome[3]? If this were the case, instead of asking how de novo gene birth is possible given the complexity of useful proteins, we might ask why de novo gene birth and retention is so rare given the pervasiveness of potentially adaptive proto-genes in the genome.

## Methods

**Classification of *S. cerevisiae* ORFs.** Genome annotations for 6310 *S. cerevisiae* ORFs from the *Saccharomyces* Genome Database (SGD)[81] were used keeping a consistent version with ref. [3]. Protein-coding signatures were derived from 3 lines of evidence as in ref. [3]: signatures of translation in ribosome profiling datasets; signatures of intra-species purifying selection estimated from 8 *S. cerevisiae* strains; ORF length longer than expected by random mutations (300nt). ORFs were categorized as young when they (1) had no homologs detectable by phylostratigraphy beyond the *Saccharomyces* clade (conservation level ≤4) as in[3] and (2) had no syntenic homolog with more than 50% sequence similarity in *S. kudriavzevii* or *S. bayanus*.

The synteny analysis was performed as follows. We identified syntenic blocks for young ORFs across four *Saccharomyces* species by using the downstream and upstream genes of the putative young ORFs as anchors. The orthologs of the anchor genes were downloaded from ref. [82] and the genomic sequences were taken from ref. [83] for *S. paradoxus*, and ref. [82] for *S. mikatae*, *S. bayanus* and *S. kudriavzevii*. In cases where a continuous syntenic block could not be constructed between two anchor genes, it was constructed by aligning the ±1 kb region surrounding the anchor gene with the largest number of orthologs (identified by ref. [84]; downloaded from SGD). Multiple alignment of the syntenic blocks were generated using MUSCLE[85] using the default parameters of the msa R package[86].

For each syntenic block, we identified all non-*S. cerevisiae* ORFs that overlapped with the *S. cerevisiae* young ORFs within the ORF ± 200 bp region of the alignment. The overlapping region of the ORFs were extracted from the alignment, translated, and re-aligned pairwise. Similarity between ORFs was calculated by taking the ratio of the number of identical amino acids over the overlapping portion of the alignment divided by the length of *S. cerevisiae* ORF.

All annotated young ORFs (per phylostratigraphy and synteny analysis) that displayed fewer than 3 of the protein-coding signatures were assigned to the emerging ORFs group. All other ORFs (old ORFs and young ORFs with strong protein-coding signatures) were assigned to the established ORFs group. The list of emerging and established ORFs is this study is available in Supplementary Data 1.

We also defined artificial intergenic ORFs (iORFs) and genomic small ORFs (sORFs) as follows. iORFs were generated as in ref. [8]: first, non-annotated genomic regions were extracted from genomic sequences using *bedtools subtract*[87] and the annotation GFF file downloaded from SGD. Then, stop codons in the +1 reading frame were removed, and the sequences in that frame were translated and used to calculate the various properties. sORFs include all non-annotated ORFs that showed no signs of translation from ref. [3], that were longer than 75nt, and did not entirely overlap annotated ORFs in any strand ($n = 18,503$). Coordinates of iORFs and sORFs are available in Supplementary Data 4.

**Description of yeast ORFs.** We used data from external sources, and sequence-based prediction tools, in our analyses. These descriptions can be found together with results of our other analyses in Supplementary Data 1. Annotation status of emerging ORFs was downloaded from SGD. At least 95% of emerging ORFs are annotated as dubious or uncharacterized according to both the 2011 *S. cerevisiae* genome annotation (consistent with ref. [3]) and the current one (R64-2-1). RNA expression levels were lifted from ref. [3], where RNA-seq data in rich media from ref. [88] were re-analyzed. Disruption fitness estimates for individual ORFs were extracted from the summary across a double mutant array at 30 °C from ref. [26], averaging multiple alleles corresponding to the same ORF together. Competitive fitness of overexpression was estimated from read intensity after 20 generations of growth as reported by ref. [30]. Estimates for intrinsic disorder were lifted from ref. [3], where DISOPRED2, a prediction tool with per residue false positive rate of 3.2%[89], was used. GC content was calculated using a python script as in ref. [8]. TM propensities were estimated using two prediction tools: TMHMM and Phobius[41–44]. Two measures of TM propensity were used: the presence of at least one full predicted TM domain, and the fraction of ORF length predicted by the tools to be TM (TM content). For Ybr196c-a and its extant and reconstructed homologs, we performed additional analyses of TM propensities described in the Methods section entitled "Protein structure analyses".

**Yeast strains and growth media.** Barcoded haploid yeast overexpression strains in the BY4741 background from the BarFLEX collection[30] were used for screening purposes together with the reference strain with matched genetic background. The reference strain was created by growing a randomly selected BarFLEX strain on SC + GLU + G418 + 5FOA to chase the plasmid, then transforming the plasmid-less strains with the destination plasmid pBY011. Overexpressed ORFs included in the screen analyses are listed in Supplementary Data 1. All strains were kept in SC-URA + GLU + G418 glycerol stocks at −80 °C in 384-well format until used for screening.

Using the Gateway System® (Life Technologies, Carlsbad, CA), the ORFs studied by microscopy were first cloned into donor vector pDNOR223 using BP recombination (Gateway BP Clonase II Enzyme Mix, Life Technologies) and then transferred to destination vector pAG426GAL-ccdB-EGFP by LR recombination (Gateway LR Clonase II Enzyme Mix, Life Technologies). The expression vectors were then used to transform multiple strains carrying organelle markers that were chromosomally-tagged with fluorescent proteins, using the LiAc/PEG/ssDNA protocol[90] and selected on media lacking uracil. Growth medium composition are detailed in Supplementary Table 1, strain genotypes in Supplementary Table 2 and plasmids in Supplementary Table 3.

**Overexpression strategy to estimate relative fitness.** Using a Singer ROTOR robotic plate handler (Singer Instrument Co. Ltd), overexpression and reference strains where transferred from glycerol archives to agar plates and then robotically combined into 1536-density agar plates (SC-URA + GLU + G418; see Supplementary Table 1). Cells on these plates were then transferred with the same robot at the same density to SC-URA + GAL + G418 where they were incubated for a day. This process was repeated once, following which the cells were robotically transferred to 6144-density agar plates in the screening conditions (in Supplementary Table 1). Throughout this transfer process, five 1536-density source plates were copied 4 times, yielding on average 4 technical replicates per overexpression strain and 3072 replicates for the reference strain per screening condition. Specifically, 768 replicates of the reference strain were arrayed on one out of the five 1536-density plates in an alternating pattern with a replicate of the reference strain in every other row/column. Only four 1536-density source plates can be transferred to a single 6144-density plate, therefore 4 out of the five 6144-density plates contained 768 replicates of the reference strain. The alternating pattern of the reference strain from the 1536-density plate was carried over to the 6144-density plate. As a result, the reference strain was systematically spread throughout the plate, appearing every 4 columns/rows, and each colony replicate of the same overexpression strain was always surrounded by different neighbors. This experimental set up was designed to mitigate neighbor effects, for instance a fast-growing colony negatively impacting the growth of its neighbors.

Colony size was estimated at the time point when the reference strain had reached saturation, which varied depending on the screening condition. Digital images of the plates were acquired in 8-bit JPEG with a SLR camera (18Mpixel Rebel T3i, Canon USA Inc., Melville/NY) with an 18−55 mm zoom lens. A white diffusor box with bilateral illumination and an overhead mount for the camera in a dark room were used. The workflow we used for colony size quantification was similar to[91]: (a) each plate was imaged 3 times to control for technical variability in image acquisition, (b) each image was cropped to the plate, (c) a colony grid at expected density was overlaid on the plate image, (d) each image was binarized by thresholding the ratio between background pixel intensity vs. colony pixel intensity, (e) the number of pixels at each position on the grid was counted, (f) empty positions on the grid were temporarily removed to avoid affecting the spatial normalization in the next steps, (g) the number of pixels at each position on the grid were averaged across all 3 images, (h) spatial normalization and border normalization were applied to remove local, nutrient-based growth effects as in[92], (i) grid values were normalized by the mode of pixel counts per plate to allow comparison of values across plates, (j) empty positions on the grid were added back, (k) outer rows/columns were excluded from downstream analysis (depth of 5 for 6144-density plates) because, even after applying the spatial and border normalization algorithms, reduced border artifacts could still be detected. Normalized colony size was then defined as the average of the normalized number of pixels per grid point corresponding to technical replicates of each overexpression strain. Our measurement of normalized colony size in SC-URA + GAL + G418 were highly correlated with previously published competitive fitness estimates[30] based on barcode signal intensity readings for the same overexpression collection grown in liquid media of the same composition for 20 generations (Pearson Correlation Coefficient: $R = 0.73$, $P < 2.2e^{-16}$). However, our screening strategy allowed to use our quantitative measurements of normalized colony size to identify strains with significantly increased or decreased fitness relative to the reference strain.

For every screen, we binned all screened ORFs in 3 categories (increased fitness, decreased fitness and unchanged) according to their relative fitness compared to the reference strain. The P-values of a non-parametric Mann-Whitney U-test comparing the distributions of technical replicates of each overexpression strain with that of the reference strain were corrected for multiple hypothesis testing using Q-value estimations as defined in[93]. ORFs where categorized as having increased or decreased relative fitness effect when overexpression when the Q-value was lower than 0.01 and the normalized colony size was higher than 95%, or lower than 5%, of the technical replicates reference, respectively. All other ORFs were classified as unchanged. The classification of emerging and established ORFs for each of the five screens is presented in Fig. 3 are reported in Supplementary Data 3.

These results were then integrated as follows: ORFs that increased fitness when overexpressed in at least 1/5 conditions were labeled "adaptive"; ORFs that decreased fitness in at least 1/5 conditions and never increased it in any of the 5 conditions were labeled "deleterious"; all other tested ORFs were labeled "neutral". These labels apply to Figs. 4–5.

We aimed to estimate the proportion of strains from our overexpression collection that could be erroneously found to increase or decreased relative fitness using our screening and analysis strategy. We considered that false positives could arise either due to technical reasons or biological reasons. Technical false positives could stem from inherent variability in our screening platform and colony size analysis, or from our statistical method for identifying strains with significant relative fitness effects. Biological false positives could stem from strains exhibiting true relative fitness effects where the effect would not be linked to the overexpression plasmid. To estimate a false positive rate combining all of these possible error sources, we deployed our strategy in SC + GAL + G418 growth media to a copy of our overexpression collection that we previously exposed to SC + GLU + G418 + 5FOA for 48 h to chase the plasmids out of every strain. We repeated this experiment twice. One of the replicates detected 0 strains as showing any significant relative fitness effect. The second replicate detected spurious increases in relative fitness for 0.15% of the strains and spurious decreases in relative fitness for 5.30% of the strains. On average, we therefore estimate that the combined biological and technical false positive rate of our assay is 0.08% for increases, and 2.65% for decreases, in relative fitness. These estimates are an order of magnitude lower than what is observed when the strains are overexpressing emerging and established ORFs in SC-URA + GAL + G418 (Fig. 3b, c), demonstrating the high specificity of our strategy. The quality of our dataset is further supported by the fact that the same emerging ORFs tend to be found as increasing relative fitness in multiple environments at a rate much higher than expected by chance (empirical $P < 0.00001$; Supplementary Fig. 4c).

**Intraspecific evolutionary analyses**. The VCF file for 1011 S. cerevisiae isolates sequenced by[27] was downloaded from the 1002 Yeast Genome website (http://1002genomes.u-strasbg.fr/files/1011Matrix.gvcf.gz). For each single-exon annotated ORF, nucleotide diversity and ORF intactness (defined as presence of start codon and stop codon in the same reading frame, with no intermediate stops) was derived from the VCF file based on all isolates that had calls for every position in the sequence using custom scripts. The results of these analyses are shown in Fig. 2 and Supplementary Fig. 1.

Genomes of five S. paradoxus isolates (CBS432, N44, UWOPS91917, UFRJ50816, YPS138) were acquired from ref. [94], selected to give a broad representation of S. paradoxus evolutionary diversity. The syntenic region of YBR196C-A was obtained by aligning the sequence between and including the neighboring genes YBR196C and YBR197C among all S. paradoxus strains and S. cerevisiae S288C using MUSCLE. This alignment is shown in Supplementary Fig. 10c.

**Interspecific evolutionary analyses**. Similarity searches using the Ybr196c-a protein sequence as query were performed with TBLASTN against the Saccharomyces genomes and all fungal genomes downloaded from GENBANK, and with BLASTP against the NCBI nr database, using a relaxed E-value threshold of 0.001[95].

To reconstruct the ancestral state of YBR196C-A (Fig. 7a), we first identified and extracted its orthologous regions in all other Saccharomyces species. We exploited SGD's fungal alignment resource to download ORF DNA + 1 kb up/downstream for guiding analyses. A multiple alignment of these sequences was generated using MAFFT[96]. A second, codon-aware alignment was generated with MACSE[97]. Using the MAFFT alignment, a phylogenetic tree was generated with PhyML[98] with the following parameters "-d nt -m HKY85 -v e -o lr -c 4 -a e -b 0 -f e -u species_tree.nwk" where "species_tree.nwk" is the species topology. Ancestral reconstruction was performed with PRANK[99] (on an alignment performed by PRANK, and not the one generated by MAFFT) using the above-mentioned tree as a guide and the parameters "-showanc -showevents –F". The ancestral sequences were extracted from the alignment output file of PRANK, and gaps were removed to obtain the nucleotide sequences, which were then translated into amino acid sequences.

Pairwise dN/dS (omega) was calculated using yn00 from PAML[100]. Selection tests were performed using codeml from PAML. Specifically, the aforementioned codon-aware alignment (regenerated with only the 4 species Skud, Smik, Scer, Spar) and the corresponding PhyML guide tree were used together with the site model to perform the M1a—M2a (model = 0, nsites = 1 and 2) and M7—M8 (model = 0, nsites = 7 and 8) Likelihood Ratio Tests of positive selection, as detailed in the PAML manual. The Bayes Empirical Bayes method at P > 0.99 was used to identify sites under selection.

**Statistical analyses**. All statistical analyses presented in this manuscript can be reproduced using the scripts and Supplementary Data tables provided on github. The statistical analyses involved in the analyses of yeast colony sizes are described in the Methods section "Overexpression screening strategy to estimate relative fitness". All statistical analyses consisting of comparing groups of sequences were performed in R using $\chi^2$, Odds Ratio and Fisher tests (for count data), and Cliff's delta and Mann-Whitney U-tests (for continuous data). Fisher's and Mann-Whitney tests are two-sided.

We performed empirical simulations to statistically control for the distribution of ORFs' length and expression level. In order to obtain a length distribution from a target population (e.g., established ORFs) similar to a template population (emerging ORFs), we performed sampling with replacement using a version of inverse transform sampling as follows. First, the template population's distribution (the emerging ORF distribution in our case) was calculated using bins. Then, ORFs from the target populations were drawn with replacement according to this distribution.

When sampling established ORFs to control for length as presented in Supplementary Fig. 1 and Fig. 3e, the distribution of length of emerging ORFs was calculated using bins of 50 nucleotides, grouping those few between 800 and 2000 nucleotides. When sampling established ORFs to control for expression as presented in Supplementary Fig. 1 and Fig. 3e, the distribution of expression levels of emerging ORFs was calculated using bins of 1 log, grouping those few between 3 and 6 logs. When sampling established ORFs, iORFs and sORFs to control for length as presented in Fig. 5 and Supplementary Fig. 7, the distribution of length of emerging ORFs was calculated using bins of 25 codons, grouping those few between 300 and 650 codons. For sORFs specifically, all instances between 150 and 650 codons were grouped together as long sORFs since sORFs are extremely rare. The data used to perform these analyses are shared in Supplementary Data 1 and 4.

We performed empirical simulations to statistically control for ORFs sequence composition. Scrambling of nucleotide sequences was performed with a custom Python script, as follows: nucleotide positions of the sequence were randomized and whenever an in-frame stop codon was formed from the randomization, its 3 positions were randomized again until they did not form a stop codon. Sequences analyzed in Fig. 5 and Supplementary Fig. 7 were at least 25 codons long. The scrambled ORFs are listed in Supplementary Data 4.

The multinomial logistic regression presented in Fig. 4e and associated text was performed using the multinom function of the nnet R package. The simple model was defined as:

$$\text{fitness\_category} \sim \text{emergence\_status} + \text{TM\_content}$$

The complex model was defined as:

$$\text{fitness\_category} \sim \text{emergence\_status} + \text{TM\_content} + \text{emergence\_status} : \text{TM\_content}$$

The dataset included all 4647 ORFs, classified as adaptive, neutral or deleterious (fitness_category, response variable). The ORFs are also divided into emerging and

established (emergence_status, categorical predictor variable) and associated with a TM content predicted by TMHMM (continuous predictor variable; interpretation of the results does not change if Phobius predictions are used instead). Coefficient *P*-values were calculated using Z-tests. The two models (simple, complex) were compared based on residual deviance using the anova() R function. Ten-fold cross validation was performed as follows. The entire dataset was randomly assigned to ten bins. In each iteration, a bin was excluded to be used as the test set as is. The other nine bins constituted the training data. To overcome the class imbalance in our training data, especially with respect to the adaptive class, we over-sampled the adaptive class to 300 individuals while under-sampling the neutral and deleterious classes to 300 individuals. In each iteration, both models were trained on these artificially balanced training sets and then applied to the test set (which had not been artificially balanced). Mathews Correlation Coefficients were subsequently calculated for each prediction, based on total False Positives, False Negatives, True Positives and True Negatives across the three classes (micro-averaging). For example, False Positives were calculated by summing counts of cases where the prediction was neutral but the true value was not neutral, cases where the prediction was adaptive but the true value was not adaptive and cases where the prediction was deleterious but the true value was not deleterious.

**Protein structure analyses**. Secondary structure for extant and reconstructed *YBR196C-A* homologs was predicted using psiPred[101] (Supplementary Fig. 10d). Since these sequences showed no significant homology with known proteins, an ab initio prediction server, Robetta[50], was used for 3D structure predictions. Translated sequences for *YBR196C-A*, *YBR-Intermediate* and *YBR_Initial* were used as an input for ab initio prediction. For *YBR196C-A* and *YBR_Initial*, 4 out of 5, and for *YBR-Intermediate* 3 out of 5 showed high structural similarity with slight differences in the conformation of the N and C-terminal (Model 1 for each sequence is shown in Fig. 7b–d). Ybr196c-a with the predicted TM domain spanning the membrane and the C-terminal beyond R30R31 outside the membrane was further simulated using molecular dynamics (Supplementary Fig. 11).

We performed molecular dynamics simulations, for which a visual representation is presented in Supplementary Fig. 11. For these analyses, membrane simulation inputs were generated using CHARMM-GUI[102,103]. Molecular dynamics were run using CHARMM36m[104] force field parameters at 303.23 K using Langevin dynamics. The cell box size varied semi-isotropically with a constant pressure of 1 bar using Monte Carlo barostat. Six-step CHARMM-GUI protocol for 225 ps was used for equilibration. Particle Mesh Ewald (PME) was used for periodic boundary conditions (PBC) for evaluation of long-range electrostatic interactions, Lennard-Jones force-switching function used for van der Waals (VDW) and electrostatics calculations with nonbonded cutoff 12 Å. Simulations were run with 2 fs time-step utilizing SHAKE algorithm to constraint hydrogen bonds. All simulations were run for 200 ns using Ambertools 18[105] with Cuda and first 50 ns were disregarded as equilibration time. DPPC[106] was chosen for building the lipid bilayer as PC is highly abundant in yeast membranes[107] and bilayer thickness of around 37 Å is consistent with that of the ER membrane[108]. TIP3P explicit water model, KCl with 0.15M and default water thickness of 17.5 Å were used. Length of X and Y was taken as 75 Å and DPPC ratio was used 10:10, resulting in approximately 83 lipid molecules on both leaflets of the membrane. Terminal group patching applied to N and C terminals of the peptide. The initial position of the TM region of the peptide was oriented using aligned along the Z axis. The system built with replacement method and ions added with Ion Placing method of "distance". Visual Molecular Dynamics[109] and PyMol[110] version 1.8 were used for analysis and visualization of the trajectories.

**Probing the TM propensity of Ybr196c-a and homologs**. Membrane association assay[111] (Fig. 6c) were performed as follows. Yeast strain ARC0011 transformed with C-terminally EGFP-tagged *YBR196C-A* ORF expressed from the *GAL10pr* (see in Supplementary Table 2) were grown to an optical density (OD 600 nm) of 1.0. Fifty $OD_{600}$ units of cells were harvested by centrifugation and pellets were frozen in liquid nitrogen and stored at −80 °C. Cells were resuspended in lysis buffer [20 mM HEPES, (pH 7.4), 50 mM KOAc, 2 mM EDTA, 1 mM DTT, and 0.1M sorbitol plus protease inhibitors (Complete Mini, EDTA-free, Roche)], glass beads were added and agitated on a Disruptor Genie four times for 30 s each with 1 min intervals of recovery between each pulse at 4 °C. Lysates were placed in a clean tube and cleared of unbroken cells by two centrifugations at 3000g for 3 min. The resulting lysate was split into three equal volumes and membranes were pelleted by centrifugation at 21,100g for 20 min at 4 °C. Supernatant and pellet fractions were maintained from one spin and are denoted as S1 and P1, respectively. The remaining two membrane pellets were washed with lysis buffer and recollected by centrifugation at 21,100 x g for 20 min at 4 °C. The pellet from sample 2 was resuspended and mock-treated with lysis buffer while the pellet from sample 3 was resuspended and treated with lysis buffer supplemented with 6M urea for 15 min at 4 °C. Pellets from samples 2 and 3 were recollected by centrifugation and separated as P2 and P3 from their supernatants, S2 and S3, as above. Each sample was TCA precipitated with 1/10 volume of 50% TCA and incubated on ice for 20 min followed by centrifugation at 12,000 x g for 5 min. Precipitated protein was resuspended in 50 µl of SDS/Urea sample buffer [8M Urea, 200 mM Tris-HCl (pH 6.8), 0.1 mM EDTA pH 8.0, 100 mM DTT, and 100 mM Tris base], incubated at 37 °C for 10 min and centrifugation at 12,000 x g for 2 min prior to analyses via SDS-PAGE and immune-blotting. Immunoblots were probed with anti-GFP (sc-9996, Santa Cruz Biotechnology, Inc.), anti-Pdi1 (ab4644, Abcam), and anti-Sec61[112] primary antibodies and either goat-anti-mouse or goat-anti-rabbit IRDye-conjugated florescent secondary antibodies (LI-COR, Lincoln, Nebraska) before imaging on a LI-COR Clx Infrared Imaging system. Primary uncropped blots are provided in the Source Data file.

Carbonate membrane extraction[113,114] was conducted as well (Fig. 6d). The same strain background, plasmid transformations, growth conditions, initial cell harvesting and cell lysis methods were used as described above for the membrane association assay. Lysates cleared from unbroken cells were split into 3 equal volumes, and membranes were pelleted by centrifugation at 21,100g for 20 min at 4 °C. The membrane pellets were resuspended in 30 µl of buffer 88 [20 mM HEPES (pH 6.8), 150 mM potassium acetate, 5 mM magnesium acetate, 250 mM sorbitol and the protease inhibitors described above] and treated with either (1) 500 µl lysis buffer, (2) 500 µl lysis buffer with 0.1M $Na_2CO_3$, or (3) 500 µl lysis buffer with 1% SDS. Samples were incubated on ice for 30 min and subjected to centrifugation at 100,000g for 1 h at 4 °C. The supernatant was removed and set aside for TCA precipitation, while the pellet was resuspended in 500 µl of the appropriate buffer (1, 2, or 3 as listed earlier). Each sample was next TCA precipitated, resuspended, analyzed by SDS-PAGE and immunoblotting with antibodies as described for the membrane association assay above. Primary uncropped blots are provided in the Source Data file.

Cellular localization (Figs. 6a, b, 8; Supplementary Figs. 8, 9) was determined using yeast transformed with C-terminally, EGFP-tagged ORFs (see Supplementary Table 2) expressed from the *GAL10pr* and imaged using a Nikon (Tokyo) Eclipse Ti inverted swept-field confocal microscope (Prairie Instruments, Middleton, WI) equipped with an Apo100x (NA 1.49) objective. Pre-cultures were made in SC-URA + GLU + G418 and then transferred to SC-URA + GAL + G418 for 24 h, prior to imaging. Images were acquired using an electron-multiplying charge coupled device camera (iXon3; Andor, Belfast, United Kingdom) and Nikon NIS-Elements software was used to manipulate image acquisition parameters and post-acquisition processing was done using this same software, ImageJ (National Institutes of Health) and Photoshop (Adobe Systems Inc., San Jose, CA). An unsharp mask was applied in Photoshop to all images. Co-localization of EGFP-tagged emerging ORFs with subcellular membrane-bound organelles was assessed using mCherry-Scs2-TM as an ER-localized marker (Supplementary Table 2), Sec13-RFP as a ER-Golgi marker (Supplementary Table 2), Pex3-RFP as a peroxisomal marker (Supplementary Table 2), MitoTracker™ Red CMXRos (Thermo Fisher Scientific, Waltham, MA) mitochondrial superoxide indicator as a mitochondrial marker, and CellTracker Blue CMAC (7-amino-4-chloromethylcoumarin) dye (Life Technologies, Carlsbad, CA) as a vacuole lumen marker. Cells were incubated with 0.1 µM MitoTracker Red CMXRos for 20 min or 100 µM of CMAC blue for 15 min for mitochondrial or vacuolar staining, respectively, prior to imaging[94]. For *YBR196C-A* and *YAR035C-A*, we verified that the EGFP signal was generated by our plasmid construct by visualizing the mCherry-Scs2-TM, Sec13-RFP and Pex3-RFP strains in SC + GAL + G418 after a pre-culture in SC + GLU + G418.

**Reporting summary**. Further information on research design is available in the Nature Research Reporting Summary linked to this article.

## Data availability

All data generated/analyzed in this study are available in the main text, in the Supplementary Figs. and Tables and as Supplementary Data files. All supplementary data are also on github: https://github.com/annerux/AdaptiveTMproto-genes. The source data underlying Fig. 6c, d are provided as a Source Data file. Strains are available from the corresponding authors upon reasonable request.

## Code availability

Image processing and relative fitness estimations are available at: https://github.com/bbhsu/protogene-analysis. Synteny analyses are available at: https://github.com/oacar/synal. Other analyses are available at: https://github.com/annerux/AdaptiveTMproto-genes.

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

## Acknowledgements

The authors are grateful: to Dr. Brenda Andrews for sharing the BarFLEX collection of overexpression strains; to Andrew MacWilliams, Kate Licon and Dr. Shelly Trigg for technical help in handling mutant collection storage; to Dr. Philip Jaeger for help generating the reference strain used in the screens, and to UCSD undergraduates Nicholas Regent and Manuel Michaca who helped perform screens; to Drs. Graeme Sullivan, Christopher Guerriero and Jeffrey Brodsky for discussions; and to Drs. Gilles Fisher, Christian Landry, Benoit Charloteaux and Zoltan Oltvai for reviewing the manuscript prior to submission. This work was supported by: funds provided by the Searle Scholars Program to A.-R.C.; the National Institute of General Medical Sciences of the National Institutes of Health grants R00GM108865 and DP2GM137422 (awarded to A.-R.C.), P41GM103504 (awarded to T.I.), 5R01GM097084 (awarded to C.J.C.) and F32GM129929 (awarded to B.V.O.); the National Institute of Environmental Health Sciences of the National Institutes of Health grant R01ES014811 (awarded to T.I.); the National Science Foundation MCB CAREER grant 1902859 (awarded to A.F.O.); funding from the European Research Council grant agreements 309834 and 771419 (awarded to AMcL).

## Author contributions

Conceptualization: A.-R.C., T.I., N.V.; Methodology: A.-R.C., B.H., T.I., A.F.O., N.V., A.W., O.A., C.J.C.; Investigation: A.-R.C., N.V., B.H., O.A., N.C.C., B.V.O., A.W., J.I., K.M.-E., R.W.B., C.P.H., S.B.P., A.F.O.; Writing-original draft: A.-R.C., N.V.; Writing-review and editing: A.-R.C., N.V., B.V.O., T.I., A.F.O., B.H., A.W., A.M.cL., N.C.C., K.M.-E., J.I., O.A., C.J.C.; Supervision: A.-R.C., T.I., A.F.O., A.M.cL., C.J.C.

## Competing interests

The authors declare no competing interests.
