## [Peer Review File · Nature Communications]

Reviewers' Comments:

Reviewer #1:

Remarks to the Author:

This study is concerned with the topical and puzzling question of if and how novel protein-coding genes may emerge from previously non-coding DNA and, more importantly, how these novelties become useful and fixed.

Many papers have been published over recent years but several questions are still unanswered with a consistent picture only emerging very slowly from a haze of speculation and contradicting findings.

Vakirlis and coworkers report on an in-depth computational screen on de novo yeast protein-coding genes and find that several novel genes increase fitness and have transmembrane properties. Specifically, Vakirlis et al. present their findings on the effects on fitness of emerging ORFs. These ORFs represent the early stages of potential novel de novo protein-coding genes. The authors find that reduced expression of emerging ORFs is generally not deleterious to the organism. In a small number of cases over expression of emerging ORFs leads to a relative fitness increase. The authors find that these emerging ORFs with beneficial effects preferentially encode transmembrane (TM) domain proteins and speculate that in yeast intergenic regions which preferentially encode such domains are more likely to produce de novo genes. Moreover, they localize the expression to specific regions of the cell.

Generally, this is a highly relevant paper which presents novel insights, is well structured, easy to read, comprehensively backed up by literature and well argued. It will certainly move forward the discourse and have a long lasting impact on the field. It is easy to follow and therefore a good match for Nature Communications as it will reach out to its broad readership. I support publications after a couple of issues which mostly concern the writing have been rectified.

The manuscript is an impressive combination of computational and wet lab work. The results are interesting, the methodology is sound, and the data is presented well. However, I found some of the discussion regarding the model of de novo gene emergence to be somewhat confusing. In particular, I am not sure what benefit the concept of 'adaptive potential' adds to the literature. The field already has an abundance of overlapping terms (e.g. novel gene, orphan gene, proto-gene) and I'm not sure what benefit adding new ones provides.

Several additional literature should be mentioned in context:

First, the studies by Moyers & Zhang. While I would argue that the issue is settled (in favour of de novo gene being real), it is still worth a mention and could be used to frame the importance of this study due to its solid results and comprehensive experiments.

Next, the study by Peisajovich from the Wendel Lim group in Science 2010. They also showed that adding new domains may increase yeast fitness. While this is not related to de novo genes directly, it would be worthwhile the interpretation that -- at least yeast -- is very volatile and permissive and even "gratefully" accepting novel and additional genetic material. Would this be a consequence of its strongly reduced genome?

The recent study by Zhang L (last author M Long, Nat Ecol Evol 2019) in rice which used mass spec data to prove de novo proteins are also translated and under selection (demonstrated there by expression studies).

Finally, the paper by Tretyachenko from the Hlouchova group (Sci Rep 2017) -- see also below.

A couple of citations and the logics of arguments are not precise or even contradictory.

The authors mention that many studies found disorder in de novo genes to be decreasing with age. I would strongly disagree that this is a consensus. In brief, Schmitz et al. (2018) found that the results by Wilson (2017) on that matter are a consequence of using the disorder prediction programmes wrongly, essentially in a way that longer proteins will give lower disorder values.

The study by Bitard-Feildel shows that disorder is reduced in novel domains which emerge by extended reading frames. They found purifying selection in the remainder of the protein which probably has to maintain its structure against the disturbance of the new fragment. This in turn possibly provides added value by unspecific binding in first place (these might even be TMs too? Just an idea.). Here and in several other studies, positive selection (if this is what the authors mean by adaptation -- they should be more precise on this matter all through the ms) has NOT been found. Also, as has been demonstrated by Tretyachenko, random protein sequences tend to be rather rich in secondary structure elements but, quite surprisingly, disorder protects against aggregation and might make them even "fitter".

All this speaks in favour of more ordered de novo proteins, which emerge from low GC regions being more deleterious and weeded out early. This is probably the true background of the "preadaptation" theory but the word preadaptation is of course nonsensical as it is rather a "pre-selection" process which had filtered out potentially viable candidates from an overwhelmingly huge and diverse background of pervasively transcribed raw materials with neutral or close to neutral effects. Later in the paper the emerging ORFs shown to increase fitness are thought to have arisen from intergenic regions with a base composition redispersed to encode transmembrane domains. To my mind, this appears to be a contradiction as emerging ORFs which happen to encode a specific structure are more likely to increase fitness than those with high 'adaptive potential'.

Baseline is that in the introduction the authors should mention more clearly which open questions exist, which contradicting results have been put forward and what is needed to unambiguously detect a de novo protein coding gene: homology at DNA level in outgroups, expression in ingroups but not in outgroups, ideally additional proof with ribosome binding at least and mass spec if possible, structure determination -- even if without reach at the moment, cellular location, fitness effects of KO/KD mutants. This may be many hoops, but the authors jumped quite a few of them and some of the co-authors (McLysaght) have themselves raised these bars (review with Hurst) -- rightly so.

Details on writing:

Throughout -- the use of fitness and adaptation, should be clearly defined. Several statistical tests are used to determine differences. Would it also be wise to include measures of effects size? (e.g. Cliff's delta, Cohen's d)?

Abstract:

2nd sentence: most de novo genes so far have been found in intronic, not in intergenic regions. Sure this is not the case in yeast, yet the statement as is is deceptive.

Cryptic propensity -- I know what the authors mean but I think this phrasing opens a flood gate for criticism and put the paper high up on the list of prime targets for (un-)intelligent designers.

The same goes for "tend to evolve": since the authors have not followed the evolutionary trajectories in detail, this could be misunderstood as insinuating a dominant mechanism creates de novo proteins over gene duplication. As of current knowledge I would be cautious and still argue that over longer time scales duplicated genes tend to prevail with a higher frequency than de novo genes.

page3line8: I disagree that there is an agreement, see above (Wilson, Tretyachenko, etc.).

Adaptive potential is defined in the manuscript as the 'capacity to increase fitness by means of evolutionary change'. I commend the authors for defining this term outright but I am unsure how it aids our understanding of the data presented.

p4Fig:

I found parts of the legend unclear. For example, what is meant by the term 'Sequence composition' and the symbols used to denote it. It is also uncertain what 'Conservation level' refers to. The black dashed arrow between non-genic sequence and proto-genes is not explained.

The model presented appears to suggest that natural selection can convert proto-genes to true genes as well as non-genes, but true genes cannot be converted back into proto genes. Is there reason to assume that this is the case, e.g. by resurrection of a pseudo-gene?

Does the model make an prediction on the timescale over which the transition from non-gene to proto gene to true gene takes place? As emerging ORFs where found in different strains is information on their data of emergence known? At the lest this should be discussed in the Discussion.

page5line11:

what are domain experts? people with knowledge in a field or those with knowledge on some protein domains?

p5l15:

this is a very strong and essential part of the study which I liked a lot, it should feature more prominently in the abstract.

p7l18:

ref 22 is on orphans, not de novo.

l24: ".. outcomes OF laboratory ..."

p9|35:

e.g. here mention Peisajovich, also in the discussion

p10|18:

did not understand this sentence -- please disentangle

p11|7:

see above on my comment on Wilson

p11|8:

"However ..." I agree but this is in contradiction to the authors' statement that there is agreement in the Introduction.

p11|14:

Out data ... indicate ... (data is plural, datum is singular)

p13|20

ref34 is PhD thesis of the first author. I could not find it online and i am not sure if this qualifies as a valid reference.

p14|12

Again a good occasion to cite Peisajovich.

l13

Again, be more careful with phrasing interpretation in a way one might expect a pre-existing plan of what should emerge from the genome (and what possibly not).

The process may well be neutral with respect to the single protein but improve the functionality e.g. of other proteins in the membrane by binding weakly and reversibly.

p16|27

How recent is recent? Yeast phylogeny is rather old unless the strains are considered. Could you give an estimated time in mya?

p16|30

What has been reconstructed?

p18|fig6

Great experiments! But I recommend they are double checked by a yeast expert.

p20|5

A gene can not "actively establish itself" as it has no purpose and no direction. Please remove panglossian glasses.

l19

compared <-- inspected (no formal evaluation has been carried out -- this is just to avoid ambiguous understanding e.g. with structural biologists)

p21|22

What is "others" and why?

p23|11:

Please include latin name for the zebra finch

Reviewer #2:

Remarks to the Author:

Understanding how proteins can evolve de novo is an exciting challenge. In this paper, the authors divided the known ORFs in *S. cerevisiae* into established ORFs and emerging ORFs with emerging ORFs being those with less homology to known genes and weaker protein-coding signatures. Indeed, emerging ORFs were shown to be shorter and lowly transcribed, compared to established genes. Then, the authors measured the fitness effect of overexpressing these ORFs as well as disrupting them. The authors argue that roughly 10% of the emerging ORFs increase fitness when overexpressed (in at least one of the tested conditions) compare with 3% of established genes. In addition, a larger fraction of emerging genes were found to be dispensable, compared to established genes.

Focusing on emerging genes that increased the yeast fitness upon overexpression, the authors found a significant tendency for them to include transmembranal features. The authors also show some compelling evidence to support the TM-first hypothesis in which the TM features actually occur prior to the ORF, simply due to some local sequence biases and only then ORF essential features are added. Although the findings on the TM proto genes begs for a deeper insight into the mechanisms by which the cells benefit from these genes, the authors raise some interesting hypotheses, the most interesting being that the membrane provide a "safe" place for new proteins to evolve without being selected against due to harmful interactions with other functions in the cell.

This paper tackles very important and fundamental questions with a systematic set of experiments and analyses. First how de novo genes might evolve? and second, what is the nature of this (not so small) fraction of genes in the yeast genome that are annotated as dubious or uncharacterized. I think that this paper may be somewhat controversial, but if the authors can address my questions about measuring fitness effects of over-expression then I think that the paper would be a good addition to Nature Communications.

My primary question has to do with the 5% of emerging ORFs that were found to increase fitness when overexpressed. What is the false positive rate in this identification? In particular, if the authors were to analyze the set of emerging ORF strains without overexpressing then what fraction of the ORFs would have a similar (measured) fitness increase? My concern is that overexpressing genes is known to often lead to a fitness defect, so the fact that 5% is greater than 1% is expected and does not on its own mean that the emerging genes have a propensity towards providing a fitness benefit. Instead, the 5% number needs to be compared to some appropriate control set of strains (and the accompanied analysis).

Comments:

1. P. 2 line 11 - "enriched in beneficial effects" might be too strong of a statement as ~10% of these sequences were actually shown to improve fitness and the "beneficial effects" were not characterized in depth. Therefore, a modified statement might be more accurate.
2. P.5 Line 18 - The phrasing could be clearer. It might be confusing to say that the fitness measurements for deletions of emerging genes were 'higher' than those of established genes instead of stating directly that only a small fraction (8%) of emerging genes deletions resulted in decreased fitness while the vast majority of deletions did not affect the fitness, unlike established genes for which 29% of deletions had a detrimental effect on fitness.

Also, when the authors report that disruption of emerging genes had lower effect on the fitness compared with established genes. It might be worth noting that emerging genes are expected to reduce the fitness much less than established genes, especially in benign lab conditions, where their function might not be needed.

3. P.6 line 3 and Fig. 2 – The authors should explain why essential genes (which I assume most in not all of them are established genes) were excluded from the analysis, as in Fig. 2A as the reader expects a peak of essential genes deletions with a fitness value of 0.

4. P.6 line 15 – It might help the readers if the authors add a reference to the mentioned notion that emerging ORFs are primarily young established genes that for which we haven't yet found a beneficial function or the conditions in which they are beneficial.

Also, while the findings from natural isolates indeed argue against the mentioned notion, the findings on deletion of emerging ORFs in laboratory conditions might still be due to a beneficial function that is carried out by the emerging genes, yet not in the tested lab conditions.

5. Fig. 3 – Odds ratio (on Y-axis) might be better if shown in log scale.

Also, as stated previously, the authors should report the false positive rate, i.e. what fraction of genes would anyway be detected to increase fitness.

6. The effect of overexpression on fitness is indeed a good indication for beneficial emerging genes (P.7 Lines 17-23), yet the authors should elaborate on why only ~10% of emerging genes show benefit (in at least one of the conditioned tested) when overexpressed (Fig. 3e). Is this the expected fraction? Or are there reasons that can explain why this fraction might actually an underestimate. In particular, performing more such experiments in diverse conditions is expected to "shift" more emerging genes from the unchanged/deceased bins to increase fitness.

7. P. 9 line 34 – The authors should report the degree of overlap between the genes that were picked up by the pooled experiment to those found by the colony size experiments

8. P. 10 lines 18-24 – The authors should elaborate on why expression levels of beneficial emerging genes have not already increased in evolution. Promoter mutations or duplication of the gene in need are very accessible means in evolution, yet they have not occurred despite the fitness benefit reported by the authors.

9. P. 14 line 3 – please explain what is meant by "non-random TM propensities"

10. P. 19 line 24-25 – Do the authors have a reason to believe that the puncta are not an artefact of GFP aggregation? Also, please explain what does it mean that colocalization was observed with Scs2p, but not with Sec13P.

11. P. 22 lines 18-20 – This explanation is interesting, but the fitness measurements were not done in stressful conditions so this can't explain the benefit of over-expressing the emerging genes, can it?

12. P. 7 line 24 – it should be "outcomes of laboratory evolution".

13. P. 11 end of line 12 - 'that' should be 'than'.

Reviewer #3:

Remarks to the Author:

Review for "De novo emergence of adaptive membrane proteins from thymine-rich intragenic sequences"

First, to provide context to my comments, I have expertise in protein chemistry, evolution, ancestral reconstruction, etc., but I'm not an expert in yeast physiology and genetics.

Overall, I thought the paper was novel, interesting, cleverly planned and likely to advance our understanding of an important aspect of evolution. However, the paper did rely a lot on relatively low accuracy bioinformatics predictions in parts, and there are places where some experimental testing would have enabled stronger, less speculative conclusions to be drawn.

The biggest conclusion (not really in the abstract but hard not to draw from the paper/discussion) seems to be that these emerging ORFs confer some sort of biophysical selective advantage to the cells, in that they are unlikely at this stage in their evolution have any specific biochemical function in signalling/catalysis, etc., but do seem to be useful to have around in some conditions. It would be nice to have stronger support for this - some suggestions are listed below. It makes a lot of sense that the weak selective pressure to keep these emerging TM proteins around could be that they are weakly beneficial to survival by modulating membrane properties or URR, and from this, some can evolve to develop specific biochemical functions. But, I can see why the authors have been conservative because the data here doesn't quite establish this.

Specific comments

1) The abstract sort of evades the more interesting/controversial aspects of the paper and the last sentence just says novel genes with useful biochemical capacities (whatever this means - I think I'd say biophysical) tend to evolve within intergenic loci that have a blueprint for these capacities (this is also vague). I think this could be rewritten to be less vague and maybe more specific to the data in the paper?

2) There are a lot of statistical tests used to support some conclusions, but when the data looks a bit less clear cut there seems to be less enthusiasm for statistical analysis. Fig 3 is a good example. First, I think it is nature policy to present raw data and that bar graphs with error bars are not acceptable. Fig 3BCDE should all be presented with raw data points shown. If appropriate, some statistical analysis should be performed. E.g. in 3C - it is claimed that overexpression is 4.5 times more likely to increase relative fitness. It is not obvious to me what unchanged is in this graph, if it is an odds ratio - what is unchanged? Wouldn't this be 1? Probably there is a simple answer but I was a bit confused. Regardless - it looks like saying overexpression is 4.5 times more likely to increase fitness is a simplification - just guessing by the limited data presented it looks like in many cases it doesn't do much, but sometimes does a lot? This is why simplifications of the data (e.g. average plus CI) can be misleading... same comment goes for all the bar graphs throughout.

3) Page 11 - I'm not sure if beneficial biochemical capacities is the best term. Without any evidence for the ORFs being involved in anything biochemical, it is a reach. Maybe beneficial adaptive capacities? It's not clear whether their beneficial effect is biochemical or biophysical (more later).

4) The conclusion that they are less disordered is based on bioinformatics predictions and isn't a dramatic difference (looks like .2 plus/minus .03 vs .3 +/- .02 or something - a SI stable for this stuff would be nice so people didn't have to guess). The authors have also used standard error measurement for this, which will give the smallest estimated error, vs std dev or 95% CI, etc. They should probably use the same error measurement throughout unless there is good reason to change. Regardless, the difference is not particularly large (again we get no indication of the distribution of data points - is this skewed by a few?) to make this claim. To make a statement like this we really

need to know the predicted accuracy of the algorithm and to then analyse if the predicted difference is large enough given the error of the method that it is significant... then the conclusion can be drawn. Maybe it is? But it would be better for the authors if they make it clear that it is a solid conclusion. Of course, it would also be nice to test this experimentally with a handful of predicted sequences... but I appreciate this is a significant amount of work. It would only be necessary if the difference in the predicted disorder is not clearly significant given the accuracy of the algorithm (and the authors still want to draw this conclusion).

5) "GC content was slightly lower" – some sort of statistical analysis to show whether GC content was within error or GC content was significantly lower.

6) I have an issue with some of the descriptions – maybe these are standard descriptions in the field, in which case ignore these comments – what is TM residue content (nothing came up on google)? Same as the other comparisons, some analysis of the known accuracy of the algorithm vs the difference in the data is required (or experimental analysis).

7) Page 14 – I'm unconformable with the term preadaptation in this context. It's also extrapolating too far to say the blueprint for the capacities (whatever these are) exists in the genome before translation. I see what is being implied but I think given this is based on a lot of bioinformatics and some nice experiments, I'd recommend being more cautious and sticking to descriptive scientific terms (rather than blueprint, etc.).

8) The next section is nice (pg 16) but it should be made clear later that this is an experiment where $n=1$. The data support the hypothesis, but...

9) "Failed to identify sequences similar..." by what criteria? What is similar – 20%, 25%, 35%? This means little without some context – some people might disagree passionately about whether 25% ID is or is not similar.

10) The ancestral sequence reconstruction is used very nicely, I liked this.

11) Fig 6C seems to actually refer to 6A? There are a few small things like this that look as though the paper has been revised but some of these have become disorganized in the process – I'd recommend going through the paper again in detail. For example – there's an orphan MD methods section – but no MD?

12) The authors point out that the predicted TM propensity isn't very accurate... this isn't a great look when the last few pages have been using predicted TM propensities to make some quite significant conclusions... They also highlight that these prediction algorithms are trained on different proteins – which could lower their accuracy in their use for this paper. Anyway, I couldn't actually find where the predicted TM propensity for this protein is listed?

13) It is great that it was tested experimentally. The figure reference is wrong – in text it refers to A/B. YAR035-C comes out of nowhere here – why was it not introduced at the start of this section with YBR196C-A as another test protein, likewise YBR_IO YBR_SP could have been tested. It would be better to say, we selected 35, 196. The way it is written 35, seems to be introduced when convenient – e.g. what are the TM propensities for it? The conclusion to this paragraph again, excludes any mention of 35?

14) Pg 20 rephrase "presented blueprints for TM domains" to more scientific language.

15) The frustrating thing is that 196 and 35 were experimentally tested in terms of their ability to associate with membrane, but then IO and SP were not (they could have been) – instead we rely on predictions with Robetta being used to speculate that 196 evolved to act as a TM protein (but not the others?). These models would be more interesting and make more sense with some experimental data to provide context – e.g. IO doesn't associate much – this makes sense because...

16) Obviously, it would have been nice to have some function of 196, but probably it doesn't have a specific function, so I'm ok with results ending here. However, it would have been really nice to have one more section with some experiments testing the idea that these emerging proteins provide selective benefit in some situations through their biophysical effects on the membrane or URR or...?. Overexpressing synthetic TM helices in yeast and monitoring effect, etc.

17) One problem (maybe I've misunderstood) with the discussion is that it implies TM proteins are overrepresented, BUT when you look at the proteome, they are not... what about all the cytosolic or

membrane associates (not TM) proteins? This point should probably be discussed.

18) A lot is made of Fig 4 and the overexpression results. It would be prudent to make 100% sure this is a statistically significant result.

19) The discussion of the potential effects of the emerging proteins is very interesting, I certainly was very excited by it. But I wish there was some more testing done. Could the authors quickly test whether the overexpression was more beneficial in environments or settings where the cells were more sensitive to membrane properties (presence of detergent?) or the URR (heat stress), etc? I can't help but feel that if they do benefit the membrane biophysics or URR, etc., there must be some way to test this?

20) The last paragraph of the paper is quite long and seems a bit out of place, listing a lot of papers that describe de novo TM domains with various physiological functions. I guess this is to show that they could be doing almost anything – maybe this is right, but the paragraph needs to be altered so it relates more clearly to this work, rather than just listing previous de novo TMs and saying implications of de novo GMs generalize beyond *S cerevisiae* (I understand it is important to make the results general but this somewhat obvious, and the sentence is quite vague). This could almost go into the introduction as a single sentence.

I hope these comments help the authors to improve the manuscript (which I did enjoy reading – apologies if I've misunderstood anything) – I'm happy to see any revision and/or clarify any of my comments. I apologise for suggesting new experiments – I don't see them as essential for acceptance if the strength of the bioinformatics analysis can be more obviously shown (but I think that there are many interesting experiments to do!).

Colin Jackson

RESPONSE LETTER - VAKIRLIS AND COLLEAGUES -

Dear Reviewers,

Thank you very much for the generally positive assessment you expressed about our submitted manuscript. We hope you find our revised manuscript much improved by the insightful comments and suggestions. We are thankful for your guidance, which we feel has helped us to advance this field. In particular, the addition of multiple novel experiments strengthens our conclusions.

*We noticed that similar points were raised independently by multiple Reviewers. In our response letter, we start by addressing these common points in a dedicated section where they are grouped by topic (#1: Abstract; #2: Open questions). We then address the remaining comments one by one in the following section, greying out those comments already addressed in the grouped section. Throughout the document, Reviewers' comments are indicated in **blue font** text to help distinguish it from our responses.*

GROUPED ANSWERS

#1: Abstract

Below please find the suggestions from the three reviewers on how to improve our Abstract:

Reviewer #1:

- Abstract:
2nd sentence: most de novo genes so far have been found in intronic, not in intergenic regions. Sure this is not the case in yeast, yet the statement as is is deceptive.
Cryptic propensity -- I know what the authors mean but I think this phrasing opens a flood gate for criticism and put the paper high up on the list of prime targets for (un-)intelligent designers.
The same goes for "tend to evolve": since the authors have not followed the evolutionary trajectories in detail, this could be misunderstood as insinuating a dominant mechanism creates de novo proteins over gene duplication. As of current knowledge I would be cautious and still argue that over longer time scales duplicated genes tend to prevail with a higher frequency than de novo genes.
- p5l15: this is a very strong and essential part of the study which I liked a lot, it should feature more prominently in the abstract.

Reviewer #2

P. 2 line 11 - “enriched in beneficial effects” might be too strong of a statement as ~10% of these sequences were actually shown to improve fitness and the “beneficial effects” were not characterized in depth. Therefore, a modified statement might be more accurate.

Reviewer #3

- 1) The abstract sort of evades the more interesting/controversial aspects of the paper and the last sentence just says novel genes with useful biochemical capacities (whatever this means – I think I’d say biophysical) tend to evolve within intergenic loci that have a blueprint for these capacities (this is also vague). I think this could be rewritten to be less vague and maybe more specific to the data in the paper?

We wholeheartedly agree with all these suggestions and have written a revised abstract that we find much improved thanks to the Reviewers’ feedback and that we feel adds the clarity the Reviewers were striving for:

“Recent evidence demonstrates that novel protein-coding genes can arise *de novo* from non-genic loci. This evolutionary innovation is thought to be facilitated by the pervasive translation of non-genic transcripts, which exposes a reservoir of variable polypeptides to natural selection. Here, we systematically characterize how these *de novo* emerging coding sequences impact fitness in budding yeast. Disruption of emerging sequences is generally inconsequential for fitness in the laboratory and in natural populations. Overexpression of emerging sequences, however, is enriched in adaptive fitness effects compared to overexpression of established genes. We find that adaptive emerging sequences tend to encode putative transmembrane domains, and that thymine-rich intergenic regions harbor a widespread potential to produce transmembrane domains. These findings, together with in-depth examination of the *de novo* emerging *YBR196C-A* locus, suggest a novel evolutionary model whereby adaptive transmembrane polypeptides emerge *de novo* from thymine-rich non-genic regions and subsequently accumulate changes molded by natural selection”.

We also slightly modified the title of our manuscript in agreement with Reviewer 2’s point about intergenic regions:

“*De novo* emergence of adaptive membrane proteins from thymine-rich genomic sequences”

#2: Open questions

Reviewers 2 and 3 raised important open questions that were insufficiently addressed in the discussion of our submitted manuscript:

Reviewer #2:

- 6. The effect of overexpression on fitness is indeed a good indication for beneficial emerging genes (P.7 Lines 17-23), yet the authors should elaborate on why only ~10% of emerging genes show benefit (in at least one of the conditioned tested) when overexpressed (Fig. 3e). Is this the expected fraction? Or are there reasons that can explain why this fraction might actually an underestimate. In particular, performing more such experiments in diverse conditions is expected to “shift” more emerging genes from the unchanged/deceased bins to increase fitness.
- 8. P. 10 lines 18-24 – The authors should elaborate on why expression levels of beneficial emerging genes have not already increased in evolution. Promoter mutations or duplication of the gene in need are very accessible means in evolution, yet they have not occurred despite the fitness benefit reported by the authors.

Reviewer #3:

- 17) One problem (maybe I've misunderstood) with the discussion is that it implies TM proteins are over represented, BUT when you look at the proteome, they are not... what about all the cytosolic or membrane associates (not TM) proteins? This point should probably be discussed.
- 20) The last paragraph of the paper is quite long and seems a bit out of place, listing a lot of papers that describe de novo TM domains with various physiological functions. I guess this is to show that they could be doing almost anything – maybe this is right, but the paragraph needs to be altered so it relates more clearly to this work, rather than just listing previous de novo TMs and saying implications of de novo GMs generalize beyond *S cerevisiae* (I understand it is important to make the results general but this somewhat obvious, and the sentence is quite vague). This could almost go into the introduction as a single sentence.

We are very grateful for these questions and considerably re-organized our discussion to address these points. We do not claim to have all the answers, but our revised discussion now specifically poses these questions and offers tentative speculations accordingly.

Regarding the number of beneficial emerging sequences we might have expected, we now state that 10 % is probably less than we would find if we probed more experimental contexts and more than what is found with random sequence libraries:

“We expect that this proportion might rise above the 10% observed in this study (**Fig. 3**), a fraction that already far exceeds observations based on random sequences^{17, 18, 55, 79}, if emerging sequences were screened across additional conditions and wider ranges of expression levels”.

Our revised discussion now devotes two paragraphs to the key questions of why the emerging ORFs for which we found adaptive effects have not seen their expression increase in nature already, and, relatedly, if TM domains are beneficial, why are they depleted in the established proteome:

“Why are most adaptive proto-genes not fixed within *S. cerevisiae*? It is estimated that adaptive mutations resulting in a 10% fitness increase would reach 5% of the population in ~200 generations and fix in ~500 generations²⁹. Given that the adaptive effects we observed were in this range (**Supplementary Fig. 3**), why haven't increases in expression levels been selected for? Part of the answer likely relates to the artificial nature of our screening strategy. The highly increased expression levels triggered by our plasmid-based system may be unattainable from single regulatory mutations, and weaker changes in expression levels might be outcompeted by other genomic mutations in the wild. It is also quite likely that sequences found to be beneficial in our specific laboratory growth conditions may not have the same effects in natural environments. However, it is also possible that the circumstances that would enable these adaptive effects to manifest in nature simply have not occurred yet since these ORFs emerged.

Another explanation might be that evolutionary tradeoffs constrain the evolution of adaptive proto-genes. For instance, expressing novel sequences could be beneficial in some environments or life stages of the organism, but deleterious in others – as has been reported for multiple genes in yeast^{73, 74}. Tradeoffs could also act on the molecular properties of proto-genes as they undergo substantial evolutionary changes over time (**Figs. 1, 7**). We showed that certain TM proto-genes can confer fitness-enhancing effects if their expression increases (**Figs. 3-4**), but what would happen if their length also increased, as is thought to often occur during the maturation of young proteins²? This might trigger deleterious fitness effects, since, in yeast, elaborate cellular systems control the insertion and folding of TM proteins and prevent their aggregation⁷⁵. Such tradeoffs could explain why TM propensity is associated with adaptiveness in emerging ORFs, but not in established ORFs (**Fig. 4**), and why TM domains are generally underrepresented in the established proteome (**Fig. 5**). We are keen to speculate that evolutionary tradeoffs prevent the majority of incipient proto-genes from reaching established status – translated unconstrained sequences vastly outnumber species-specific established genes across species^{3, 13, 76} – and maintain transitory emerging sequences in the genome for millions of years^{6, 28, 37, 77, 78, 79}. Future studies will be needed to understand why sequences with the potential for adaptive change may never realize this potential”.

Finally, we have reformulated and shortened the literature review that was at the end of our submitted discussion. We aimed to better highlight the parallels of what we have observed in yeast, as part of our coherent evolutionary study, and what many published findings in other species have reported. These findings in the literature span several disciplines in biology and may appear unrelated, but we argue that in the light of our results they support the idea that the TM-first model could apply beyond yeast:

“Beyond yeast, multiple *de novo* genes with TM domains have also been characterized^{51, 52, 53, 54}. Furthermore, evidence suggests that the fitness-enhancing capacities of small TM proteins might extend to bacteria^{18, 55} as well as to mouse^{18, 55, 56, 57}. Finally, unannotated TM sequences may also be pervasively translated in bacteria, insects and mammals^{58, 59, 60}. The TM-first model could therefore represent a prevalent route of molecular innovation across phyla.”

INDIVIDUAL ANSWERS

REVIEWER #1

Comment 1

This study is concerned with the topical and puzzling question of if and how novel protein-coding genes may emerge from previously non-coding DNA and, more importantly, how these novelties become useful and fixed.

Many papers have been published over recent years but several questions are still unanswered with a consistent picture only emerging very slowly from a haze of speculation and contradicting findings.

Vakirlis and coworkers report on an in-depth computational screen on *de novo* yeast protein-coding genes and find that several novel genes increase fitness and have transmembrane properties.

Specifically, Vakirlis et al. present their findings on the effects on fitness of emerging ORFs. These ORFs represent the early stages of potential novel *de novo* protein-coding genes. The authors find that reduced expression of emerging ORFs is generally not deleterious to the organism. In a small number of cases over expression of emerging ORFs leads to a relative fitness increase. The authors find that these emerging ORFs with beneficial effects preferentially encode transmembrane (TM) domain proteins and speculate that in yeast intergenic regions which preferentially encode such domains are more likely to produce *de novo* genes. Moreover, they localize the expression to specific regions of the cell.

Generally, this is a highly relevant paper which presents novel insights, is well structured, easy to read, comprehensively backed up by literature and well argued. It will certainly move forward the discourse and have a long lasting impact on the field. It is easy to follow and therefore a good match for Nature Communications as it will reach out to its broad readership. I support publications after a couple of issues which mostly

concern the writing have been rectified.

We are extremely grateful for the Reviewer's positive assessment of our manuscript and their thoughtful comments to help improve our work.

Comment 2

The manuscript is an impressive combination of computational and wet lab work. The results are interesting, the methodology is sound, and the data is presented well. However, I found some of the discussion regarding the model of de novo gene emergence to be somewhat confusing. In particular, I am not sure what benefit the concept of 'adaptive potential' adds to the literature.

The field already has an abundance of overlapping terms (e.g. novel gene, orphan gene, proto-gene) and I'm not sure what benefit adding new ones provides.

Thank you very much for these positive remarks regarding the amount and quality of experimental and computational results presented in our manuscript.

Two key words from our submitted manuscript were however unclear to the Reviewer: "proto-gene" and "adaptive potential".

Proto-gene is a term that was introduced by Carvunis et al (Nature 2012) to denote a category of loci that are neither genes nor non-genic sequences, and thus do not fit in binary conceptualizations of the genome. The concept of proto-gene is central to the motivation of our manuscript. To clarify why, we now clearly explain in the beginning of our revised manuscript that the "emerging ORFs" under study cannot be thought of as de novo genes per se, because there is no conclusive evidence that they encode a useful protein (a few yeast de novo genes exist, but we put them in the "established" category):

"We classified annotated *S. cerevisiae* ORFs into two categories: emerging ORFs, which appear to have arisen *de novo* and to lack a useful protein product; and established ORFs, which encode a useful protein product irrespective of whether they emerged *de novo* or not (**Fig. 1b; Source Data 1; Methods**). As expected, emerging ORFs tend to be short and weakly transcribed relative to established ORFs (Cliff's Delta $d < -0.7$, Mann-Whitney U test $P < 2.2 \times 10^{-16}$ in both cases). Most emerging ORFs (>95%) are annotated as Dubious or Uncharacterized (**Methods**). Thus, based on these data, there is no evidence that emerging ORFs correspond to canonical protein-coding genes".

By adaptive potential, we meant the potential to increase fitness by means of evolutionary change, as the Reviewer noted in our submitted manuscript. We are

sorry that the expression was confusing, and its definition was not found helpful. To clarify, we changed this expression in our revised manuscript to “potential for adaptive change”. We also included a clear definition of adaptive change: “evolutionary changes that have the potential to increase fitness” and revised our Fig. 1a to incorporate these clarifications:

We hope these modifications remove any confusion from our revised manuscript.

Comment 3

Several additional literature should be mentioned in context:

First, the studies by Moyers & Zhang. While I would argue that the issue is settled (in favour of de novo gene being real), it is still worth a mention and could be used to frame the importance of this study due to its solid results and comprehensive experiments.

Next, the study by Peisajovich from the Wendel Lim group in Science 2010. They also showed that adding new domains may increase yeast fitness. While this is not related to de novo genes directly, it would be worthwhile the interpretation that -- at least yeast -- is very volatile and permissive and even "gratefully" accepting novel and additional genetic material.

Would this be a consequence of its strongly reduced genome?

The recent study by Zhang L (last author M Long, Nat Ecol Evol 2019) in rice which used mass spec data to prove de novo proteins are also translated and under selection (demonstrated there by expression studies).

November 21, 2019

Finally, the paper by Tretyachenko from the Hlouchova group (Sci Rep 2017) -- see also below.

We are grateful to the reviewer for highlighting these papers. We cite them all in our revised manuscript.

Comment 4

A couple of citations and the logics of arguments are not precise or even contradictory.

The authors mention that many studies found disorder in de novo genes to be decreasing with age. I would strongly disagree that this is a consensus. In brief, Schmitz et al. (2018) found that the results by Wilson (2017) on that matter are a consequence of using the disorder prediction programmes wrongly, essentially in a way that longer proteins will give lower disorder values.

The study by Bitard-Feildel shows that disorder is reduced in novel domains which emerge by extended reading frames. They found purifying selection in the remainder of the protein which probably has to maintain its structure against the disturbance of the new fragment. This in turn possibly provides added value by unspecific binding in first place (these might even be TMs too? Just an idea.). Here and in several other studies, positive selection (if this is what the authors mean by adaptation -- they should be more precise on this matter all through the ms) has NOT been found. Also, as has been demonstrated by Tretyachenko, random protein sequences tend to be rather rich in secondary structure elements but, quite surprisingly, disorder protects against aggregation and might make them even "fitter".

All this speaks in favour of more ordered de novo proteins, which emerge from low GC regions being more deleterious and weeded out early. This is probably the true background of the "preadaptation" theory but the word preadaptation is of course nonsensical as it is rather a "pre-selection" process which had filtered out potentially viable candidates from an overwhelmingly huge and diverse background of pervasively transcribed raw materials with neutral or close to neutral effects. Later in the paper the emerging ORFs shown to increase fitness are thought to have arisen from intergenic regions with a base composition redisposed to encode transmembrane domains. To my mind, this appears to be a contradiction as emerging ORFs which happen to encode a specific structure are more likely to increase fitness than those with high 'adaptive potential'.

We thank the Reviewer for this overview of the controversies surrounding the relationship between disorder and gene birth, and for their summary of how these ideas relate to the “pre-adaptation” theory. We apologize for unintentionally giving the impression that there was a consensus in our submitted manuscript. In our revised manuscript, we make it clear that these are highly debated topics:

“It has been suggested that high levels of intrinsic structural disorder may be associated with adaptive fitness effects⁴, and it was recently shown that random sequences with high intrinsic structural disorder have low aggregation propensity and are generally well-tolerated by cells¹⁹. However, the relationship between disorder and *de novo* gene birth is debated^{3, 8, 33, 34, 35, 36, 37}. In *S. cerevisiae*, in particular, recently-evolved ORFs are predicted to be less disordered than conserved ones^{3, 8, 34, 36} and increasing the expression of disordered proteins causes deleterious promiscuous interactions³⁸”.

We have also removed explicit mentions to the pre-adaptation theory, which is very peripheral to our work.

Finally, we address the contradiction perceived by the reviewer between having a specific structure and having adaptive potential. Indeed, we show that having specific secondary structures (TM domains) seems associated with an increased probability to increase fitness when expression is increased. We are sorry that the definition of adaptive potential in our submitted manuscript was confusing and led to a perceived contradiction. We hope that having reformulated and redefined the concept clarifies that our results support exactly what the reviewer wrote, and erases what was perceived as a contradiction.

Comment 5

Baseline is that in the introduction the authors should mentioned more clearly which open questions exist, which contradicting results have been put forward and what is needed to unambiguously detect a *de novo* protein coding gene: homology at DNA level in outgroups, expression in ingroups but not in outgroups, ideally additional proof with ribosome binding at least and mass spec if possible, structure determination -- even if without reach at the moment, cellular location, fitness effects of KO/KD mutants. This may be many hoops, but the authors jumped quite a few of them and some of the co-authors (McLysaght) have themselves raised these bars (review with Hurst) -- rightly so.

We thank the Reviewer for suggesting these clarifications that we believe have greatly improved our revised manuscript. We now explicitly state that we used the stringent standards set by the field, including those described by McLysaght and Hurst, while also being clear that the set of emerging ORFs we study are not (as far as we can tell) *de novo* genes (emphasis on genes), since there is no evidence that they encode a useful protein product; they are closer to the concept of proto-genes, lacking selected effect but showing the potential for adaptive change.

“Two criteria were considered to determine the emergence status of ORFs: whether they appeared to be emerging *de novo*, and whether they appeared to encode a useful protein product under selective constraints. In keeping with rigorous best practices^{1, 22, 23}, young *de novo* ORFs were identified based on a combination of inter-specific sequence similarity searches (phylostratigraphy) and syntenic alignments. Similarly, ORFs encoding useful protein products were identified stringently based on multiple lines of evidence^{1, 8, 24, 25}: inter-specific conservation, translation signatures, length, and evidence of intra-specific purifying selection at the codon level. We classified annotated *S. cerevisiae* ORFs into two categories: emerging ORFs, which appear to have arisen *de novo* and to lack a useful protein product; and established ORFs, which encode a useful protein product irrespective of whether they emerged *de novo* or not (**Fig. 1b; Source Data 1; Methods**). As expected, emerging ORFs tend to be short and weakly transcribed relative to established ORFs (Cliff’s Delta $d < -0.7$, Mann-Whitney U test $P < 2.2 \times 10^{-16}$ in both cases). Most emerging ORFs (>95%) are annotated as Dubious or Uncharacterized (**Methods**). Thus, based on these data, there is no evidence that emerging ORFs correspond to canonical protein-coding genes”.

Comment 6

Throughout -- the use of fitness and adaptation, should be clearly defined. Several statistical tests are used to determine differences. Would it also be wise to include measures of effects size? (e.g. Cliff's delta, Cohen's d)?

We thank the Reviewer for these useful suggestions, which we followed in revising our manuscript. We now include clear definitions of adaptation (see answers to your comment #2) and specify exactly what is meant by fitness every time the word is used to designate a specific type of fitness rather than the general evolutionary notion. We also include Cliff’s delta and Odds Ratio estimates of effect sizes in every instance where effect size was not already provided in the submitted manuscript.

Comment 7 – Addressed above in Abstract section

Abstract:

2nd sentence: most *de novo* genes so far have been found in intronic, not in intergenic regions. Sure this is not the case in yeast, yet the statement as is is deceptive.

Cryptic propensity -- I know what the authors mean but I think this phrasing opens a flood gate for criticism and put the paper high up on the list of prime targets for (un-)intelligent designers.

The same goes for "tend to evolve": since the authors have not

followed the evolutionary trajectories in detail, this could be misunderstood as insinuating a dominant mechanism creates *de novo* proteins over gene duplication. As of current knowledge I would be cautious and still argue that over longer time scales duplicated genes tend to prevail with a higher frequency than *de novo* genes.

Comment 8

page3line8: I disagree that there is an agreement, see above (Wilson, Tretyachenko, etc.).

Thank you for pointing out this imprecision of language in our submitted manuscript. We did not mean that there was a general agreement in the literature, but that the studies listed in the sentence provide support to the proto-gene model since they show young ORFs with intermediate features between non-genes and genes. We have clarified this in our revised manuscript:

“The genomic sequences encoding these novel polypeptides have been called “proto-genes”, to denote that they correspond to a distinct class of genetic elements that are intermediates between non-genic sequences and established genes³. Several non-mutually exclusive models of *de novo* gene birth exist². The proto-gene model is supported by several studies which reported that *de novo* emerging coding sequences tend to display features intermediate between those observed in non-genic sequences and those observed in established genes; these features include length, transcript architecture, transcription level, strength of purifying selection, sequence composition, structural properties and integration in cellular networks^{3, 5, 6, 7, 8, 9}. Furthermore, pervasive translation of non-genic sequences has been observed repeatedly by ribosome profiling and proteo-genomics^{3, 10, 11, 12, 13}, and studies have shown that random sequence libraries can form defined secondary structures and harbor bioactive effects^{14, 15, 16, 17, 18, 19}. Nonetheless, it remains unknown if, how, how often and how rapidly native proto-genes accumulate adaptive, fitness-enhancing changes to become established genes”.

Comment 9

Adaptive potential is defined in the manuscript as the 'capacity to increase fitness by means of evolutionary change'. I commend the authors for defining this term outright but I am unsure how it aids our understanding of the data presented.

We understand the confusion and address this issue in our revised manuscript as described in answer to your comment #2.

Comment 10

p4Fig: I found parts of the legend unclear. For example, what is meant by the term 'Sequence composition' and the symbols used to denote it. It is also uncertain what 'Conservation level' refers to. The black dashed arrow between non-genic sequence and proto-genes is not explained.

Thank you for pointing out these shortcomings. The figure, legend and caption have been revised accordingly.

a. Theoretical model. Left: The evolution from non-genic sequences to proto-genes to genes is represented as in ref³. Sequence composition (stars) refers to the distribution of nucleotides and codons in the sequence (more stars signify a more “gene-like” sequence composition). Left: The transition from non-genic sequences to proto-genes is mediated by gains of ORFs, transcription and translation; the transition from proto-genes to genes occurs along a continuum; the processes governing transitory emergence (gains and losses) of proto-genes through neutral mutations or toxic purging are not investigated in this manuscript (faded out). Right: Our focus is to understand how evolutionary changes to proto-genes impact fitness. Proto-genes are predicted to display an increased potential for adaptive evolutionary change because they are depleted in selected effects relative to established genes

Comment 11

The model presented appears to suggest that natural selection can convert proto-genes to true genes as well as non-genes, but true genes cannot be converted back into proto genes. Is there reason to assume that this is the case, e.g. by resurrection of a pseudo-gene?

Does the model make an prediction on the timescale over which the transition from non-gene to proto gene to true gene takes place? As emerging ORFs where found in

different strains is information on their data of emergence known? At the least this should be discussed in the Discussion.

The reviewer’s interpretation of the arrows of Fig. 1a is correct: a gene cannot be converted back into a proto-gene. Mutations can trigger the conversion of a gene into a pseudo-gene, which can then resurrect to re-create a novel gene. In that case, though, the novel gene would have evolved from a pre-existing gene (the one that became a pseudo-gene) thus would not be considered a de novo gene. The similarities and differences between proto-genes and pseudo-genes are described in the original paper that proposed the proto-gene model (Carvunis 2012) which is referenced in the figure caption and throughout the manuscript. Although we find these questions fascinating, we chose not to discuss them further here because they are less relevant to the specific results presented in this manuscript. We did however modify the discussion extensively to reflect the suggestions of the majority of the points raised by Reviewers and did not want to add further to our already long discussion.

Thank you also for raising the important question of the timescale at which de novo gene birth occurs. Our model does not make quantitative predictions in this respect. However, we have made changes to the revised manuscript that give estimates of the timescales that our study addresses within the Saccharomyces genus.

“a recent origin within the past 18 Myrs⁴⁷”

Comment 12

page5line11:

what are domain experts? people with knowledge in a field or those with knowledge on some protein domains?

We thank the reviewer for calling our attention to this obscure term. We meant that the annotations are performed by the team of experts in the Saccharomyces Genome Database (SGD), as explained in the methods. We have removed this unnecessary and unclear expression.

In the revised result section:

“Most emerging ORFs (>95%) are annotated as Dubious or Uncharacterized (**Methods**).”

In the revised methods section:

“Annotation status of emerging ORFs was downloaded from SGD. At least 95% of emerging ORFs are annotated as dubious or uncharacterized according to both the 2011 *S. cerevisiae* genome annotation (consistent with ref³) and the current one (R64-2-1).”

Comment 13 – Addressed above in Abstract section

p5I15:

this is a very strong and essential part of the study which I liked a lot, it should feature more prominently in the abstract.

Comment 14

p7I18:

ref 22 is on orphans, not de novo.

We thank the reviewer for drawing our attention to this, and edited this sentence accordingly:

“Across kingdoms, one type of evolutionary change that typically accompanies the maturation of young genes is an increase in expression level²⁸.”

Comment 15

I24: “.. outcomes OF laboratory ...”

We thank the Reviewer for pointing out this typo. We have modified the sentence for clarity in our revised manuscript:

“Systematic overexpression screens have been shown to identify adaptive mutations that also occur in laboratory evolution experiments²⁹”.

Comment 16

p9I35:

e.g. here mention Peisajovich, also in the discussion

Thanks to your suggestion, we cite this manuscript in our revised discussion. The sentence p9I35 of the submitted manuscript, however, refers to a primary analysis that we performed ourselves on the specific ORFs we are studying, and reference the corresponding supplementary figure. We did not add reference to this manuscript in this particular sentence by fear it could be misinterpreted.

Comment 17

p10I18:

did not understand this sentence -- please disentangle

We thank the reviewer for drawing our attention to the wording here. We have rewritten the sentence to improve clarity.

“Disruption of the 28 adaptive emerging ORFs appeared similarly inconsequential for fitness as disruption of other emerging ORFs, both in laboratory and in natural settings ($P > 0.05$ when comparing fitness cost of deletion, ORF intactness and nucleotide diversity across isolates).”

Comment 18

p11I7:

see above on my comment on Wilson

We agree with your comment above and we have re-written this section to make the complexity of the literature on the subject of disorder more apparent:

“It has been suggested that high levels of intrinsic structural disorder may be associated with adaptive fitness effects⁴, and it was recently shown that random sequences with high intrinsic structural disorder have low aggregation propensity and are generally well-tolerated by cells¹⁹. However, the relationship between disorder and *de novo* gene birth is debated^{3, 8, 33, 34, 35, 36, 37}. In *S. cerevisiae*, in particular, recently-evolved ORFs are predicted to be less disordered than conserved ones^{3, 8, 34, 36} and increasing the expression of disordered proteins causes deleterious promiscuous interactions³⁸”.

Comment 19

p11|18:

"However ..." I agree but this is in contradiction to the authors' statement that there is agreement in the Introduction.

We believe this apparent contradiction was caused by our imprecise use of language in the submitted manuscript. We have corrected this, as in our answer to this Reviewer's comment 8.

Comment 20

p11|14:

Out data ... indicate ... (data is plural, datum is singular)

We thank the reviewer for pointing out this typo, which we corrected. We also ensured that data is correctly used as plural throughout our revised manuscript

Comment 21

p13|20

ref34 is PhD thesis of the first author. I could not find it online and i am not sure if this qualifies as a valid reference.

To our knowledge there is no editorial guideline that prevents us from citing a defended PhD thesis. The thesis is available from this link: <https://www.theses.fr/2016PA066342> . It unfortunately is not open access, similar to multiple other references we cite. We are able to provide a copy to the Reviewer(s) if they would like to read it.

Comment 22

p14|12

Again a good occasion to cite Peisajovich.

l13

Again, be more careful with phrasing interpretation in a way one might expect a pre-existing plan of what should emerge from the genome (and what possibly not). The process may well be neutral with respect to the single protein but improve the functionality e.g. of other proteins in the membrane by binding weakly and reversibly.

We completely agree with the reviewer and are grateful for this writing suggestion. We present in our revised manuscript a more careful framing of our results in Figure 5 as hypothesis-generating:

“This discovery converges with our finding that overexpressing emerging ORFs with TM domains tends to increase relative fitness (**Fig. 4**). Together, they suggest the plausibility of a TM-first model of gene birth, whereby thymine sequence biases in intergenic regions that pre-date the acquisition of translation signals may facilitate the emergence of adaptive proto-genes with TM domains (**Fig. 5d**).”

Comment 23

p16l27

How recent is recent? Yeast phylogeny is rather old unless the strains are considered. Could you give an estimated time in mya?

Thank you for pointing this out. To address this Reviewer’s concern we now provide the oldest timetree estimate (<http://www.timetree.org/>) in our revised manuscript (18Mys).

Comment 24

p16l30

What has been reconstructed?

We thank the Reviewer for pointing out this ambiguity in the writing and have edited the passage as indicated below to improve clarity.

“Ancestral reconstruction of the genomic region along the clade...”

Comment 25

p18fig6

Great experiments! But I recommend they are double checked by a yeast expert.

We are glad these experiments were appreciated by the Reviewer. We hope the multiple additional experiments we added to the revised manuscript in answer to Reviewer 3’s questions will also be appreciated.

Comment 26

P2015-

A gene can not "actively establish itself" as it has no purpose and no direction. Please remove panglossian glasses.

Thank you for catching this. We have rephrased the corresponding sentence accordingly:

“Thus, our screening possibly captured *YBR196C-A* in the process of becoming established in the *S. cerevisiae* genome, after going through substantial changes since *YBR_Initial* may have first presented its TM domains to the action of natural selection.”

Comment 27

I19

compared <-- inspected (no formal evaluation has been carried out -- this is just to avoid ambiguous understanding e.g. with structural biologists)

We thank the Reviewer for drawing our attention to the possibility for semantic confusion. We have made the recommended change.

Comment 28

p21I22 What is "others" and why?

We thank the Reviewer for pointing out the lack of clarity here. The specific sentence changed in our revised manuscript but we still made sure to address this. The “others” referred to previously are proto-genes without TM domains.

“Our analyses suggest that a simple thymine bias suffices to generate a diverse reservoir of novel TM peptides (**Fig. 5a-c**), and that incipient proto-genes with TM domains are more likely to increase fitness than proto-genes without TM domains (**Fig. 4**).”

Comment 29

p23I11:

Please include latin name for the zebra finch

We significantly shortened this discussion in answer to Reviewer 3 and no longer mention individual species by name.

REVIEWER #2

General Remarks

Understanding how proteins can evolve de novo is an exciting challenge. In this paper, the authors divided the known ORFs in *S. cerevisiae* into established ORFs and emerging ORFs with emerging ORFs being those with less homology to known genes and weaker protein-coding signatures. Indeed, emerging ORFs were shown to be shorter and lowly transcribed, compared to established genes. Then, the authors measured the fitness effect of overexpressing these ORFs as well as disrupting them. The authors argue that roughly 10% of the emerging ORFs increase fitness when overexpressed (in at least one of the tested conditions) compare with 3% of established genes. In addition, a larger fraction of emerging genes were found to be dispensable, compared to established genes.

Focusing on emerging genes that increased the yeast fitness upon overexpression, the authors found a significant tendency for them to include transmembranal features. The authors also show some compelling evidence to support the TM-first hypothesis in which the TM features actually occur prior to the ORF, simply due to some local sequence biases and only then ORF essential features are added. Although the findings on the TM proto genes begs for a deeper insight into the mechanisms by which the cells benefit from these genes, the authors raise some interesting hypotheses, the most interesting being that the membrane provide a “safe” place for new proteins to evolve without being selected against due to harmful interactions with other functions in the cell.

This paper tackles very important and fundamental questions with a systematic set of experiments and analyses. First how de novo genes might evolve? and second, what is the nature of this (not so small) fraction of genes in the yeast genome that are annotated as dubious or uncharacterized. I think that this paper may be somewhat controversial, but if the authors can address my questions about measuring fitness effects of over-expression then I think that the paper would be a good addition to Nature Communications.

We are very grateful for the Reviewer’s enthusiasm about our manuscript, and

also for their constructive comments that greatly helped strengthen our manuscript and the results we now report.

Major Concern

My primary question has to do with the 5% of emerging ORFs that were found to increase fitness when overexpressed. What is the false positive rate in this identification? In particular, if the authors were to analyze the set of emerging ORF strains without overexpressing then what fraction of the ORFs would have a similar (measured) fitness increase? My concern is that overexpressing genes is known to often lead to a fitness defect, so the fact that 5% is greater than 1% is expected and does not on its own mean that the emerging genes have a propensity towards providing a fitness benefit. Instead, the 5% number needs to be compared to some appropriate control set of strains (and the accompanied analysis).

We thank the Reviewer for raising the very important issue of false positive rates.

False positives in our experiments could come from two sources:

- ***technical false positives coming from shortcomings of our pipeline***
- ***biological false positives coming from yeast strains growing to bigger colony sizes than expected for reasons unrelated to the overexpression plasmid (for instance, background mutations in the genomic DNA).***

As suggested by the Reviewer, we performed an experiment that allows estimation of the false positive rate from these two sources combined by measuring fitness for the strains without overexpressing.

Starting from the BarFlex overexpression strains, we arrayed them on plates containing 5FOA, a chemical that renders cells containing the URA3-marked overexpression plasmid inviable. This “plasmid loss” procedure results in an arrayed strain collection that maintains the same background genomic DNA but now lacks the overexpression plasmids. We then subjected this plasmid-free collection to our pipeline for measuring relative fitness in SC-URA+GAL media.

From these analyses we found that 0.08% of strains appeared as adaptive and 2.65% appeared as deleterious. In contrast, when overexpressing, the fractions of strains found adaptive and deleterious in the same growth media were orders of magnitude greater (adaptive: 5% for emerging and 1% for established ORFs; deleterious: 20% for emerging and 40% for established ORFs). These very low false positive rates show that our pipeline has high specificity and that the strains in the BarFlex collection very rarely exhibit strong growth phenotypes when not overexpressing the ORFs on the URA3-marked plasmids.

We are grateful for the opportunity to include this experiment and the associated analyses in our revised manuscript, as these controls considerably strengthen our results.

Mention of these controls is made briefly in our revised result section:

“We deployed our screening strategy on a plasmid-based overexpression collection³⁰ containing 285 emerging ORFs and 4,362 established ORFs (**Figs. 1b, 3a; Source Data 1**), having verified that the presence of an overexpression plasmid did not lead to a detectable growth defect relative to a plasmid-free strain (**Supplementary Fig. 2**) and that our strategy could detect significant changes in relative fitness with high specificity (False positive rate for increased relative fitness: 0.08%; for decreased relative fitness: 2.65%; **Methods**).”

In addition, we provide a detailed explanation in our revised methods section:

“Estimating specificity of our screening and analysis strategy.

We aimed to estimate the proportion of strains from our overexpression collection that could be erroneously found to increase or decreased relative fitness using our screening and analysis strategy. We considered that false positives could arise either due to technical reasons or biological reasons. Technical false positives could stem from inherent variability in our screening platform and colony size analysis, or from our statistical method for identifying strains with significant relative fitness effects. Biological false positives could stem from strains exhibiting true relative fitness effects where the effect would not be linked to the overexpression plasmid. To estimate a false positive rate combining all of these possible error sources, we deployed our strategy in SC+GAL+G418 growth media to a copy of our overexpression collection that we previously exposed to SC+GLU+G418+5FOA for 48hrs to chase the plasmids out of every strain. We repeated this experiment twice. One of the replicates detected 0 strains as showing any significant relative fitness effect. The second replicate detected spurious increases in relative fitness for 0.15% of the strains and spurious decreases in relative fitness for 5.30% of the strains. On average, we therefore estimate that the combined biological and technical false positive rate of our assay is 0.08% for increases, and 2.65% for decreases, in relative fitness. These estimates are an order of magnitude lower than what is observed when the strains are overexpressing emerging and established ORFs in SC-URA+GAL+G418 (**Fig. 3b-c**), demonstrating the high specificity of our strategy. The quality of our dataset is further supported by the fact that the same emerging ORFs tend to be found as increasing relative fitness in multiple environments at a rate much higher than expected by chance (empirical $P < 0.00001$; **Supplementary Fig. 4c**).”

Comment 1 – Addressed above in Abstract section

1. P. 2 line 11 - “enriched in beneficial effects” might be too strong of a statement as ~10% of these sequences were actually shown to improve fitness

and the “beneficial effects” were not characterized in depth. Therefore, a modified statement might be more accurate.

Comment 2

2. P.5 Line 18 – The phrasing could be clearer. It might be confusing to say that the fitness measurements for deletions of emerging genes were ‘higher’ than those of established genes instead of stating directly that only a small fraction (8%) of emerging genes deletions resulted in decreased fitness while the vast majority of deletions did not affect the fitness, unlike established genes for which 29% of deletions had a detrimental effect on fitness.

Also, when the authors report that disruption of emerging genes had lower effect on the fitness compared with established genes. It might be worth noting that emerging genes are expected to reduce the fitness much less than established genes, especially in benign lab conditions, where their function might not be needed

We thank the Reviewer for drawing our attention to the lack of clarity here. We have modified the text to address the specific concerns raised as follows:

“Fitness cost estimates were markedly lower when comparing emerging ORFs to established ORFs, as expected for loci that lack evidence of encoding a useful protein product (Cliff’s Delta $d = -0.32$, Mann-Whitney U test $P=1.5 \times 10^{-17}$). For example, only 8% of emerging ORFs were associated with even a small fitness cost ($n=19$; mutant fitness estimate < 0.9), relative to 29% of established ORFs ($n=1,290$; Odds ratio = 0.2; Fisher’s exact test $P < 3.6 \times 10^{-15}$) (Fig. 2a).”

We also clarify that these observations are expected:

“Altogether, our results confirmed that disrupting emerging ORFs is generally inconsequential for survival of yeast in both laboratory and natural settings, as expected for loci that lack evidence of encoding a useful protein product, and consistent with a lack of selected effects.”

Comment 3

3. P.6 line 3 and Fig. 2 – The authors should explain why essential genes (which I assume most in not all of them are established genes) were excluded from the analysis, as in Fig. 2A as the reader expects a peak of essential genes deletions with a fitness value of 0.

The reviewer is indeed correct that essential genes are encoded by established ORFs. Fitness upon loss was estimated based on the extensive work presented by Costanzo et al, 2016, who used a high-throughput approach to measure fitness at the genome scale. Essential genes were included in this work, and thus in our analysis, in the form of hypomorphic alleles. This was necessary to maintain cell viability while studying the consequences of reduced function for these loci. As such, the fitness estimates for these genes are higher than 0, which explains why there is not a peak at 0. This strain choice leads to a global underestimation of the fitness costs associated with loss of established ORFs, which we had not pointed out in our originally submitted manuscript. To clarify this matter, we have made explicit in the main text of our revised manuscript that different types of genetic strains were constructed for essential vs non-essential genes:

“To this end, we first examined fitness estimates generated from a large collection of systematic deletion (non-essential ORFs) and hypomorphic (essential ORFs) alleles²⁶”

And clearly stated that these methods of fitness measurement underestimate the fitness costs of established ORFs:

“The true difference in fitness costs between emerging and established ORFs is more pronounced in reality, given that hypomorphic alleles were used for essential ORFs instead of deletion alleles, which would have been lethal.”

We also made the appropriate changes to the legends of the corresponding figures (Fig. 2 and Supplementary Fig. 1).

Comment 4

4. P.6 line 15 – It might help the readers if the authors add a reference to the mentioned notion that emerging ORFs are primarily young established genes that for which we haven't yet found a beneficial function or the conditions in which they are beneficial.

Also, while the findings from natural isolates indeed argue against the mentioned notion, the findings on deletion of emerging ORFs in laboratory conditions might still be due to a beneficial function that is carried out by the emerging genes, yet not in the tested lab conditions.

We wholeheartedly agree with the Reviewer and edited our manuscript accordingly:

“Altogether, our results confirmed that disrupting emerging ORFs is generally inconsequential for survival of yeast in both laboratory and natural settings, as expected for loci that lack evidence of encoding a useful protein product, and consistent with a lack of selected effects. The findings in natural isolates, in particular, show that emerging ORFs evolve under weaker selective pressures than established ORFs. It is thus unlikely that emerging ORFs correspond to canonical protein-coding genes whose physiological implications outside of the laboratory remain to be discovered^{24, 25}”

Comment 5

5. Fig. 3 – Odds ratio (on Y-axis) might be better if shown in log scale.

Also, as stated previously, the authors should report the false positive rate, i.e. what fraction of genes would anyway be detected to increase fitness.

Thank you for this suggestion. We have included the Odds Ratio plots in log scale in our revised Figure 3:

We also stated explicitly our false positive rate in the revised methods section, as detailed in answer to this Reviewer’s major concern.

Comment 6 – Addressed in Open questions revision above

6. The effect of overexpression on fitness is indeed a good indication for beneficial emerging genes (P.7 Lines 17-23), yet the authors should elaborate on why only ~10% of emerging genes show benefit (in at least one of the conditioned tested) when overexpressed (Fig. 3e). Is this the expected fraction? Or are there reasons that can explain why this fraction might actually an underestimate. In

particular, performing more such experiments in diverse conditions is expected to “shift” more emerging genes from the unchanged/deceased bins to increase fitness.

Comment 7

7. P. 9 line 34 – The authors should report the degree of overlap between the genes that were picked up by the pooled experiment to those found by the colony size experiments.

We thank the Reviewer for this comment. Unfortunately, the pooled experiment evaluated competitive fitness for each overexpression strain, yielding an absolute value that does not allow to infer relative fitness from comparison with a reference strain. Thus, a defined set of ORFs with increased relative fitness cannot be directly defined from the competitive experiment and compared with our set. In fact, the reason why we developed a novel screening strategy was specifically because there was no existing methodology for high throughput measurements of relative fitness described in the literature.

That being said, we can calculate a correlation between an individual strain competitive fitness from the pooled experiment and an individual strain colony sizes from our experiment in the same media. We find that they agree very well, and have added this information in the methods section of our revised manuscript:

“Our measurement of normalized colony size in SC-URA+GAL+G418 were highly correlated with previously published competitive fitness estimates³⁰ based on barcode signal intensity readings for the same overexpression collection grown in liquid media of the same composition for 20 generations (Pearson Correlation Coefficient: $R=0.73$, $P < 2.2e-16$).”

Comment 8 - Addressed in Open questions revision above

8. P. 10 lines 18-24 – The authors should elaborate on why expression levels of beneficial emerging genes have not already increased in evolution. Promoter mutations or duplication of the gene in need are very accessible means in evolution, yet they have not occurred despite the fitness benefit reported by the authors.

Comment 9

9. P. 14 line 3 – please explain what is meant by “non-random TM propensities”

Thank you for pointing out how confusing this sentence was in our originally submitted manuscript. We have increased clarity by describing the observations in more detail in our revised manuscript:

“A strong influence of thymine content on TM propensity was observed regardless of ORF emergence status, ORF length, or whether the sequences were real or scrambled (**Fig. 5b, Supplementary Fig. 7b, Source Data 4**). Established ORFs appeared depleted in TM domains given their thymine content (Fisher’s exact $P < 2.2 \times 10^{-16}$, Odds Ratio: 0.54); yet, for those with TM domains, the fraction of sequence length predicted to be in the domains was higher than expected from their thymine content (Cliff’s Delta $d = 0.37$, Mann-Whitney U test $P < 2.2 \times 10^{-16}$) (**Fig. 5a**). In contrast, emerging ORFs and iORFs appeared enriched in TM domains relative to their thymine content (iORFs: Fisher’s exact $P < 2.1 \times 10^{-16}$, Odds Ratio: 2.1 ; emerging ORFs: Fisher’s exact $P = 0.02$, Odds Ratio: 1.3).”

Comment 10

10. P. 19 line 24-25 – Do the authors have a reason to believe that the puncta are not an artefact of GFP aggregation? Also, please explain what does it mean that colocalization was observed with Scs2p, but not with Sec13P.

We deduce that the puncta are caused by Ybr196c-a, and not by GFP aggregation, because GFP does not aggregate when expressed on its own from the same promoter, nor when fused with other proteins such as our control Ybr035c for example. The fact that we observe colocalization in the puncta with Scs2p but not Sec13p might suggest that Ybr196c-a and Scs2p physically interact, directly or indirectly, but we chose not to discuss it because we do not have further evidence to support this hypothesis at this point. The biochemistry we now present in the revised manuscript further supports that Ybr196c-a is a bona fide transmembrane protein, with a limited cytosolic pool present in the cells, which would also argue against with a GFP aggregation hypothesis. The results of these biochemistry assays are shown below:

- c.** Membrane association assay. Cell lysates were fractionated by centrifugation into a cytosolic fraction (S1) in the supernatant (S1) or a pelleted membrane fraction (P1). The pellet was suspended in either lysis buffer (LB) and centrifuged to generate S2 and P2 fractions, or buffer containing 6M urea, which solubilizes peripheral membrane proteins, to generate S3 and P3 fractions. Fractions were then subjected to SDS-PAGE and immunoblotted with anti-GFP (to detect Ybr196c-a), anti-Sec61 (an integral ER-membrane protein), and anti-Pdi1 (an ER luminal protein) antibodies. A dot indicates the full-length Ybr196c-a-EGFP fusion protein and the bands of lower MW are degradation products of this fusion. * indicates a spurious soluble protein of higher MW than Sec61 that is recognized non-specifically by the anti-Sec61 antibody.
- d.** Carbonate extraction assay. Cellular membranes were treated with a buffer control (S1/P1), Na_2CO_3 (S2/P2) to extract peripherally associated membrane proteins or luminal proteins, such as the ER luminal protein Pdi1, or 1% SDS (S3/P3) to at least partially solubilize integral membrane proteins, such as the Sec61 integral membrane protein control. Integral membrane proteins such as Sec61 and Ybr196c-a remain in the pellet fraction post carbonate treatment (P2), unlike soluble proteins like Pdi1 which shift to the solubilized supernatant (S2). Fractions were assessed by immunoblotting as in **Fig. 6c**.

Comment 11

11. P. 22 lines 18-20 – This explanation is interesting, but the fitness measurements were not done in stressful conditions so this can't explain the benefit of over-expressing the emerging genes, can it?

The Reviewer is correct that the stress conditions employed in the articles we cite and in our experiments are different in nature. The growth conditions in our experiments are only stressful to the extent that they consist of changes of carbon and nitrogen sources, which are known to induce a mild stress response. In our revised manuscript, we add more qualifiers to this speculation:

“Expression of TM proto-genes may cause a preconditioning stress, modestly inducing the unfolded protein response, or the expression of heat-shock proteins or other protein chaperones. This type of preconditioning stress has been shown across species to confer a benefit

November 21, 2019

in responding to subsequent exposure to stressors^{62, 63, 64, 65, 66}, which might extend more generally to nutrient-related stresses such as the ones used in our experiments⁶⁷”.

Comment 12

12. P. 7 line 24 – it should be “outcomes of laboratory evolution”.

We thank the Reviewer for pointing out this typo. We have modified the sentence for clarity in our revised manuscript:

“Systematic overexpression screens have been shown to identify adaptive mutations that also occur in laboratory evolution experiments²⁹”.

Comment 13

13. P. 11 end of line 12 - ‘that’ should be ‘than’.

We thank the Reviewer for pointing out this typo, which we have corrected.

REVIEWER #3

General remarks

First, to provide context to my comments, I have expertise in protein chemistry, evolution, ancestral reconstruction, etc., but I’m not an expert in yeast physiology and genetics.

Overall, I thought the paper was novel, interesting, cleverly planned and likely to advance our understanding of an important aspect of evolution. However, the paper did rely a lot on relatively low accuracy bioinformatics predictions in parts, and there are places where some experimental testing would have enabled stronger, less speculative conclusions to be drawn.

The biggest conclusion (not really in the abstract but hard not to draw from the paper/discussion) seems to be that these emerging ORFs confer some sort of biophysical selective advantage to the cells, in that they are unlikely at this stage in their evolution have any specific biochemical function in signalling/catalysis, etc., but do seem to be useful to have around in some conditions. It would be nice to have stronger support for this - some suggestions are listed below. It makes a lot of sense that the

weak selective pressure to keep these emerging TM proteins around could be that they are weakly beneficial to survival by modulating membrane properties or URR, and from this, some can evolve to develop specific biochemical functions. But, I can see why the authors have been conservative because the data here doesn't quite establish this.

Thank you very much for showing such enthusiasm for our manuscript. The exciting notions of biophysical advantages of TM proto-genes mentioned by the Reviewer constitute indeed, in our view, new hypotheses that emerge from our results, rather than conclusions to be explored here per se. We would like to remain conservative in our writing at this stage because we have not shown how the overexpression of novel TM domains might increase growth. We have significantly re-organized the discussion of our revised manuscript to clarify this.

We performed many of the analyses and experiments suggested by this Reviewer, which did get us a little closer to understanding how the novel TM proteins that we discovered may be advantageous. We are very grateful for the suggestions this Reviewer gave us, and glad that the results of some of the additional experiments we performed in response are now integrated in our revised manuscript. As a result, the description of the YBR196C-A locus has been extended from one figure in the submitted manuscript (Fig. 6) to three figures in our revised manuscript (Figs. 6, 7, and 8).

Comment 1 – Addressed above in ‘Abstract’ section

1) The abstract sort of evades the more interesting/controversial aspects of the paper and the last sentence just says novel genes with useful biochemical capacities (whatever this means – I think I'd say biophysical) tend to evolve within intergenic loci that have a blueprint for these capacities (this is also vague). I think this could be rewritten to be less vague and maybe more specific to the data in the paper?

Comment 2

2) There are a lot of statistical tests used to support some conclusions, but when the data looks a bit less clear cut there seems to be less enthusiasm for statistical analysis. Fig 3 is a good example. First, I think it is nature policy to present raw data and that bar graphs with error bars are not acceptable. Fig 3BCDE should all be presented with raw data points shown. If appropriate, some statistical analysis should be performed. E.g. in 3C – it is claimed that overexpression is 4.5 times more likely to increase relative

fitness. It is not obvious to me what unchanged is in this graph, if it is an odds ratio – what is unchanged? Wouldn't this be 1? Probably there is a simple answer but I was a bit confused. Regardless – it looks like saying overexpression is 4.5 times more likely to increase fitness is a simplification – just guessing by the limited data presented it looks like in many cases it doesn't do much, but sometimes does a lot? This is why simplifications of the data (e.g. average plus CI) can be misleading... same comment goes for all the bar graphs throughout.

We wholeheartedly agree with the Reviewer that it is better to show the raw data than simplified statistics. This is the policy we followed throughout our submitted manuscript, with the exception of Figures 4 and S5, where we had chosen to summarize panels abcd with averages to keep visually consistent with panels ef, which represent fractions of ORFs (so there are no raw data besides the value of the fractions shown in the bar graphs). In our revised manuscript, we changed the representations for panels ab in order to display all the raw data, and removed panels cd since the raw data was already presented in Supplementary Figure 6 in the form of distributions. The revised figure panels for Figure 4 are as follows:

Fig. 4: TM propensity is associated with beneficial fitness effects in emerging ORFs

- a.** High disorder is not associated with adaptive fitness effects. The distributions of the fraction of ORF length predicted to encode disordered residues (Disorder content) in adaptive, neutral and deleterious emerging ORFs are shown as violin plots. Adaptive emerging ORFs appear slightly less disordered than neutral and deleterious emerging ORFs (Mann-Whitney U test $P=0.03$ and $P=0.02$, respectively).
- b.** High GC content is not associated with adaptive fitness effects. The distributions of the fraction of ORF length that is G/C in adaptive, neutral and deleterious emerging ORFs are

shown as violin plots. Adaptive emerging ORFs appear to have a slightly lower GC content than neutral emerging ORFs (Mann-Whitney U test $P=0.004$)

The same visualization was applied also for Supplementary Figure 5, where the same measurements are shown for established ORFs.

All the bar graphs that remain in the figures of our revised manuscript correspond to counts, fractions and odds ratios. As such, the visualizations as bar graphs is appropriate since these type of data cannot be further subdivided into data points.

In Figure 3 bcde, for instance, panels b and e represent fractions and panels c and d represent Odds Ratios. Accordingly, vertical error bars represent standard error of the proportion (panels b and e) and CI of the Odds Ratio (panels c and d), rather than SD or SEM. For panels c and d, horizontal dashed red lines indicate Odds ratios of 1, where the likelihood to display increased, decreased or unchanged fitness relative to the reference strain would be equal for established and emerging ORFs.

The “unchanged” bar in Figure 3c being above 1 shows that the odds of having no detectable effect on relative fitness are greater for emerging ORFs than for established ORFs. This is shown in Fig. 3b, where we can see that nearly 80% of emerging ORFs have no detectable effect on fitness (blue bar in the “unchanged” category), whereas this is only the case for ~60% of established ORFs (white bar in the “unchanged category”).

Similarly, when we write, eg in Fig. 3c, that overexpression is 4.5 more likely to increase fitness, we mean that the odds of increasing fitness (to any value) are 4.5 times greater for emerging ORFs than for established ORFs. These odds are calculated from the number of emerging and established ORFs that displayed a colony size distribution than was significantly larger than expected given the colony size distribution of the reference strain.

By essence, our classification of ORFs by overexpression fitness is categorical: they show either increased, decreased, or unchanged relative fitness. This classification scheme has the advantage of being statistically rigorous, since it involves comparing distributions of colony sizes between mutant strains and a reference strain in a controlled statistical framework (if there is a statistically significant difference, the fitness is increased or decreased; else, the fitness is categorized as unchanged). However, in turn, the categorical nature obscures differences in strength of effect between strains, as correctly pointed out by the reviewer. That said, we do show individual replicate colony size in Supplementary Fig. 3b and state the range of effects observed in the results section (7.9% to 19%). We acknowledge this is an oversimplification, and perhaps an avenue for further methods development in colony size analysis methods.

To increase clarity around the meanings of these fitness categories, we added a description in the result section of our revised manuscript:

“This strategy allowed us to identify ORFs whose overexpression significantly increased colony size relative to the reference (“increased relative fitness”), ORFs whose overexpression significantly decreased colony size relative to the reference (“decreased relative fitness”), and ORFs whose colony sizes were statistically indistinguishable from those of the reference strain (“unchanged relative fitness”) (Fig. 3b).”

We also extended the figure caption to improve clarity, at the risk of letting it be slightly longer than recommended in the Nature Communications editorial guidelines.

Fig. 1. Overexpression of emerging ORFs can increase relative fitness.

- a. Strategy to screen for relative fitness of overexpression strains. Yeast strains overexpressing emerging and established ORFs, and reference strains, are arrayed at ultra-high-density on plates containing agar media. Fitness is estimated from the distributions of colony sizes of technical replicates. Number of colonies are rounded to the nearest hundred. See **Methods**.
- b. Fraction of emerging (blue) and established (white) ORFs displaying increased, decreased and unchanged fitness effects relative to the reference. Environmental condition was SC-URA+GAL+G418 media (**Supplementary Table 1**). Error bars: standard error of the proportion.
- c. Emerging ORFs are 4.5 times more likely to increase relative fitness when overexpressed than established ORFs, and 3.1 times less likely to decrease relative fitness. Odds ratios derived from the data shown in panel b. Vertical error bars represent 95% confidence intervals. Horizontal dashed line indicates odds ratio of 1, where the likelihood to display increased, decreased or unchanged fitness relative to the reference strain would be indistinguishable for emerging and established ORFs. All odds ratios are significantly different from 1 (Fisher's exact $P < 0.00002$).
- d. Emerging ORFs are consistently more likely to increase fitness and less likely to decrease fitness than established ORFs when overexpressed in five different environments. "N": poor (-), complete (+) or rich (++) supplementation of amino-acids; C: complete (+) or rich (++) supplementation of carbon sources (**Supplementary Table 1**). Odds ratios represent the likelihood of emerging ORFs to increase or decrease relative fitness compared to established ORFs. Vertical error bars represent 95% confidence intervals. Horizontal dashed lines indicate odds ratio of 1, where the likelihood to display increased, decreased or unchanged fitness relative to the reference strain would be indistinguishable for emerging and established ORFs. All odds ratios are significantly different from 1 (Fisher's exact $P < 0.00002$).
- e. Proportion of ORFs displaying increased fitness effects relative to the reference in at least one of five different environments (adaptive ORFs). Blue: emerging ORFs (28/285); White with solid contour line: established ORFs (126/4,305); White with dashed contour line: established ORFs sampled with replacement according to the distribution of ORFs lengths and native RNA expression levels of emerging ORFs. While sampling shorter or less expressed established ORFs did marginally increase the proportion found adaptive, none of these factors was sufficient to explain the high proportion of emerging ORFs found adaptive. Error bars represent standard error of the proportion.

In summary, we apologize for the lack of clarity and we hope these modifications remedy the problem and are helpful for the Reviewer.

Comment 3

3) Page 11 – I'm not sure if beneficial biochemical capacities is the best term. Without any evidence for the ORFs being involved in anything biochemical, it is a reach. Maybe beneficial adaptive capacities? It's not clear whether their beneficial effect is biochemical or biophysical (more later).same comment goes for all the bar graphs throughout.

We fully agree with the Reviewer and have removed this adjective from the description of beneficial capacities throughout our revised manuscript.

Comment 4

4) The conclusion that they are less disordered is based on bioinformatics predictions and isn't a dramatic difference (looks like .2 plus/minus .03 vs .3 +/- .02 or something – a SI stable for this stuff would be nice so people didn't have to guess). The authors have also used standard error measurement for this, which will give the smallest estimated error, vs std dev or 95% CI, etc. They should probably use the same error measurement throughout unless there is good reason to change. Regardless, the difference is not particularly large (again we get no indication of the distribution of data points – is this skewed by a few?) to make this claim. To make a statement like this we really need to know the predicted accuracy of the algorithm and to then analyse if the predicted difference is large enough given the error of the method that it is significant... then the conclusion can be draw. Maybe it is? But it would be better for the authors if they make it clear that it is a solid conclusion. Of course, it would also be nice to test this experimentally with a handful of predicted sequences... but I appreciate this is a significant amount of work. It would only be necessary if the difference in the predicted disorder is not clearly significant given the accuracy of the algorithm (and the authors still want to draw this conclusion)

We thank the reviewer for this important comment. In fact, this conclusion is not central to our work. Other scientists had proposed that high disorder would be beneficial – although, as pointed by Reviewer 1, this proposition was based on controversial ideas. Since we measured for the first time whether emerging ORFs can be beneficial, we looked for evidence of this prediction in our data, but we did not find any support for it. The fact that there is a little difference may or may not be meaningful, but either way it does not support the idea that high disorder is beneficial since the difference we see is in the other direction.

We addressed this comment nonetheless in our revised manuscript in the following ways:

- ***As per your comment 2, we displayed all the data points instead of bar graphs and error bars***
- ***We added the effect sizes and p values of all the comparisons we made between ORF groups***
- ***We added language to clarify that this is essentially a negative result:***

“Disorder predictions suggested that the translated products of the 28 adaptive emerging ORFs were slightly less disordered than neutral and deleterious emerging ORFs (Cliff's Delta $d = -0.25$ and $d = -0.31$, respectively; Mann-Whitney U test $P=0.03$ and $P=0.02$, respectively). Our data (Fig. 4a, Supplementary Fig. 5a, Source Data 1) thus indicate that high disorder is unlikely to

be a beneficial capacity that promotes *de novo* gene birth in *S. cerevisiae*, although it may be in other lineages with differing regulatory systems³⁹.”

- ***We included references to the specificity of the disorder prediction algorithm we used:***

“Estimates for intrinsic disorder were lifted from ref³, where DISOPRED2, a prediction tool with per residue false positive rate of 3.2%⁸⁹, was used.”

Comment 5

5) “GC content was slightly lower” – some sort of statistical analysis to show whether GC content was within error or GC content was significantly lower.

The mild statistical significance of the result (Mann-Whitney U test $P = 0.04$) is indicated in the corresponding figure legend of the submitted manuscript. We made the statistics more complete and more apparent in the revised manuscript, including clarifying that this is, again, essentially a negative result:

“GC content was slightly lower in adaptive than neutral emerging ORFs (Cliff’s Delta $d = -0.24$; Mann-Whitney U test $P = 0.04$) but statistically indistinguishable between adaptive and deleterious emerging ORFs (Mann-Whitney U test $P = 0.33$) (Fig. 4b, Supplementary Fig. 5b, Source Data 1). It is thus unlikely that high GC content promotes *de novo* gene birth in *S. cerevisiae* by conferring beneficial capacities to the expression products of emerging ORFs”.

Comment 6

6) I have an issue with some of the descriptions – maybe these are standard descriptions in the field, in which case ignore these comments – what is TM residue content (nothing came up on google)? Same as the other comparisons, some analysis of the known accuracy of the algorithm vs the difference in the data is required (or experimental analysis).

We apologize for the lack of clarity and have included the definition of TM content in the results section of our revised manuscript (it was initially only defined in the figure legend):

“Similarly, the fraction of residues predicted as TM over the length of the ORFs (TM content) was significantly associated with fitness benefits in emerging ORFs”.

Furthermore, we considerably increased the amount of statistical analyses of TM properties related to this figure to address this comment.

- **We include references to the specificity and sensitivity of TMHMM and Phobius, the two algorithms used in the figure to predict TM propensities**

“Adaptive emerging ORFs however displayed a strikingly higher propensity to form TM domains than neutral and deleterious emerging ORFs, according to two prediction algorithms with high specificity and sensitivity, TMHMM and Phobius^{41, 42, 43, 44}.”

- **We include effect size and p-value for every comparison between every group ORFs using both algorithms**
 - o **Comparing fractions with TM domains (count data)**

“Comparing the proportion of ORFs with predicted TM domains between adaptive and neutral emerging ORFs yielded Odds Ratio > 2.7 and Fisher’s exact $P < 0.025$ for both algorithms; comparisons between adaptive and deleterious emerging ORFs yielded Odds Ratio > 3.7 and Fisher’s exact $P < 0.007$ for both algorithms (Figs. 4c, d, Source Data 1). In contrast, there were no significant differences between adaptive, neutral and deleterious established ORFs (Odds Ratio < 1.5 and Fisher’s exact $P > 0.06$ for both algorithms; Supplementary Figs. 5c, d, Source Data 1).”

- o **Comparing distributions of TM content (continuous data)**

“Comparing TM content between adaptive and neutral emerging ORFs yielded Cliff’s Delta $d > 0.3$ and Mann-Whitney U test $P < 0.003$ for both algorithms, and comparing TM content between adaptive and deleterious emerging ORFs yielded Cliff’s Delta $d > 0.4$ and Mann-Whitney U test $P < 0.0005$ for both algorithms (Supplementary Fig. 6, Source Data 1). This association was again negligible and insignificant in established ORFs (comparisons between adaptive and neutral or deleterious established ORFs: Cliff’s Delta $d < 0.07$ and Mann-Whitney U test $P > 0.19$ for both algorithms; Supplementary Fig. 6, Source Data 1).”

Altogether, these additional statistical analyses leave no doubt that the association between TM propensity and fitness is significant in our dataset.

Comment 7

7) Page 14 – I’m unconformable with the term preadaptation in this context. It’s also extrapolating too far to say the blueprint for the capacities (whatever these are) exists in the genome before translation. I see what is being implied but I think given this is based on a lot of bioinformatics and some nice experiments, I’d recommend being more cautious and sticking to descriptive scientific terms (rather than blueprint, etc.).

We agree and have removed mention to pre-adaptation and to blueprints here, and throughout our revised manuscript.

Comment 8

8) The next section is nice (pg 16) but it should be made clear later that this is an experiment where n=1. The data support the hypothesis, but...

We thank for Reviewer for pointing this out. We have carefully reworded our abstract, introduction, discussion, and several parts of the results section to make it abundantly clear that “TM-first” is only a hypothesis that emerges from our work, and that YBR196C-A is the only locus that we studied so far to corroborate this hypothesis.

- ***The last sentence of the revised abstract now clarifies the TM-first is only a suggested model:***

“These findings, together with in-depth examination of the de novo emerging YBR196C-A locus, suggest a novel evolutionary model whereby adaptive transmembrane polypeptides emerge de novo from thymine-rich non-genic regions and subsequently accumulate changes molded by natural selection”.

- ***We make the same point in the end of our revised introduction:***

“Approximately 10% of emerging ORFs show beneficial fitness effects when overexpressed, a 3-fold enrichment relative to established ORFs consistent with a higher potential for adaptive change. In emerging but not established ORFs, beneficial fitness effects are associated with a high propensity to encode transmembrane (TM) domains. Analyses of genome-wide TM propensities led us to hypothesize that novel adaptive TM peptides may spontaneously emerge when thymine-rich non-genic regions become translated: a “TM-first” model of gene birth. The plausibility of this model is supported by a detailed reconstruction of the evolutionary history of one locus where an ORF (*YBR196C-A*) emerged *de novo* in a thymine-rich ancestral non-genic region, accumulated substantial changes under positive selection and progressively increased its TM propensity to give rise to a protein that integrates the membrane of the endoplasmic reticulum (ER) while retaining the potential for adaptive change”.

- ***We make the same point again in the results section after we describe Fig.5:***

“This discovery converges with our finding that overexpressing emerging ORFs with TM domains tends to increase relative fitness (**Fig. 4**). Together, they suggest the plausibility of a

November 21, 2019

TM-first model of gene birth, whereby thymine sequence biases in intergenic regions that pre-date the acquisition of translation signals may facilitate the emergence of adaptive proto-genes with TM domains (**Fig. 5d**)”.

- ***In the results section, when we introduce our work on YBR196C-A, we clarify that it is an $n = 1$ test of our hypothesis:***

“We sought to test the plausibility of the TM-first model by retracing the evolutionary history of one specific locus”.

- ***In the revised discussion we clarify again that this was an $n=1$ test:***

“To date, this is the only locus whose evolutionary history has been investigated in enough detail to corroborate a TM-first model of *de novo* gene emergence (**Fig. 5d**). The TM-first model is an attractive hypothesis that may explain how sequences that were not translated previously could spontaneously exhibit secondary structures with the potential for adaptive change”.

Comment 9

9) “Failed to identify sequences similar...” by what criteria? What is similar – 20%, 25%, 35%? This means little without some context – some people might disagree passionately about whether 25% ID is or is not similar.

We thank the Reviewer for pointing out the ambiguity here. We have changed the text to clarify that a test for statistical significance has been applied, as further described in the methods.

“Extensive sequence similarity searches across a broad phylogenetic range (BLAST E-value threshold 0.001; **Methods**) failed to identify sequences similar to *YBR196C-A* in species beyond the *Saccharomyces* genus, consistent with a recent origin within the past 18 Myrs⁴⁷.”

Comment 10

10) The ancestral sequence reconstruction is used very nicely, I liked this.

We are delighted that the Reviewer was pleased with this part of our study.

Comment 11

11) Fig 6C seems to actually refer to 6A? There are a few small things like this that look as though the paper has been revised but some of these have become disorganized in the process – I'd recommend going through the paper again in detail. For example – there's an orphan MD methods section – but no MD?

The Reviewer guessed correctly that our manuscript was revised prior to submission. We are embarrassed and we apologize for the resulting disorganization of our submitted manuscript. We re-organized the methods section and combed our revised manuscript to make sure such things do not happen again.

We wish to clarify, though that an MD analysis was in fact presented in the submitted manuscript. It was shown in Figure 6G, and is now shown as Supplementary Fig. 11:

Supplementary Fig. 11. Ybr196c-a is predicted to stably integrate membranes.

Molecular dynamics simulation shows that, after 200ns, the peptide has kept the helix intact, with N and C terminal tails interacting with the surface of the lipid bilayer. Structure model from Fig. 7d.

Comment 12

12) The authors point out that the predicted TM propensity isn't very accurate... this isn't a great look when the last few pages have been using predicted TM propensities to make some quite significant conclusions... They also highlight that these prediction algorithms are trained on different proteins – which could lower their accuracy in their

use for this paper. Anyway, I couldn't actually find where the predicted TM propensity for this protein is listed?

We thank the Reviewer for these comments. It turns out that our statement regarding the accuracy of TM predictions based on primary sequences was not only misleading but also incorrect. This was shown by a study that we were not aware of when we submitted the manuscript (Tsirigos et al, NAR 2015). We have therefore removed this statement from our revised manuscript, and we instead cite references for the high specificity and sensitivity of TMHMM and Phobius.

“two prediction algorithms with high specificity and sensitivity, TMHMM and Phobius^{41, 42, 43, 44}”

We also specify explicitly the predicted TM propensity of the protein we study in depth, Ybr196c-a:

“YBR196C-A is a 150 nt uncharacterized ORF located on chromosome II with a putative TM domain that accounts for almost half of the protein length (22/49 aa).”

That said, we generally tend to trust experiments over predictions. We thus verified the TM propensity of this protein using biochemical methods: a membrane association assay and a carbonate extraction assay. The results, shown in revised Fig. 6, demonstrate that this is an integral TM protein:

- c. Membrane association assay. Cell lysates were fractionated by centrifugation into a cytosolic fraction (S1) in the supernatant (S1) or a pelleted membrane fraction (P1). The pellet was suspended in either lysis buffer (LB) and centrifuged to generate S2 and P2 fractions, or buffer containing 6M urea, which solubilizes peripheral membrane proteins, to generate S3 and P3 fractions. Fractions were then subjected to SDS-PAGE and immunoblotted with anti-GFP (to detect Ybr196c-a), anti-Sec61 (an integral ER-membrane protein), and anti-Pdi1 (an ER luminal protein) antibodies. A dot indicates the full-length Ybr196c-a-EGFP fusion protein and the bands of lower MW are degradation products of this fusion. * indicates a spurious soluble protein of higher MW than Sec61 that is recognized non-specifically by the anti-Sec61 antibody.
- d. Carbonate extraction assay. Cellular membranes were treated with a buffer control (S1/P1), Na₂CO₃ (S2/P2) to extract peripherally associated membrane proteins or luminal proteins, such as the ER luminal protein Pdi1, or 1% SDS (S3/P3) to at least partially solubilize integral membrane proteins, such as the Sec61 integral membrane protein control. Integral membrane proteins such as Sec61 and Ybr196c-a remain in the pellet

fraction post carbonate treatment (P2), unlike soluble proteins like Pdi1 which shift to the solubilized supernatant (S2). Fractions were assessed by immunoblotting as in **Fig. 6c**.

Comment 13

13) It is great that it was tested experimentally. The figure reference is wrong – in text it refers to A/B. YAR035-C comes out of nowhere here – why was it not introduced at the start of this section with YBR196C-A as another test protein, likewise YBR_IO YBR_SP could have been tested. It would be better to say, we selected 35, 196. The way it is written 35, seems to be introduced when convenient – e.g. what are the TM propensities for it? The conclusion to this paragraph again, excludes any mention of 35?

We again apologize for the confusion in panel labeling. We combed our revised manuscript to make sure such things do not happen again.

We also apologize for the confusion with 35 which seemed to come out of nowhere in our submitted manuscript. We are reluctant to say “we selected Yar035c-a, Ybr196c-a...” because we only included Yar035c-a as a control, to show that not all emerging ORFs localize at the ER. We changed the phrasing in our revised manuscript to clarify this:

“As a control, we visualized using the same methods the protein encoded by another emerging ORF (*YAR035C-A*) which has no predicted TM domains and was not found to be adaptive in our screens. We observed colocalization with the mitochondria (**Supplementary Fig. 9**). In sum, microscopy confirmed that Ybr196c-a associates with a select subset of cellular membranes.”

Comment 14

14) Pg 20 rephrase “presented blueprints for TM domains” to more scientific language.

Thank you for pointing out the lack of scientific rigor in this sentence. It is reformulated in our revised manuscript:

“Thus, our screening possibly captured *YBR196C-A* in the process of becoming established in the *S. cerevisiae* genome, after going through substantial changes since *YBR_Initial* may have first presented its TM domains to the action of natural selection.”

Comment 15

15) The frustrating thing is that 196 and 35 were experimentally tested in terms of their ability to associate with membrane, but then IO and SP were not (they could have been)

November 21, 2019

– instead we rely on predictions with Robetta being used to speculate that 196 evolved to act as a TM protein (but not the others?). These models would be more interesting and make more sense with some experimental data to provide context – e.g IO doesn't associate much – this makes sense because...

We wholeheartedly agree. To address this comment, we considerably expanded the scope of our microscopy analyses. We generated and expressed tagged versions of the initial ORF and all its descendants, then visualized them by microscopy in *S. cerevisiae*. This was a very exciting experiment, and the results are nicely consistent with Robetta simulations (where we now include the intermediate ORF rather than the *S. paradoxus* homologue in a novel Figure 7).

The expanded microscopy is shown in a novel Figure 8, which beautifully illustrates our emerging understanding of YBR196C-A evolution:

Fig. 8. Evolution of TM domain propensity throughout the process of *de novo* emergence. The sequences of extant and reconstructed homologues of *YBR196C-A* (Fig. 6a) were fused to EGFP and expressed in *S. cerevisiae*

cells with chromosomally integrated mCherry-Scs2-TM (Methods). A representative micrograph taken with consistent imaging and adjustments is shown for each, except for the strain overexpressing *YBR_Initial*, for which two micrographs are shown. The top one uses consistent imaging and adjustments, but it is not representative. Instead, it shows a rare instance where colocalization with ER is observed. The bottom one is representative for this strain, with largely cytosolic fluorescence and a few puncta. It is adjusted differently than the other micrographs because the cytosolic signals were significantly higher than those for other strains. The label “Ybr-EGFP” indicates the imaging column where homologues of *YBR196C-A* are shown.

Our interpretation of these new results is summarized at the end of the result section of our revised manuscript:

“To explore this hypothesis, we used confocal microscopy to visualize EGFP-tagged extant and reconstructed homologous ORFs in the family of *YBR196C-A* when they were overexpressed in *S. cerevisiae* cells (**Fig. 8; Methods**). We observed colocalization with Scs2-TMp at the ER membrane and in puncta again for Ybr196c-a, as in **Fig. 6a**. Similar fluorescence patterns were observed for the *S. paradoxus* homologue of *YBR196C-A*, but not for the *S. mikatae* or *S. kudryavzevii* homologues. The closest reconstructed ancestor of Ybr196c-a and its *S. paradoxus* homologue, Ybr_intermediate, also colocalized with Scs2p at the ER membrane and in puncta. In contrast, cells overexpressing Ybr_initial showed generally diffuse cytoplasmic fluorescence and intense puncta, with only exceedingly rare cases where faint ER localization could be discerned. These observations are consistent with the evolutionary and structural models presented in **Fig. 7**, supporting the hypothesis that *YBR_Initial* has accumulated adaptive changes that increased its TM propensity in the phylogenetic branch that lead to *S. cerevisiae* and *S. paradoxus*.”

In sum, the primordial TM propensity of the initial ORF appears to have increased significantly, but only in the branch leading to S. cerevisiae and S. paradoxus. Based on these new microscopy images, we made further changes to the presentation our 3D structural prediction analyses. Indeed, the images showed that the paradoxus protein localizes at the membrane of the ER, which was not immediately consistent with the 3D structure Robetta had predicted that we had shown in our submitted manuscript. We started investigating why, and we have some interesting hypotheses that require further experimental testing. We thus decided to remove this structure from our revised manuscript, leaving this investigation for a follow up study. Instead, we now introduce the structure of “YBR_intermediate” (between the initial ORF and YBR196C-A). We are thrilled by this change because this structure is even more informative. As shown in the novel Figure 7 and described in the text pasted below, addition of this structure reveals that beta sheets are progressively disappearing from the protein throughout its evolution.

Fig. 7. Evolutionary history of the *YBR196C-A* locus.

- a.** *YBR196C-A* emerged in an ancestral non-genic sequence with high TM propensity. Results of ancestral reconstruction software are summarized along the *Saccharomyces* genus phylogenetic tree. Branch lengths are not meaningful. Theoretical translations of extant and reconstructed sequences are shown, with ORF boundaries indicated by a green M and a red *. For ORF-less sequences, frames displayed were chosen for illustration purposes. Except for the intergenic ancestor, the translation of the sequence that aligns to the start codon of *YBR196C-A* is shown, in the relevant frame, and until the first stop codon is reached. TM propensity as predicted by Phobius is shown in pink.
- b-d.** Predicted 3D structures for the translation products of *YBR_Initial* (**b**), *YBR_Intermediate* (**c**) and *YBR196C-A* (**d**). Model 1 predictions by Robetta are shown. *YBR_Initial* is predicted to encode an all- β strand protein. *YBR_Intermediate* is predicted to encode a protein with β -strands and an α -helix. *YBR196C-A* is predicted to encode a protein with a long α -helix and no β -strands.

“To investigate the impact of this rapid sequence evolution at the protein level, we inspected the predicted 3D structures of the putative proteins encoded along the *S. cerevisiae* branch of the phylogenetic tree: *YBR_Initial*, *YBR_Intermediate* and *YBR196C-A* (**Fig. 7**). All three shared a conserved predicted TM domain in the N-terminal region, but this domain contained a Gly in *YBR_Initial* and *YBR_Intermediate* that mutated to Ala in *YBR196C-A* (**Supplementary Fig. 10d**). A second, Phe-rich, low complexity TM domain was predicted for *YBR_Initial* and *YBR_Intermediate* in the C-terminal region of the alignment, presenting Gly and Pro residues (**Supplementary Fig. 10d**). Gly and Pro are known for their low helix propensity in solution⁴⁸ and in a membrane environment⁴⁹, and partially hindered the formation of α -helical structures in 3D models generated with the *ab initio* prediction server Robetta⁵⁰. In fact, the protein models

for *YBR_Initial* were almost all β strands rather than helices (**Fig. 7b**). Protein models of *YBR_Intermediate* as well were either mostly β strands (two out of five models) or presented a helix at the N-terminal region and β strands at the C-terminal region (three out of five models; **Fig. 7b**). In contrast, all models of *YBR196C-A* were devoid of β strands and showed α helices flanked by Pro in the N-terminus and a pair of charged Arg in the C-terminus. Four out of five models yielded a robust TM helical structure spanning 20 residues (**Fig. 7d**). Explicit solvent molecular dynamics simulations of these 3D structures in a membrane bilayer strongly supported the potential for *YBR196C-A* to encode a single-pass membrane-spanning protein (**Supplementary Fig. 11**). These simulations suggest that evolutionary changes accumulating over millions of years since *YBR_Initial* first emerged have increased the TM propensity at this locus”.

Again, thank you for suggesting these experiments. We feel our manuscript is much improved by these additions.

Comment 16

16) Obviously, it would have been nice to have some function of 196, but probably it doesn't have a specific function, so I'm ok with results ending here. However, it would have been really nice to have one more section with some experiments testing the idea that these emerging proteins provide selective benefit in some situations through their biophysical effects on the membrane or URR or...?. Overexpressing synthetic TM helices in yeast and monitoring effect, etc.

As the Reviewer surmised, we think that Ybr196c-a is unlikely to have a function in the evolutionary sense of the term, since it is an emerging ORF. We too are very curious about the mechanisms leading to its fitness effects. However, we were unable to pursue this far enough to include in this revised manuscript. We hope to discover the mechanisms and describe them in a follow up study.

That said, we hope the reviewer will agree that our manuscript contains enough novel findings to leave this (and other) question open:

- ***we present a novel method for measuring relative fitness***
 - ***we show for the first time that emerging ORFs can increase fitness***
 - ***we uncover a strong association between TM domains and fitness***
 - ***we uncover a widespread TM propensity in the genome***
 - ***we propose a novel model for the de novo emergence of membrane proteins***
 - ***we present the first reconstruction of the emergence of an adaptive ORF***
-

Comment 17 – Addressed earlier in ‘Open Questions’ section

17) One problem (maybe I’ve misunderstood) with the discussion is that it implies TM proteins are over represented, BUT when you look at the proteome, they are not... what about all the cytosolic or membrane associates (not TM) proteins? This point should probably be discussed.

Comment 18

18) A lot is made of Fig 4 and the overexpression results. It would be prudent to make 100% sure this is a statistically significant result.

We agree! In addition to the augmented statistical results summarized in response to your Comment 6, we introduce an entirely novel multinomial regression analysis to our revised manuscript:

“To formalize these observations, we re-analyzed the relationship between ORF emergence status, TM content and fitness measurements using multinomial logistic regression modeling. Two nested models were fitted to predict experimental relative fitness (categorical: adaptive, neutral, deleterious). In the first one, the predictor variables were TM content (continuous) and ORF emergence status (categorical: emerging, established). In the second one, we added an interaction term between TM content and emergence status. We found that adding this interaction term significantly increased the fit to the data (ANOVA. $P = 0.005$; $D = 10.7$; $df = 2$; **Fig. 4e**). In a ten-fold cross validation of the two models, the model with the interaction term consistently showed better predictive power than the model without (higher Mathews Correlation Coefficient in all ten iterations, with an average of 0.29 versus 0.23 for the model without the interaction term; Paired Wilcoxon test $P = 0.006$). The interaction term had a coefficient statistically different from zero in the comparison between adaptive and neutral ORFs (Z-test $P = 0.002$) but not in the comparison between neutral and deleterious ORFs (Z-test $P = 0.97$). Thus, this statistical modeling was also consistent with the notion that overexpression of emerging ORFs containing TM domains promotes higher fitness”.

e. Multinomial logistic regression modeling supports a statistical interaction between emergence status and TM content. Frequency of adaptive, neutral and deleterious ORFs as a function of their predicted TM content (TMHMM) is indicated by vertical bars, for established and emerging ORFs separately. Probabilities predicted at 42 values in the range [0, 1] (matching the frequency bins) by multinomial logistic regression models are indicated by lines (full: model with interaction; dashed: model without interaction). The model with interaction between TM content and emergence status is a better fit to the data than the model without interaction

In summary, this analysis shows that TM propensity has very different fitness implications for emerging ORFs than for established ORFs. We believe this leaves little doubt as to the significance of this association.

Comment 19

19) The discussion of the potential effects of the emerging proteins is very interesting, I certainly was very excited by it. But I wish there was some more testing done. Could the authors quickly test whether the overexpression was more beneficial in environments or

settings where the cells were more sensitive to membrane properties (presence of detergent?) or the URR (heat stress), etc? I can't help but feel that if they do benefit the membrane biophysics or URR, etc., there must be some way to test this?

We have begun some of these experiments and found that overexpression of Ybr196c-a became toxic in one membrane altering condition, but not in another. We are not able to draw conclusions from these exploratory forays without further experiments, which cannot be done in the timeframe of this revision. There is so much to investigate about the biology of these novel and utterly understudied ORFs, including how they affect fitness and interact with membranes. We look forward to pursuing all these questions as we further develop this research program.

Comment 20 — Addressed earlier in 'Open Questions' section

20) The last paragraph of the paper is quite long and seems a bit out of place, listing a lot of papers that describe de novo TM domains with various physiological functions. I guess this is to show that they could be doing almost anything – maybe this is right, but the paragraph needs to be altered so it relates more clearly to this work, rather than just listing previous de novo TMs and saying implications of de novo GMs generalize beyond *S cerevisiae* (I understand it is important to make the results general but this somewhat obvious, and the sentence is quite vague). This could almost go into the introduction as a single sentence.

Concluding remarks

I hope these comments help the authors to improve the manuscript (which I did enjoy reading – apologies if I've misunderstood anything) – I'm happy to see any revision and/or clarify any of my comments. I apologise for suggesting new experiments – I don't see them as essential for acceptance if the strength of the bioinformatics analysis can be more obviously shown (but I think that there are many interesting experiments to do!).

We really appreciated your comments, including experimental suggestions. We hope that the efforts we made to strengthen our analyses will be satisfactory. We apologize for not answering all the remaining open questions yet, but look forward to tackling them in future studies.

Reviewers' Comments:

Reviewer #1:

Remarks to the Author:

Vakirlis and co-authors have greatly improved their manuscript with the extensive revisions and additional experiments. I am very happy to see that my comments as well as those of the reviewers have been taken into consideration. The points raised in my initial review have all been satisfactorily addressed. I include a few additional suggestion. All of these, with exception of double checking the statistical analyses, are very minor and I leave them as suggestions that the authors can take on board as they feel appropriate.

I feel this will be an important and influential work so I encourage the authors to take the time to clarify any ambiguous or unclear passages of text (for example some of the sentences I have mentioned below). Doing so will greatly enhance an already excellent manuscript and ensure it gets the attention it deserves. I would happy to review a further minor revision if deemed appropriate by the editors.

General Responses

The re-written abstract is very well written.

Specific Responses

Comment 2 dealing with definitions and terminology

The terminology used has been clarified. However, there is still a bit of imprecise wording. For example, emerging ORFs are classified as “[lacking] ... a useful protein product”. Yet, the manuscript describes several examples of emerging ORFs which code for useful products when over-expressed.

Comment 4 dealing with disordered proteins and de novo gene birth

P13L8 “However, the relationship between disorder and de novo gene birth debated”.

I appreciate that a more rounded explanation has now been provided. However, the use of the word “debated” seems a little vague to me. To the best of my knowledge all of the citations provided suggest that young de novo genes are not more disordered than older genes. Would it be fairer to say that these studies “refute” the disordered de novo gene theory rather than “debate it”?

Figure 1.

The caption to figure 1 has been greatly improved. However, as far as I can see the “inter-species conservation” is not explained? Is it referring to conservation between species (e.g. the gene is conserved between species or is it sequence similarity between orthologs?)

Minor comments

Introduction

p 3. L 17. Sentence starting “Nonetheless, it remains unknown if, how, how often ...” Appears to have gotten somewhat mixed up.

P 4. L 5. Part of the sentence "... predicts that proto-genes differ from established genes in how mutations affecting their sequence or regulations are expected to impact fitness". This sentence is difficult to read. As it is very important for understanding the research question I suggest that special attention be paid to clarifying it and ensuring that it is easy to follow.

Results

P.6 L 14. Effect size and p values are given for two comparisons (ORF length and transcription levels). The text states that for both comparisons the statistical test are identical ("in both cases"). Is this really correct? It seems unlikely that they would give the exact same values.

P7. L 16-17 Similar to the above point, in the text two comparisons (ORF length and transcription) are said to have the same odds ratio and p-value. Can you double-check that these values are correct?

Figure 2.

Are there units for the variables used in the fig2a "Fitness upon loss" and fig2c "Nucleotide diversity" ?

P8 Last line: Sentence starting "It follows that, ..." makes a prediction based on the manuscript's central hypothesis. A citation to figure 1a is provided as support. However, I don't fully follow the logic.

P9L3 Sentence starting "Alternatively, ..." states over expression of non-genic loci (i.e. not proto-genes) should be neutral or toxic. However, would they be expected to be more toxic when over-expressed than canonical protein-coding genes as well?

Figure 5b.

Do the categories in fig5a (established ORF etc.) correspond to the panels in fig. 5b?

P18 L10. This paragraph sums up previous results on TM domains and presents a "TM-first model of gene birth". How generalization is this model thought to be? Does it apply to all species or just those with genomes similar to yeast? Is it thought to be the dominant model by which de novo genes emerge or just a subset?

P20L16 Were there any particular reasons for selecting YBR196C-A as the focal locus?

P31L7 Sentences starting "If this were the case, ... we might ask why de novo gene birth is so rare..."

I do not think de novo gene birth, in and of itself, is rare but rather the conservation of de novo genes over time is rare. In my view there is no contradiction between the two perspectives, other than "classical" molecular/structural biology would totally dismiss the idea of de novo proteins.

Considering that the early stages of de novo proteins are presumably neutral with respect to fitness, they will likely be lost again before becoming essential. Thus, I think we would need more studies on shorter time scales, possibly at the population level to pin down the early evolution of young de novo proteins. Though there we might end up with so much more chaff than wheat.

Reviewer #2:

Remarks to the Author:

The authors have addressed all our comments. We find the new version more coherent, the claims to be more justified, and the experiments' description to be better as well. The updated version of the manuscript needs no further modification and will have a unique contribution to the field of de novo gene birth.

Reviewer #3:

Remarks to the Author:

I'm satisfied with the revisions - the authors did an excellent and thorough job. The novelty and impact of the paper is not in question.

- Colin Jackson

Reviewer #1 (Remarks to the Author):

Vakirlis and co-authors have greatly improved their manuscript with the extensive revisions and additional experiments. I am very happy to see that my comments as well as those of the reviewers have been taken into consideration. The points raised in my initial review have all been satisfactorily addressed. I include a few additional suggestions. All of these, with exception of double checking the statistical analyses, are very minor and I leave them as suggestions that the authors can take on board as they feel appropriate.

I feel this will be an important and influential work so I encourage the authors to take the time to clarify any ambiguous or unclear passages of text (for example some of the sentences I have mentioned below). Doing so will greatly enhance an already excellent manuscript and ensure it gets the attention it deserves. I would be happy to review a further minor revision if deemed appropriate by the editors.

General Responses

The re-written abstract is very well written.

We thank the reviewer for his positive view of our changes.

Specific Responses

Comment 2 dealing with definitions and terminology

The terminology used has been clarified. However, there is still a bit of imprecise wording. For example, emerging ORFs are classified as “[lacking] ... a useful protein product”. Yet, the manuscript describes several examples of emerging ORFs which code for useful products when over-expressed.

We thank the reviewer for this comment. We have now added a sentence in the discussion to specify that although there is no evidence that the proteins are useful in the present state of the organism, our results suggest that they have the potential to be useful in the future.

“Overall, our results indicate that increased expression of a substantial proportion of emerging coding sequences, especially those with TM domains, is likely beneficial under some conditions. Hence, while these sequences show no evidence of encoding a useful protein product in the present state of the organism, they have the potential to do so in the future.”

Comment 4 dealing with disordered proteins and de novo gene birth

P13L8 “However, the relationship between disorder and de novo gene birth debated”.

I appreciate that a more rounded explanation has now been provided. However, the use of the word “debated” seems a little vague to me. To the best of my knowledge all of the citations provided suggest that young de novo genes are not more disordered than older genes. Would it be fairer to say that these studies “refute” the disordered de novo gene theory rather than “debate it”?

We thank the reviewer for this comment. We now use rephrased to better represent the state of the field on that question by writing that the relationship has been “contested”.

Figure 1.

The caption to figure 1 has been greatly improved. However, as far as I can see the “inter-species conservation” is not explained? Is it referring to conservation between species (e.g. the gene is conserved between species or is it sequence similarity between orthologs?)

We thank the reviewer for this comment. We have now modified the caption to clarify:

“Emerging ORFs are young (our inter-species conservation analyses found no detectable homologues outside of the sensu stricto genus and no conserved syntenic homologue identified in *S. kudryavzevii* and *S. bayanus*) and do not display strong evidence that they encode a useful protein product under intraspecific purifying selection (Methods).”

Minor comments

Introduction

p 3. L 17. Sentence starting “Nonetheless, it remains unknown if, how, how often ...” Appears to have gotten somewhat mixed up.

We have added commas and the verb “may” to help with the clarity of this complex sentence.

P 4. L 5. Part of the sentence “... predicts that proto-genes differ from established genes in how mutations affecting their sequence or regulations are expected to impact fitness”. This sentence is difficult to read. As it is very important for understanding the research question I suggest that special attention be paid to clarifying it and ensuring that it is easy to follow.

We thank the reviewer for this comment. We have now rephrased this sentence to make it clear:

“This reasoning is akin to Sartre’s “existence precedes essence” dictum²¹, and predicts that mutations affecting the sequence or regulation of proto-genes should impact fitness differently than mutations affecting the sequence or regulation of established genes.”

Results

P.6 L 14. Effect size and p values are given for two comparisons (ORF length and transcription levels). The text states that for both comparisons the statistical test are identical (“in both cases”). Is this really correct? It seems unlikely that they would give the exact same values.

The tests are in fact not identical, “both cases” refers to an inequality (both values less than the specific value indicated).

P7. L 16-17 Similar to the above point, in the text two comparisons (ORF length and transcription) are said to have the same odds ratio and p-value. Can you double-check that these values are correct?

This is a similar case as in the previous comment.

Figure 2.

Are there units for the variables used in the fig2a “Fitness upon less” and fig2c “Nucleotide diversity” ?

We thank the reviewer for this comment, however there are no units for the variables shown in this figure.

P8 Last line: Sentence starting “It follows that, ...” makes a prediction based on the manuscripts central hypothesis. A citation to figure 1a is provided as support. However, I don’t fully follow the logic.

The figure panel simply illustrates the concept summarized in the sentence.

P9L3 Sentence starting “Alternatively, ...” states over expression of non-genic loci (i.e. not proto-genes) should be neutral or toxic. However, would they be expected to be more toxic when over-expressed than canonical protein-coding genes as well?

The current state of knowledge does not allow to make such a prediction. That said, the end of the sentence, which states that such loci should “not provide fitness benefits”, is the key: if these sequences had no role in gene birth, you would not expect them to be beneficial when overexpressed.

Figure 5b.

Do the categories in fig5a (established ORF etc.) correspond to the panels in fig. 5b?

We thank the reviewer for this comment, the categories do correspond and we now explicitly write this in the legend.

P18 L10. This paragraph sums up previous results on TM domains and presents a “TM-first model of gene birth”. How generalization is this model thought to be? Does it apply to all species or just those with genomes similar to yeast? Is it thought to be the dominant model by which de novo genes emerge or just a subset?

We thank the reviewer for this comment. At that page, we present a model that emerges from our study, and speculate about its possible prevalence in the discussion.

P20L16 Were there any particular reasons for selecting YBR196C-A as the focal locus?

There were no other reasons for selecting this locus other than it was found to be beneficial, fixed in budding yeast, and it contains a TM domain.

P31L7 Sentences starting “If this were the case, ... we might ask why de novo gene birth is so rare...”

I do not think de novo gene birth, in and of itself, is rare but rather the conservation of de novo genes over time is rare. In my view there is no contradiction between the two perspective, other than "classical" molecular/structural biology would totally dismiss the idea of de novo proteins.

Considering that the early stages of de novo proteins are presumably neutral with respect to fitness, they will likely be lost again before becoming essential. Thus, I think we would need more studies on shorter time scales, possibly at the population level to pin down the early evolution of young de novo proteins. Though there we might end up with so much more chaff than wheat.

We thank the reviewer for this comment. We agree and we have modified the phrase in question to include this consideration:

“If this were the case, instead of asking how de novo gene birth is possible given the complexity of useful proteins, we might ask why de novo gene birth and retention is so rare given the pervasiveness of potentially adaptive proto-genes in the genome.”

We refrain from expanding on this point at this time since it is not directly related to our work, but we agree that more studies on shorter time scales are very important.